# Grids Often Outperform Implicit Neural Representation at Compressing Dense Signals

**Namhoon Kim    Sara Fridovich-Keil**
Department of Electrical and Computer Engineering
Georgia Institute of Technology
{namhoon, sfk}@gatech.edu

## Abstract

Implicit Neural Representations (INRs) have recently shown impressive results, but their fundamental capacity, implicit biases, and scaling behavior remain poorly understood. We investigate the performance of diverse INRs across a suite of 2D and 3D real and synthetic signals with varying effective bandwidth, as well as both overfitting and generalization tasks including tomography, super-resolution, and denoising. By stratifying performance according to model size as well as signal type and bandwidth, our results shed light on how different INR and grid representations allocate their capacity. We find that, for most tasks and signals, a simple regularized grid with interpolation trains faster and to higher quality than any INR with the same number of parameters. We also find limited settings–namely fitting binary signals such as shape contours–where INRs outperform grids, to guide future development and use of INRs towards the most advantageous applications.

## 1   Introduction

Signal representation is fundamental to sensing and learning, especially for spatial and visual applications including computer vision and computational imaging [1–4]. Recently, implicit neural representations (INRs) have shown promising performance in a range of imaging and inverse problems, yielding high perceptual quality with small memory footprint. However, their representation capacity, scaling behavior, and implicit biases are not well understood, limiting the impact and confidence with which they can be deployed. Our work aims to shed light on these properties of different representations through principled experiments, to guide practitioners toward the most suitable strategy for their specific datasets and use cases as well as inform development of future signal representations.

Modern signal representations rely on four main strategies, each with unique advantages and limitations. Interpolated grid-based representations [5–8] are fundamentally continuous, enjoy well-behaved gradients for fast optimization, and inherit classical representation guarantees rooted in sampling theory [9]. However, they suffer the curse of dimensionality: for fixed resolution, model size grows exponentially with dimension. In contrast, truly discrete representations such as point clouds and surface meshes [10–13] can use fewer parameters to represent a sparse signal, but often lack useful gradients and require heuristic discrete optimization strategies that are sensitive to initialization. INRs model continuous signals using neural networks to map input coordinates to output signal values [1, 14–18]. They can provide plausible and stable reconstructions even with very limited model size, but suffer slow optimization and poorly understood resolution and scaling behavior. Hybrid approaches merge the strengths of both grids and neural networks [19–26] by learning grid-based features alongside a lightweight neural network decoder, but questions remain regarding their representation capacity and implicit biases.

39th Conference on Neural Information Processing Systems (NeurIPS 2025).

We compare these representation methods under a comprehensive suite of conditions, including both synthetic signals (e.g., bandlimited noise, fractal structures) and real-world data (e.g., natural images, CT scans). We consider tasks including overfitting images/volumes, denoising, super-resolution, and tomography. Our contributions are threefold:

1. We quantify the capacity of state-of-the-art INRs and hybrid models by evaluating their performance stratified by signal bandwidth, signal type, and model size. For our 2D Bandlimited signal class (see Figure 1), we observe that most models exhibit a power law relationship between model size and bandwidth.

2. We identify diverse scenarios, including overfitting 2D and 3D Bandlimited signals and solving inverse problems with both synthetic and real signals, in which *simple uniform grids with interpolation offer pareto-optimal memory, speed, and reconstruction quality*.

3. We identify specific scenarios where INRs and hybrid models can outperform grid-based representations, namely *signals with underlying lower-dimensional structure*, such as shape occupancy masks.

## 2 Methods

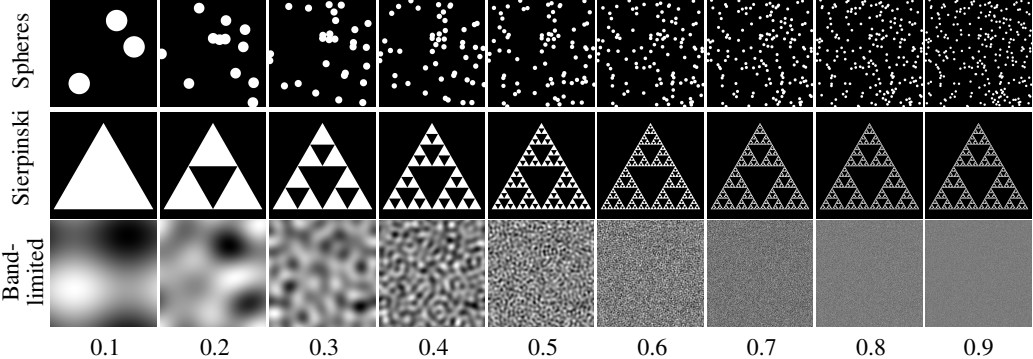

Figure 1: **Synthetic Signals.** Rows represent synthetic signal types, and columns represent effective bandlimits from 0.1 to 0.9. Signal detail and complexity increase with effective bandlimit.

Our experiments are designed to systematically assess diverse signal representations on both synthetic and real-world signals. We aim to address key gaps in understanding by evaluating each method's ability to model signals with varying frequency content, generalize and solve inverse problems in 2D and 3D, and trade off parameter efficiency, representation power, and computation time. Specifically, we evaluate each model's:

1. **Expressivity:** Ability to accurately overfit signals with varying feature scales within a fixed parameter budget.

2. **Scalability:** How performance changes with model size, particularly in terms of effective resolution and representation capacity.

3. **Computation Time:** Training and inference speeds, with a focus on how these scale with model size.

4. **Inverse Problem Performance:** Generalization performance for tasks such as super-resolution, denoising, and computed tomography (CT) reconstruction.

By systematically comparing representative grid-based, INR, hybrid, discrete models, we provide practical guidance for selecting appropriate representations for different application-specific needs.

### 2.1 Models

A broad overview of strategies for 2D and 3D signal representation in inverse problems is provided in Section 5.1. In our experiments, we compare eight representative approaches including pure INRs (Fourier Feature Networks [15], SIREN [14], WIRE [16]), hybrid methods (GA-Planes [26],

Table 1: Summary of Synthetic and Real Signals

| Dataset Type | Dimensions | Description |
|---|---|---|
| **Synthetic Signals with "Bandlimits"** | | |
| Bandlimited Signals | 2D, 3D | Random noise filtered radially in the Fourier domain with exponentially-spaced frequency cut-offs. |
| Spheres | 2D, 3D | Randomly arranged disks or balls that vary in size and number to represent features at different scales. |
| Sierpinski | 2D | Iterative fractal patterns with increasing structural complexity and fine detail. |
| Star Target | 2D | Radial triangular wedges, such that feature scale increases radially from the center. |
| **Real Signals** | | |
| DIV2K Images [27] | 2D | High-resolution image dataset for (1) image overfitting, (2) $4\times$ super-resolution along each axis, and (3) denoising with Gaussian noise (standard deviation $\epsilon \in \{0.05, 0.1\}$). |
| Computed Tomography (CT) [28] | 2D | X-ray CT scan of a human chest, used to evaluate signal recovery in a classic underdetermined inverse problem. |
| 3D Dragon Shape [29] | 3D | A solid 3D object with approximately $1 \times 10^6$ voxels (before super-resolution). Models are evaluated for (1) overfitting and (2) super-resolution (doubling the total number of voxels). |
| 3D Dragon Surface [29] | 3D | The surface of a 3D object with approximately $1 \times 10^6$ voxels (before super-resolution). Models are evaluated for (1) overfitting and (2) super-resolution (doubling the total number of voxels). |

Instant-NGP [22]), explicit representations (Gaussian Splatting [10] in 2D, via GSplat [30]), INRs with adaptive bandwidth (BACON [17]), and pure grids with interpolation (similar to [5]). Each model is tested across an exponential sweep of model sizes, with the number of trainable parameters ranging from $1 \times 10^4$ to $3 \times 10^6$ on signals with typical underlying dimension of $1 \times 10^6$. All models are tuned by optimizing hyperparameters on our Star Target image (see row 4 in Figure 2). Further implementation details are provided in Section 5.5, Table 2, and Table 3 ; the code for both signal generation and model evaluation is provided at `https://github.com/voilalab/INR-benchmark`.

**Fourier Feature Networks (FFN)** [15] embed the input coordinates $\mathbf{x} \in [-1, 1)^d$ into $2m$ Fourier features, where $m = 1000$ in our experiments. The Fourier feature mapping is defined as

$$\gamma(\mathbf{x}) = \begin{bmatrix} a_1 \cos(2\pi\omega_1^T \mathbf{x}) & a_1 \sin(2\pi\omega_1^T \mathbf{x}) \\ \vdots & \vdots \\ a_m \cos(2\pi\omega_m^T \mathbf{x}) & a_m \sin(2\pi\omega_m^T \mathbf{x}) \end{bmatrix},$$

where the frequency vectors $\omega_i$ are drawn from an isotropic multivariate Gaussian distribution with standard deviation $\sigma$ that tunes the bandwidth of the representation. In our implementation, these Fourier features are then decoded by a 2-hidden-layer ReLU MLP whose hidden dimension is scaled to vary model size while keeping the Fourier features unchanged.

**SIREN** [14] uses a coordinate multilayer perceptron (MLP) with sinusoidal activation functions to represent smooth signals at varying scales. The elementwise activation function is:

$$\psi(x; \omega) = \sin(\omega x),$$

where the scalar hyperparameter $\omega$ tunes the bandwidth of the representation. While the original paper sets $\omega = 30$ as the default, we use $\omega = 90$ based on tuning on our Star Target image. Based on the original paper, we use a 3-hidden-layer SIREN and adjust the hidden dimension to control the model size.

**WIRE** [16] uses a coordinate MLP with wavelet activation functions to capture multi-scale signal details that are localized in both space and frequency. The elementwise activation function is:

$$\psi(x; \omega, s) = e^{j\omega x} e^{-|sx|^2},$$

where the hyperparameters $\omega$ and $s$ control the frequency and spatial spread (width) of the wavelet. Based on the original paper, we use a 2-hidden-layer WIRE network and vary its hidden dimension to adjust the model size. We use $\omega = 15$ and $s = 10$ based on tuning on our Star Target image.

**GA-Planes** [26] learns features in line, plane, and volume grids (if the signal is 3D) made continuous by linear, bilinear, and trilinear interpolation. For a query point $\mathbf{q} \in \mathbb{R}^2$ or $\mathbb{R}^3$, the features are:

$$e_c = \psi(g_c, \pi_c(\mathbf{q})),$$

where $g_c$ is the feature grid for dimension tuple $c$, $\pi_c$ extracts the relevant coordinates from $\mathbf{q}$, and $\psi$ performs (bi/tri)linear interpolation. These interpolated features are then combined by multiplying combinations that yield the appropriate output dimension (line $\times$ line for 2D signals, and line $\times$ line $\times$ line and line $\times$ plane for 3D signals) and then these combined features are summed and decoded by a 2-layer ReLU MLP. The model size is scaled by varying the resolution and feature dimensions of the grids as well as the hidden dimension of the decoder MLP, following the general strategy recommended in the original paper to prioritize high resolution in lower-dimensional grids.

**Instant-NGP** [22] uses a multi-resolution hash table encoding to represent 2D and 3D signals efficiently in both time and memory. For each resolution $l \in \{1, \ldots, L\}$, the hash encoding is defined as:

$$\mathbf{e}_l(\mathbf{x}) = \psi\left(\mathbf{H}_l[\text{hash}(\mathbf{p}_l(\mathbf{x}))]\right),$$

where $\mathbf{p}_l(\mathbf{x})$ scales the input coordinate $\mathbf{x}$ to the resolution of the $l$-th grid, $\text{hash}(\cdot)$ maps grid indices to feature table indices (allowing collisions), $\mathbf{H}_l$ is the hash table for resolution $l$, and $\psi$ performs (bi/tri)linear interpolation. Following the original paper implementation, the hash features from each resolution are concatenated and decoded by a 2-layer ReLU MLP with 64 neurons at each layer. We adjust model size by varying the length of the hash tables, while the MLP decoder remains fixed.

**Gaussian Splatting (GSplat).** [10, 30] represents a scene with $N$ Gaussians $\mathcal{G}_i = (\boldsymbol{\mu}_i, \boldsymbol{\Sigma}_i, \mathbf{c}_i, \alpha_i)$. Each Gaussian is projected to screen space and rendered as a 2D density; the splats are then alpha-blended front-to-back in a fully differentiable rasterizer. Because this model does not provide a mechanism for *direct* 3D fitting, we restrict our evaluation of Gaussian Splatting to 2D signals, using the open-source `gsplat` implementation [30] to fit 2D Gaussians directly in image space.

**Band-Limited Coordinate Networks (BACON)** [17] is a coordinate-based network that applies fixed sine filters at each layer. Each layer samples a frequency range $[-B_i, B_i]$ and a phase shift, resulting in a cumulative bandwidth $\sum_{j=0}^{i} B_j$ after $i$ layers. Inputs are normalized to $[-0.5, 0.5]^d$ and the signal is treated as periodic, allowing only a discrete set of frequencies to be represented.

**Basic Interpolated Grid** directly stores parameter values at discrete lattice points in a grid, represented as $\mathbf{g} \in \mathbb{R}^{n_1 \times n_2 \times \cdots \times n_d}$, where $n_i$ represents the resolution of the grid along the $i$-th dimension. For a query point $\mathbf{q} \in \mathbb{R}^d$, the value is computed through interpolation between neighboring grid points. If the output is vector valued (e.g., RGB color channels) this is accommodated by adding an output dimension to the grid. Our 2D experiments use bicubic interpolation, which weights the 16 nearest grid values to produce a continuous and smooth (differentiable) function over 2D space. For 3D, we use trilinear interpolation which weights the nearest 8 grid values to produce a continuous but nonsmooth 3D function. Higher-order interpolation is also possible in 3D if a smooth function is desired, at higher computational cost. Further details on grid interpolation are provided in Section 5.2. We adjust model size by varying the grid resolution hyperparameters $n_i$, as the total model size scales with $\prod_{i=1}^{d} n_i$.

## 2.2 Synthetic Signals

We generate diverse 2D and 3D synthetic signals to test how well different models can overfit to representative components found in natural signals: Spheres, Bandlimited signals, Sierpinski fractals,

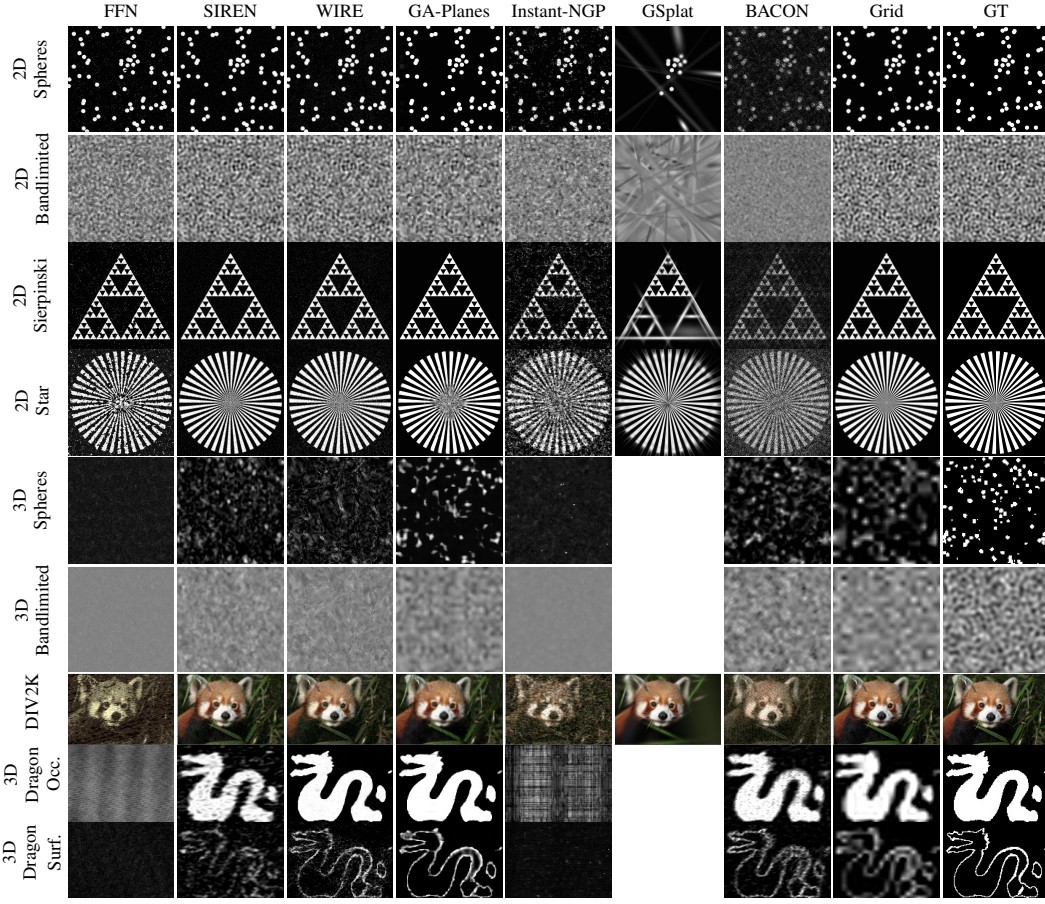

Figure 2: **Qualitative overfitting results.** Visualizations of each model on each overfitting task with $1 \times 10^4$ parameters, roughly $1\%$ of the pixels/voxels in the original 2D and 3D signals. For 3D signals, a slice is visualized. For synthetic signals with a bandwidth parameter, bandwidth 0.5 is shown. GSplat is restricted to 2D signals. Full visualizations varying model size and signal bandwidth are provided in Section 5.6. Different parameterizations induce different characteristic qualitative compression artifacts.

and a radial Star Target. Each synthetic signal has approximately $1 \times 10^6$ total spatial values, usually arranged as resolution $1000^2$ for 2D signals and $100^3$ for 3D signals. This ensures that our model parameter budgets (from $1 \times 10^4$ to $3 \times 10^6$) focus on the compressive regime while also including some overparameterized models, and that the level of compression is independent of the signal dimension (2D or 3D). Each type of synthetic signal is designed with a variable scale parameter representing qualitative or quantitative bandwidth on a scale from 0.1 to 0.9, so that we can learn how well different models fit signals with different bandwidths. These signals are summarized in the top half of Table 1 and figs. 1 and 2. More detail on our synthetic signals may be found in Section 5.3.

## 2.3   Real Signals

In addition to synthetic experiments, we evaluate each model on diverse tasks involving real-world 2D images and 3D volumes. These experiments assess the models' ability to overfit, generalize, and solve inverse problems with natural signals. We use 10 images from DIV2K [27] to test image overfitting, $4\times$ super-resolution, and denoising at two different levels of Gaussian noise. We use 7 CT scans, including a 2D chest CT scan from Clark et al. [28] and 6 CT scans from the Generalizable Dose Prediction for Heterogeneous Multi-Cohort and Multi-Site Radiotherapy Planning challenge at AAPM 2025 [31], to evaluate performance in a classic underdetermined inverse problem, in which the task is to recover an image from undersampled X-ray projections. We evaluate volumetric performance using the occupancy function and surface of the 3D Stanford Dragon [29]. On both of

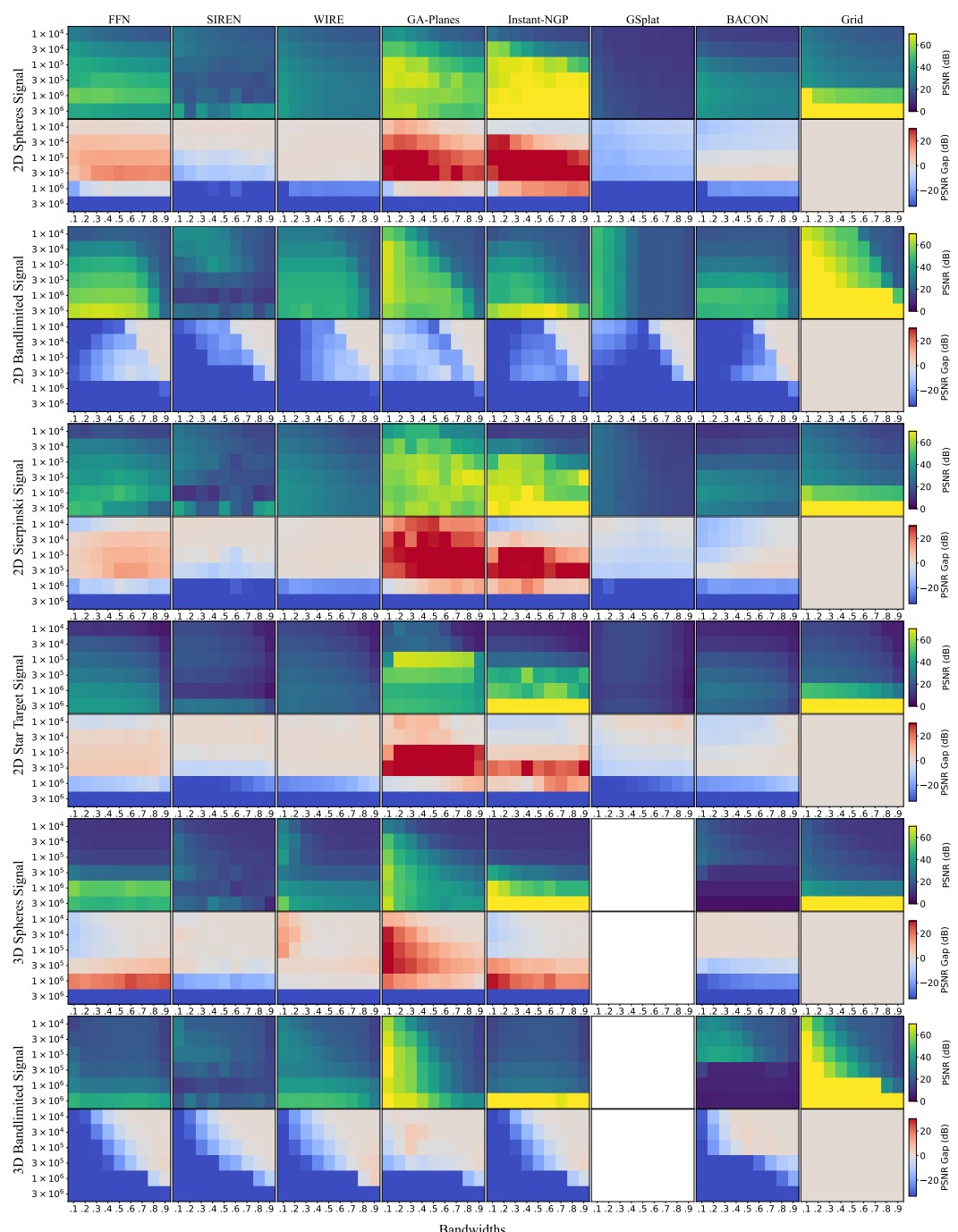

Figure 3: **Overfitting Synthetic Signals: INR and INR - Grid Heatmaps.** Red indicates regimes where other models outperform the Grid baseline; blue indicates regimes dominated by the Grid baseline. Each **first row** shows absolute PSNR values, while each **second row** shows the PSNR gap relative to the Grid baseline (i.e., PSNR - Grid PSNR). See Section 3.1 for in-depth discussion.

these 3D signals we test both volume overfitting and super-resolution. Our real-data signals and tasks are summarized in the bottom half of Table 1, with further details provided in Section 5.4.

## 2.4  Evaluation Metrics

We use peak signal-to-noise ratio (PSNR) as the primary metric, with higher PSNR values indicating better reconstruction fidelity.  All quantitative results evaluated using PSNR, SSIM, and LPIPS

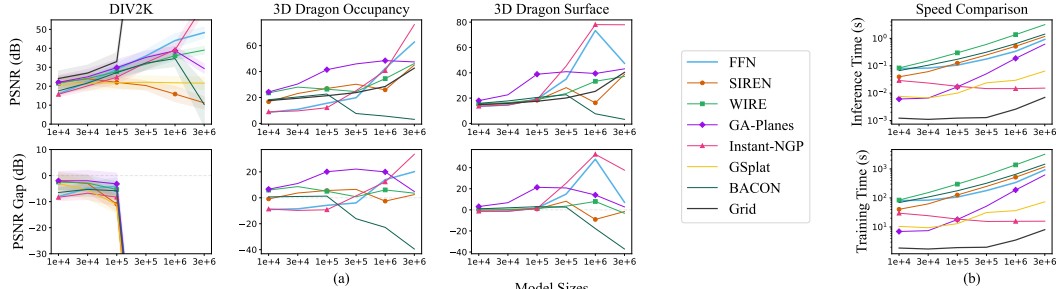

Figure 4: **Overfitting Capacity and Computational Efficiency.** (a) Overfitting capacity of different models evaluated on the 2D DIV2K and 3D Stanford Dragon datasets. PSNR trends indicate how well each model fits the training data as a function of model size (top) as well as relative performance compared to the Grid baseline (bottom). (b) Computational efficiency analysis, showing inference and training times for each model. While WIRE achieves superior overfitting capacity in some cases, it requires significantly more computation time—approximately $10\times$ that of the next slowest model.

for 2D and PSNR and IOU for 3D can be found in the appendix. Note that our Grid baseline model is unregularized for overfitting tasks but uses total variation (TV) regularization for noisy or underdetermined tasks. Additional implementation details may be found in Section 5.5.

## 3 Results

Our experiments span overfitting tasks for synthetic and real signals, as well as more generalization-oriented tasks such as computed tomography (CT), denoising, and super-resolution (SR).

### 3.1 Overfitting

We first analyze how well each model can overfit 2D and 3D signals with a fixed parameter budget. Our goal is to quantify how representation capacity scales with model size: a successful overfitting indicates that the model can capture the signal's complexity within the given parameter constraints.

**Qualitative Evaluation.** Figure 2 visualizes the outputs of all models on all overfitting tasks, for the most compressive setting where each model is constrained to at most $1 \times 10^4$ parameters, roughly $1\%$ of the raw signal sizes. For synthetic signals with variable "bandwidth" we show an intermediate-complexity signal (bandwidth=0.5). These visualizations illustrate the types of artifacts induced by compression with different signal representations. For example, Fourier Feature Networks [15] can induce aliasing-like artifacts. SIREN [14] and WIRE [16] work well for many signals but introduce texture artifacts for others, especially in 3D, while BACON suffers texture artifacts on all signals at this extreme compression level. Surprisingly, WIRE struggles to fit solid 3D Spheres but excels at representing a solid 3D dragon. GA-Planes [26] tends to yield the highest quality overfitting results at this most compressed scale, except for the highest-frequency portion of the Star Target and pure bandlimited signals, for which the grid excels. Under this extreme compression, Instant-NGP [22] suffers noisy artifacts on all signals, likely due to extensive hash collisions that are a byproduct of small hash tables. GSplat tends to represent a subset of each signal well, while failing to capture other portions, perhaps due to its random initialization and poorly-conditioned gradients for optimization. The simple interpolated Grid parameterization turns out to be quite a strong baseline across these diverse signals, with the expected limitation of blurring over super-Nyquist details that are beyond the grid resolution that is affordable under strong compression.

Comprehensive visualizations including all signal bandwidths and all model sizes are provided in Section 5.6. We specifically highlight Figure 10 which shows visual error maps for each model on the 2D Star Target. This allows us to visualize the distribution of modeling errors with respect to the scale of local image features, and how these errors change with model size. For example, errors in the Grid baseline are, as expected, concentrated at sharp edges; the Grid is the only model that successfully reconstructs the smooth background even at the smallest model size. We also observe that all models suffer the largest reconstruction errors towards the center of the Star Target, where the local signal frequency is highest. The size of this high-error region decays with increasing model size,

demonstrating that *INRs as well as hybrid, discrete, and grid representations all have an inherent bandwidth that grows with model size*.

**Quantitative Evaluation.** Figure 3 and Figure 4 report PSNR for each model on each overfitting task. Figure 3 focuses on synthetic signals with a notion of bandwidth, and stratifies performance within each heatmap according to model size ($y$ axis) and effective signal bandwidth ($x$ axis). For each signal, the top row of heatmaps reports the PSNR achieved by each model, while the second row reports the difference in PSNR between the INR, hybrid, or discrete method and the Grid baseline for each experiment. This second row allows rapid identification of the regimes (in red) in which INR, hybrid, or discrete methods can outperform the Grid baseline, versus regimes (in blue) where the Grid is the best-performing representation. First, we observe that several models, notably GA-Planes and Instant-NGP as well as some Fourier Feature Networks, can outperform the Grid baseline at intermediate model sizes on the synthetic signals with constant-value regions and sharp edges (Spheres, Sierpinski, and Star Target). We posit that these signals are easier to fit with an adaptive representation due to their underlying lower-dimensional structure, as each of these signals has its complexity concentrated at either a 2D surface embedded in 3D space or a 1D edge embedded in 2D space. However, we also observe two key takeaways evident in 2D and 3D Bandlimited signals: (1) most models exhibit an inherent bandwidth that increases with model size, apparently following a power law since both the model size and signal bandlimit are on a log scale for this signal class, and (2) *no model can reliably outperform the simple Grid baseline for 2D or 3D Bandlimited signals, regardless of model size.* This is a strong and perhaps surprising negative result suggesting that INRs are not an all-purpose solution but are instead best suited to representing specific types of signals.

Figure 4 (a) reports quantitative overfitting performance on real 2D and 3D signals including images from DIV2K [27] as well as the 3D Stanford Dragon [29] posed as occupancy and surface fitting tasks. While Grid outperforms other methods on 2D image overfitting, INRs exhibit competitive performance in 3D overfitting tasks—perhaps because, like our synthetic Spheres signal, the 3D Dragon consists of constant regions and sharp edges with underlying lower-dimensional structure. In particular, GA-Planes and WIRE demonstrate strong compression capabilities in the 3D object-fitting task, highlighting a potential use case for INR development and applications.

Figure 4 (b) compares inference and training times for different models on the 2D Star Target across various parameter budgets. The simple Grid model with bicubic interpolation provides the fastest inference and training times, pure INRs require the most computation time, and hybrid models (GA-Planes, Instant-NGP) and discrete (GSplat) models are in between. We note that the training and inference times for the Grid and hybrid models are in principle independent of model size, with some caveats depending on batch size and feature dimension (which explain the increasing trend for GA-Planes). For pure INRs there is an architecturally inherent increase in computation time with model size.

### 3.2 Generalization and Inverse Problems

Visual and quantitative results for computed tomography, denoising, and super-resolution are presented in Figure 5 and Figure 6, respectively. For all of these tasks involving natural 2D signals, we find that the simple Grid baseline with total variation (TV) regularization is optimal across all model sizes. However, on our 3D Dragon occupancy and surface super-resolution tasks, GA-Planes, WIRE, and to some extent SIREN outperform the Grid at the smallest model sizes. As in our overfitting experiments, this suggests that some INRs and hybrid models may be best suited to representing and compressing signals with constant regions and sharp edges characteristic of lower-dimensional structure. Comprehensive visualizations and quantitative metrics including an additional denoising task with $\epsilon = 0.05$ are provided in Section 5.7.

## 4 Discussion

Our results shed light on the implicit biases of different signal representation methods: how each method allocates its limited capacity across signals with varying characteristics, how this capacity scales with model size, and what visual artifacts are induced by compression with each method. Our experiments span 2D and 3D synthetic and real datasets as well as diverse tasks including overfitting, tomography, denoising, and super-resolution. By stratifying performance across signal bandwidth, model size, and signal type, we provide insights into the strengths and limitations of

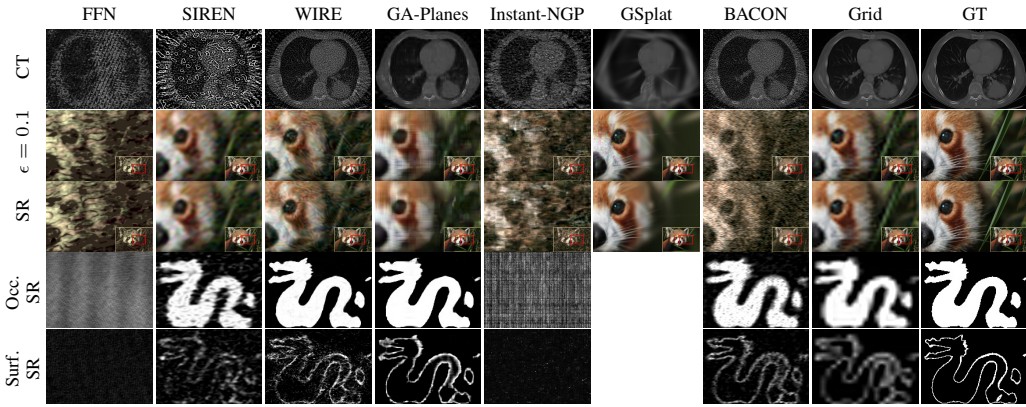

Figure 5: **Qualitative Generalization and Inverse Problems Results.** In CT reconstruction, Grid with TV regularization achieves the best results. Experiments on the DIV2K dataset are zoomed in to highlight image details. For image denoising and super-resolution, WIRE produces sharp images but introduces texture artifacts. GA-Planes outperforms other methods in volume and surface super-resolution. Artifacts in each model's output throughout the inverse problem tasks reveal their inherent structural biases (e.g., sinusoidal artifacts in FFNs and SIREN, line artifacts in GA-Planes).

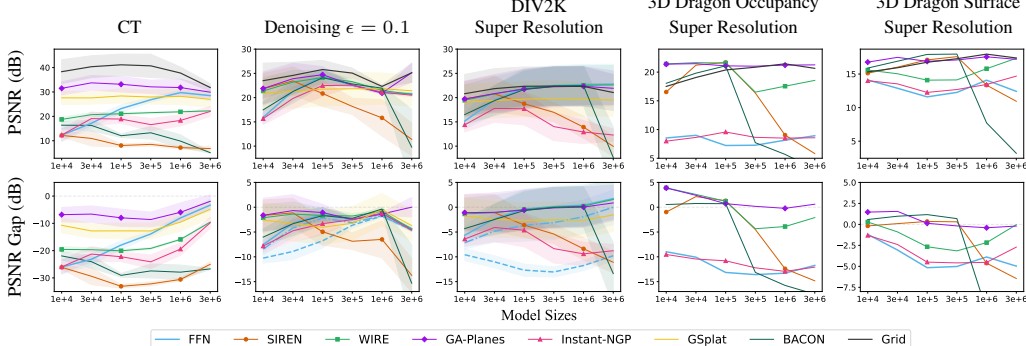

Figure 6: **Quantitative Generalization and Inverse Problems Results.** PSNR vs. model size trends indicate how well each model performs each generalization task as a function of model size (top) as well as relative performance compared to the TV-regularized Grid baseline (bottom). WIRE and GA-Planes perform best at highly compressed 3D super-resolution, but the simple Grid baseline generally outperforms other signal representation methods across these tasks.

implicit neural representations (INRs), hybrid models, discrete models, and a simple yet surprisingly strong grid-based baseline.

Our experiments reveal several key findings. First, for bandlimited signals, we observe a power-law relationship between model size and bandwidth across nearly all models, providing a tool to quantify the inherent "resolution" of INRs as a function of parameter count. For 2D and 3D bandlimited signals, a simple interpolated grid representation consistently outperforms other representations in both computational efficiency and reconstruction quality, using equal parameter budgets. In most inverse problems, interpolated grids with regularization again offer superior performance, training faster and achieving higher quality results than other models with the same number of parameters.

However, INRs exhibit advantages in specific scenarios for both overfitting and generalization tasks in 2D and 3D, specifically when the target signal has an underlying lower-dimensional structure, such as constant-value regions with sharp edges. Our results suggest that this underlying lower dimensional structure (such as a 2D surface embedded in 3D, or a 1D curve or edge embedded in 2D) is key for INRs to be able to compress. For pure bandlimited noise, the interpolated grid performs best, perhaps due to the Nyquist sampling and interpolation theorem. An intriguing aspect of our findings is that many "dense" natural signals, such as CT scans and natural images, behave more like bandlimited noise than like the "sparse" synthetic signals and shapes where there is simple underlying lower dimensional structure like sharp edges and constant regions. We believe that there is still some

underlying structure to these "dense" natural signals that should enable them to be compressed, but our work reveals that current INRs are not meeting that goal.

Overall, our results emphasize that a simple grid with interpolation remains the most practical and effective choice for a wide range of applications involving dense natural signals, offering simplicity, interpretability, computational efficiency, and in many cases superior reconstruction quality. However, INRs provide distinct advantages in representing specific types of signals, specifically those with simple underlying lower-dimensional structure such as object edges or surfaces. By analyzing these trade-offs, our study offers practical guidelines for selecting the most suitable representation method for different signal types and applications and for guiding the development of future compressive signal representations.

## Acknowledgments

This work was supported in part by the NSF Mathematical Sciences Postdoctoral Research Fellowship under award number 2303178. Any opinions, findings, and conclusions or recommendations expressed in this material are those of the authors and do not necessarily reflect the views of the National Science Foundation.

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

# 5 Appendix

## 5.1 Related Work Discussion

Representing and recovering image, volume, and higher-dimensional signals using a compact yet expressive parameterization is a long-standing goal in fields ranging from computer vision to medical imaging to scientific computing. While some recent overviews [1, 32] focus on the rapidly evolving landscape of Implicit Neural Representations (INRs), here we give a broad introduction to four main strategies for signal parameterization: (1) traditional grid interpolation, (2) INRs, (3) discrete representations, and (4) hybrid models.

**Traditional interpolation methods** [5–8] assume signals lie within a predefined space of continuous functions, and represent them through coefficients in a chosen finite basis. For example, the Nyquist–Shannon sampling theorem ensures that bandlimited signals can be faithfully reconstructed by interpolating sufficiently dense samples on a regular grid [9]. Although signals may be parameterized as discrete samples or coefficients, continuous basis functions ensure that the final representation is continuous and differentiable, unlike truly discrete representations. Interpolation techniques are valued for their fast and stable optimization and robust theoretical guarantees, including error bounds and clear scaling laws relating grid resolution and signal bandwidth. However, grid-based models struggle to capture sharp boundaries or other high-frequency details without dense sampling, making them impractical for high-dimensional data due to the exponential growth of model size with dimension.

**Implicit neural representations (INRs)** [14–16, 33, 34] have garnered significant attention for their ability to model intricate 3D+ signals as continuous functions parameterized by memory-efficient neural networks. These networks learn to map input coordinates (e.g., spatial or temporal) to output signal values (e.g., intensity, color), often leveraging a specialized activation function or input embedding to mitigate spectral bias. However, INRs are often slow to optimize, and their implicit biases, representational capacity, and scaling behavior remain understudied. While some posit that INRs possess "infinite resolution" [35] due to their continuous nature, we set out to empirically assess these properties and answer the question: When should you use an INR?

**Discrete representations** model a signal using discrete primitives such as point clouds or polygonal meshes [10, 11, 13, 36], without an underlying continuous function. Discrete representations can be particularly efficient for parameterizing sparse scenes and object surfaces, though dense signals can require substantial memory or suffer artifacts [12]. The primary strengths of discrete representations lie in their interpretability and suitability for downstream processing, including integration into fast low-level pipelines for optimization and rendering. However, these methods are often sensitive to initialization and require heuristic optimization strategies because gradients do not flow well in a discrete representation.

**Hybrid approaches** seek to fill out the pareto frontier between the adaptability and memory efficiency of INRs and the interpretability and time efficiency of interpolated grids and discrete representations. These methods often combine learned grid-based features, which may be uniform [21] or compressed via a hash table [22] or tensor factorization [20, 23, 24, 26], with a lightweight neural feature decoder. Although hybrid methods can achieve performance gains over purely implicit or explicit representations, it is often unclear when and why specific combinations of neural and grid-based representations work well.

**Adaptive-bandwidth representations** dynamically adjust a signal's spatial–spectral bandwidth during optimization. Notable examples include [17], which assigns each layer a learnable, coordinate-dependent frequency cutoff, and [18], which progressively unmasks higher-frequency components through a spatially adaptive mask updated by a feedback loop during training. These approaches seek to represent mixed-frequency content more efficiently than fixed-bandwidth INRs.

We thoroughly evaluate the behavior of representative grid-based, INR-based, discrete, and hybrid representations to shed light on the tradeoffs made by each method and offer practical guidance on which representations are best suited to which applications.

## 5.2 Interpolation Methods Used in the Grid Model

The Grid model relies on interpolation to query continuous values from its discrete parameterization. In our experiments, we use bicubic interpolation for 2D signals and trilinear interpolation for 3D signals, to ensure a continuous representation while maintaining computational efficiency.

**Bicubic Interpolation (2D).** For 2D experiments, the interpolated value at a query point $\mathbf{q} = (x, y)$ is computed using the values at the 16 nearest grid points (a $4 \times 4$ region) surrounding $(x, y)$:

$$f(x, y) = \sum_{i=-1}^{2} \sum_{j=-1}^{2} w_i(x) \cdot w_j(y) \cdot g(x_i, y_j),$$

where $g(x_i, y_j)$ are the values at the neighboring grid points and $w_i(x)$, $w_j(y)$ are the bicubic weights derived from the cubic kernel function:

$$k(t) = \begin{cases} (a+2)|t|^3 - (a+3)|t|^2 + 1 & \text{if } |t| \leq 1, \\ a|t|^3 - 5a|t|^2 + 8a|t| - 4a & \text{if } 1 < |t| \leq 2, \\ 0 & \text{otherwise,} \end{cases}$$

where $a = -0.5$ is a common choice for visually smooth interpolation.

**Trilinear Interpolation (3D).** For 3D experiments, the interpolated value at a query point $\mathbf{q} = (x, y, z)$ is computed using the values at the eight nearest grid points:

$$f(x, y, z) = \sum_{i=0}^{1} \sum_{j=0}^{1} \sum_{k=0}^{1} w_i(x) \cdot w_j(y) \cdot w_k(z) \cdot g(x_i, y_j, z_k),$$

where $g(x_i, y_j, z_k)$ are the neighboring grid values and $w_i(x)$, $w_j(y)$, $w_k(z)$ are linear interpolation weights, defined as:

$$w_i(x) = 1 - |x - x_i|.$$

By using these interpolation techniques, the Grid model effectively reconstructs continuous signals in both 2D and 3D.

## 5.3 Synthetic Signals

Here we provide additional details on the synthetic signals summarized in Figure 1 and Table 1.

**Bandlimited Signals (2D, 3D).** We start with random uniform noise and apply a discrete Fourier transform; we then apply a circular low-pass filter in the Fourier domain, followed by an inverse DFT, to generate noise with different bandwidths. To align the naming convention with other signals, we divide the frequency range into nine exponentially spaced intervals but label the frequency cutoffs linearly from 0.1 (lowest bandwidth) to 0.9 (highest bandwidth).

**Spheres (2D, 3D).** We randomly place a small number of large filled circles (or spheres in 3D) to create lower "bandwidth" (0.1) signals. As the effective bandwidth moves toward 0.9, we increase the number of circles/spheres while decreasing their individual radii to create more fine-grained structures.

**Sierpinski (2D).** Sierpinski triangles with each fractal iteration mapped to a linear increase of 0.1 in qualitative bandwidth. Greater "bandwidth" yields fractal patterns with more fine-scale detail.

**Star Target (2D).** Triangular wedges arranged radially around the origin. Qualitative bandwidth increases radially from periphery to center. To quantify effective bandwidth we divided the image into 9 concentric rings, assigning bandwidth values from 0.1 (outermost ring) to 0.9 (center disk).

### 5.4 Real Signals

**Computed Tomography (CT).** For CT experiments, we train models on a real chest CT slice from the dataset in Clark et al. [28], which was also used in WIRE [16]. The training data was 100 projection measurements of the original $326\times435$ chest CT slice, forming a $100\times435$ sinogram equivalent to approximately 30% of the total pixel count in the original image. Since this inverse problem is inherently underdetermined, we apply TV regularization in our Grid model. The TV hyperparameter was tuned using the classic Shepp-Logan phantom image [37] as a reconstruction target.

**DIV2K Dataset.** We sample 10 high-resolution images (about $2 \times 10^5$ pixels each) from the DIV2K dataset [27] and evaluate all models on three tasks: image overfitting, $4\times$ super-resolution, and denoising Gaussian noise with standard deviation 0.1 or 0.05.

**3D Dragon Object.** We evaluate the models on a 3D object [29] with approximately $1 \times 10^6$ voxels in two versions, one where the object is solid and the other with only the object surface. We evaluate all models on overfitting and super-resolution tasks, for which models are evaluated on a higher resolution grid with double the total number of voxels as the training data.

### 5.5 Implementation Details

To ensure the parameter count remains consistent while preserving the characteristics of each model, we solve a quadratic equation to calculate the number of neurons $x$ to use in each hidden layer of an MLP, using the convention that the input and output layers are distinct from the hidden layers:

$$Lx^2 + (L + d_{in \text{ or } enc} + d_{out})x + d_{in} + d_{out} + \text{Params}_{enc} = P.$$

The quadratic term has a coefficient corresponding to the number of hidden layers $L$ in the MLP. The linear term includes contributions from the weights of the input and output layers ($d_{in \text{ or } enc}$ and $d_{out}$) of the MLP as well as the bias parameters of the hidden layers. The constant term includes the total model size, the bias parameters of the input and output layers, and any trainable parameters in a separate embedding, $\text{Params}_{enc}$, which is only used for Instant-NGP (to represent the parameters in the hash tables). By solving this quadratic equation, we can determine the appropriate number of neurons in each hidden layer for an INR constrained by a specific parameter count. For hybrid methods (Instant-NGP and GA-Planes), most of the parameters are allocated to the grid-based features rather than the MLP decoder, so we adjust model size primarily by varying the size of these feature grids. For Instant-NGP this is done by changing the size of the hash tables and keeping the width of the decoder MLP fixed; for GA-Planes we vary the feature dimension of the grids, which also controls the width of the decoder MLP. For Gaussian Splatting, we counted the number of parameters for one gaussian and divided the allocated number of parameters to get the total number of Gaussians. Since the structure of BACON is more complex, we manually swept the model architecture parameters to achieve the allocated model sizes. Details of these model parameters are in Table 3.

All models are trained using the Adam optimizer with $\beta = (0.9, 0.999)$ and learning rates tuned via grid search on the Star Target image at model size $1 \times 10^4$ (see Table 2). Initialization schemes follow prior work: Fourier Features use Gaussian-initialized embeddings [15] and SIREN uses small uniform weights to prevent sinusoidal explosion [14]. Mean-squared error loss is used for all experiments, with optional total-variation regularization applied to grid-based models to encourage smoothness on generalization tasks. All experiments are conducted on an NVIDIA RTX A6000 GPU with 48GB VRAM, with memory usage posing no limitations.

### 5.6 Signal Overfitting: Full Results

**Extended Results for Overfitting Tasks.** For each synthetic signal we visualize two experiments: either the bandlimit is fixed at 0.5 while varying the model parameter budget, or the model size is fixed at $1 \times 10^4$ while sweeping across different bandlimits. See Figure 7 and tables 5 to 7 (2D Spheres), Figure 8 and tables 8 to 10 (2D Bandlimited), Figure 9 and tables 11 to 13 (Sierpinski), Figure 10 and tables 14 to 16 (Star Target, with error map), Figure 11 and tables 17 and 18 (3D Spheres), and Figure 12 and table 19 (3D Bandlimited).

For real signals without an inherent notion of bandwidth, such as the DIV2K dataset and the 3D Dragon, only the model sizes vary. See Figure 13 (DIV2K), Figures 14 and 16 (3D Dragon occupancy),

| Model Size | Learning Rate | Grid | FFN |
|:---:|:---:|:---:|:---:|
| 1e4 | 5e−4 | 9.95 | 10.46 |
| | 1e−3 | 13.09 | 10.84 |
| | 5e−3 | 14.26 | 11.83 |
| | 1e−2 | 14.26 | 12.04 |
| | 5e−2 | 14.26 | 11.46 |
| | 1e−1 | 14.26 | 11.47 |
| 3e4 | 5e−5 | 10.39 | 18.43 |
| | 1e−4 | 6.28 | 13.53 |
| | 5e−4 | 10.39 | 18.43 |
| | 1e−3 | 14.67 | 18.88 |
| | 5e−3 | 16.55 | 19.13 |
| | 1e−2 | 16.55 | 19.01 |
| | 5e−2 | 16.55 | 19.48 |
| | 1e−1 | 16.55 | 6.36 |
| 1e5 | 1e−4 | 6.29 | 19.05 |
| | 5e−4 | 10.66 | 22.81 |
| | 1e−3 | 16.17 | 23.08 |
| | 5e−3 | 19.37 | 23.62 |
| | 1e−2 | 19.37 | 23.77 |
| | 5e−2 | 19.37 | 6.36 |
| | 1e−1 | 19.37 | 6.36 |
| 3e5 | 5e−4 | 10.83 | 25.99 |
| | 1e−3 | 17.25 | 25.74 |
| | 5e−3 | 22.25 | 28.13 |
| | 1e−2 | 22.25 | 26.04 |
| | 5e−2 | 22.25 | 6.36 |
| | 1e−1 | 22.25 | 6.36 |
| 1e6 | 1e−4 | 6.30 | 25.23 |
| | 5e−4 | 10.96 | 29.15 |
| | 1e−3 | 18.44 | 31.76 |
| | 5e−3 | 36.85 | 6.36 |
| | 1e−2 | 40.60 | 6.36 |
| | 5e−2 | 46.86 | 6.36 |
| | 1e−1 | 47.36 | 6.36 |
| 3e6 | 1e−4 | 6.29 | 31.34 |
| | 5e−4 | 10.96 | 38.81 |
| | 1e−3 | 18.61 | 35.96 |
| | 5e−3 | 122.78 | 6.36 |
| | 1e−2 | 133.35 | 6.36 |
| | 5e−2 | 152.26 | 6.36 |
| | 1e−1 | 149.76 | 6.36 |

Table 2: PSNR comparison between Grid and FFN across different learning rates and model sizes on the Star Target image. We note that the Grid is in some cases more stable with respect to varying learning rate. However, the model ranking is unaffected by learning rate tuning, so in all experiments we use the optimal learning rate for each model tuned at the smallest model size.

and Figures 15 and 17 (3D Dragon surface). All quantitative results for the real signals can be found in Tables 24 and 25.

| Model | Hyperparameter | Value |
|---|---|---|
| FFN | Learning Rate | 1e-3 |
| | Embedding Size | 2000 |
| | Hidden Layers | 2 |
| | $\sigma$ | 20 |
| SIREN | Learning Rate | 1e-4 |
| | Hidden Layers | 3 |
| | $\sigma$ | 90 |
| WIRE | Learning Rate | 5e-4 |
| | Hidden Layers | 2 |
| | $\sigma$ | 10 |
| | $\omega$ | 15 |
| Instant-NGP | Learning Rate | 1e-2 |
| | Hidden Layers | 1 (2-layer MLP) |
| | $L$ | 16 |
| | $F$ | 2 |
| | Number of Neurons | 64 |
| GA-Planes | Learning Rate | 1e-2 |
| | Line Resolution | size per axis of signal |
| | Plane Resolution | 20 |
| | Volume Resolution | 5 |
| | Hidden Layers | 1 (2-layer MLP) |
| GSplat | Learning Rate | 5e-2 |
| BACON | Learning Rate | 5e-2 |
| Grid | Learning Rate | 1e-1 |

Table 3: **Fixed hyperparameters used for each model configuration.** For 2D GA-Planes at model size 1e4, we used a line resolution of 550 and a plane resolution of 11, so that the feature dimension can be larger than 1.

| Model | Var | 1e4 | 3e4 | 1e5 | 3e5 | 1e6 | 3e6 |
|---|---|---|---|---|---|---|---|
| FFN | | 3 | 13 | 46 | 131 | 364 | 820 |
| SIREN | | 56 | 99 | 181 | 315 | 576 | 998 |
| WIRE | | 48 | 85 | 156 | 272 | 498 | 864 |
| Instant NGP | | 7 | 9 | 11 | 13 | 14 | 16 |
| GA Planes | 2D | 8 | 12 | 41 | 119 | 362 | 907 |
| | 3D | 6 | 18 | 59 | 167 | 476 | 1100 |
| GSplat | Gray | 833 | 2500 | 8333 | 25000 | 83333 | 250000 |
| | RGB | 714 | 2142 | 7142 | 21428 | 71428 | 214285 |
| BACON | | 48 | 85 | 156 | 272 | 498 | 684 |
| Grid | 2D | 100 | 173 | 316 | 547 | 1000 | 1732 |
| | 3D | 21 | 31 | 46 | 66 | 99 | 144 |

Table 4: **Model size-specific parameters.** Values are computed using Equation 5.5. FFN, SIREN, and WIRE denote the number of neurons $x$ per hidden layer. Instant-NGP values refer to $\log_2$ hashmap sizes. GA-Planes values represent feature dimensions and final MLP widths (which must match). For Gsplat, each Gaussian requires two additional parameters for RGB scenes compared to grayscale scenes, necessitating slightly different numbers of Gaussians to maintain equal total model size. For the Grid, we report the resolution (side length). All other hyperparameters are fixed across model sizes.

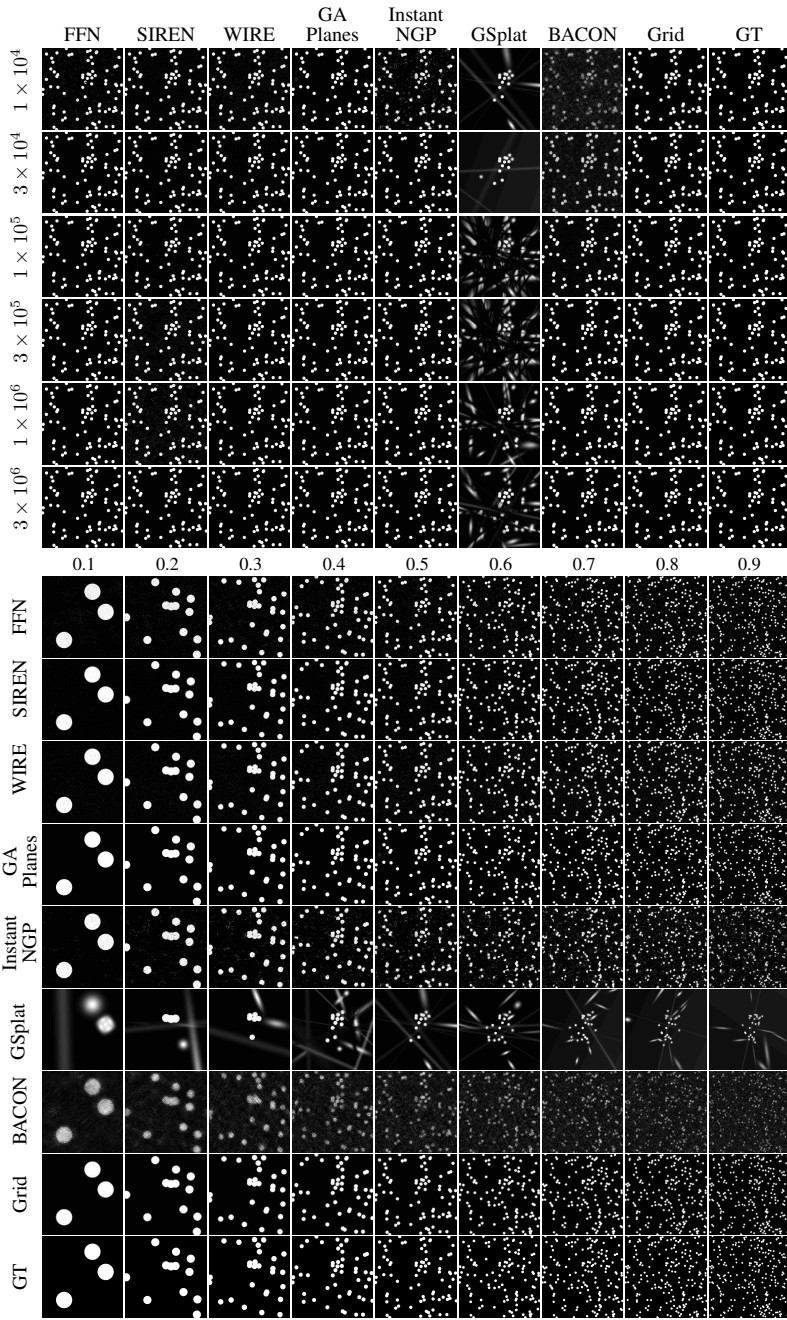

Figure 7: **2D Spheres overfitting with bandwidth = 0.5 (top) and model size = $1 \times 10^4$ (bottom).** For the 2D Spheres signal, most models effectively capture the structure even in highly compressed settings. However, GSplat struggles to fit some regions of the image and BACON produces a blurry representation at small model sizes. Detailed quantitative results are in Tables 5 to 7.

Table 5: Comparison of model performance across bandwidths on 2D Spheres signal (PSNR).

| Model | Size | 0.1 | 0.2 | 0.3 | 0.4 | 0.5 | 0.6 | 0.7 | 0.8 | 0.9 |
|---|---|---|---|---|---|---|---|---|---|---|
| FFN | 1e+04 | 29.46±6.20 | 27.28±4.98 | 25.05±4.13 | 23.56±3.50 | 22.53±2.86 | 21.40±2.58 | 20.32±2.15 | 19.28±1.85 | 18.43±1.67 |
| | 3e+04 | 38.57±1.28 | 35.86±0.89 | 33.98±0.71 | 32.33±0.65 | 31.09±0.60 | 30.15±0.53 | 29.41±0.50 | 28.62±0.47 | 27.94±0.36 |
| | 1e+05 | 43.11±1.81 | 42.47±0.94 | 41.03±0.81 | 39.65±0.61 | 38.01±0.44 | 37.37±0.51 | 36.73±0.38 | 35.69±0.31 | 34.93±0.22 |
| | 3e+05 | 43.54±0.86 | 44.78±1.22 | 47.66±1.29 | 46.98±0.82 | 47.25±0.49 | 45.51±0.51 | 44.58±0.69 | 43.52±1.21 | 43.56±0.67 |
| | 1e+06 | 54.69±1.06 | 53.16±1.30 | 53.80±0.81 | 53.87±1.57 | 51.46±2.65 | 50.98±1.42 | 50.27±1.54 | 50.23±0.90 | 48.89±1.17 |
| | 3e+06 | 42.23±0.69 | 43.02±4.80 | 41.66±4.22 | 40.81±5.27 | 39.93±4.34 | 38.97±4.23 | 38.42±3.34 | 38.24±4.63 | 37.30±2.64 |
| SIREN | 1e+04 | 28.82±0.59 | 26.07±0.26 | 24.23±0.14 | 22.99±0.19 | 21.70±0.74 | 21.16±0.19 | 20.36±0.12 | 19.82±0.08 | 18.95±0.61 |
| | 3e+04 | 30.02±1.07 | 27.76±0.24 | 25.68±0.96 | 24.06±1.66 | 23.15±1.52 | 22.61±0.93 | 22.13±0.46 | 21.50±0.39 | 21.00±0.28 |
| | 1e+05 | 24.63±4.13 | 24.58±4.42 | 24.16±3.44 | 21.36±3.48 | 19.91±3.22 | 20.63±3.58 | 21.62±2.20 | 22.18±1.87 | 19.15±2.67 |
| | 3e+05 | 20.39±2.60 | 20.68±2.40 | 21.43±1.90 | 18.79±3.44 | 18.17±2.46 | 18.53±1.52 | 17.94±2.51 | 16.82±2.35 | 19.78±2.99 |
| | 1e+06 | 23.73±2.56 | 18.77±3.94 | 23.93±2.45 | 23.68±4.78 | 19.07±3.25 | 24.84±5.85 | 16.53±4.84 | 19.34±4.06 | 17.56±2.06 |
| | 3e+06 | 42.06±10.00 | 21.73±15.54 | 36.46±11.25 | 27.08±12.36 | 29.08±16.24 | 35.77±16.08 | 38.34±17.28 | 37.19±6.57 | 37.53±16.24 |
| WIRE | 1e+04 | 27.65±0.90 | 25.31±0.48 | 23.57±0.25 | 22.31±0.08 | 21.15±0.10 | 20.35±0.06 | 19.61±0.10 | 19.02±0.12 | 18.47±0.08 |
| | 3e+04 | 30.77±0.57 | 28.11±0.36 | 26.35±0.09 | 25.01±0.10 | 23.95±0.04 | 23.16±0.05 | 22.47±0.04 | 21.86±0.02 | 21.29±0.03 |
| | 1e+05 | 34.29±0.64 | 31.45±0.37 | 29.54±0.14 | 28.08±0.11 | 26.88±0.08 | 26.05±0.07 | 25.29±0.05 | 24.66±0.07 | 24.05±0.05 |
| | 3e+05 | 37.10±0.63 | 33.95±0.41 | 32.17±0.10 | 30.66±0.17 | 29.34±0.11 | 28.39±0.14 | 27.62±0.14 | 26.91±0.09 | 26.21±0.11 |
| | 1e+06 | 38.19±1.08 | 34.44±0.49 | 32.22±0.19 | 30.68±0.08 | 29.73±0.09 | 29.05±0.15 | 28.45±0.25 | 27.88±0.16 | 27.37±0.13 |
| | 3e+06 | 38.44±1.07 | 34.62±0.90 | 31.91±0.31 | 30.26±0.07 | 29.06±0.11 | 28.43±0.19 | 27.92±0.15 | 27.34±0.27 | 27.02±0.11 |
| GA-Planes | 1e+04 | 35.36±13.49 | 37.34±15.09 | 33.54±12.70 | 29.79±10.52 | 25.20±7.64 | 21.61±5.54 | 21.54±5.49 | 19.52±4.41 | 17.73±3.51 |
| | 3e+04 | 51.20±10.15 | 51.69±9.97 | 48.60±9.32 | 43.25±6.96 | 41.70±1.72 | 34.82±4.35 | 31.62±3.39 | 28.52±2.71 | 27.99±3.06 |
| | 1e+05 | 65.15±1.04 | 64.86±2.57 | 60.45±4.13 | 61.20±6.53 | 52.66±3.90 | 50.36±1.12 | 47.04±2.56 | 43.64±2.77 | 43.27±2.61 |
| | 3e+05 | 66.10±1.09 | 63.75±2.07 | 65.24±2.31 | 65.00±5.63 | 61.34±4.33 | 53.77±3.31 | 58.80±4.34 | 52.37±1.17 | 52.80±2.07 |
| | 1e+06 | 61.34±2.59 | 59.44±2.77 | 59.08±3.76 | 57.97±4.29 | 57.63±4.34 | 58.24±2.21 | 56.20±5.31 | 54.37±2.62 | 53.45±0.58 |
| | 3e+06 | 61.97±1.67 | 62.27±4.96 | 62.96±2.81 | 57.37±1.80 | 54.50±3.51 | 50.82±3.60 | 50.34±1.44 | 48.79±3.88 | 49.88±3.80 |
| Instant-NGP | 1e+04 | 26.26±2.74 | 22.72±0.64 | 20.74±0.39 | 19.29±0.75 | 17.11±0.34 | 16.58±0.18 | 15.78±0.26 | 14.98±0.31 | 14.51±0.20 |
| | 3e+04 | 59.21±7.36 | 56.57±3.28 | 44.75±3.43 | 38.82±1.55 | 32.42±1.01 | 29.70±1.06 | 26.73±0.75 | 25.52±0.98 | 23.67±0.61 |
| | 1e+05 | 64.17±6.69 | 66.96±1.43 | 66.64±3.74 | 68.09±1.86 | 65.51±4.40 | 60.99±4.42 | 62.46±6.42 | 52.05±2.12 | 48.00±3.77 |
| | 3e+05 | 63.10±2.63 | 66.07±2.79 | 68.88±1.69 | 69.27±2.19 | 68.29±2.03 | 66.51±3.46 | 68.93±1.94 | 68.91±1.76 | 68.15±2.24 |
| | 1e+06 | 67.44±2.99 | 69.37±1.44 | 68.14±4.18 | 70.30±1.99 | 69.80±2.01 | 70.85±3.22 | 69.80±2.37 | 71.44±3.06 | 69.63±2.53 |
| | 3e+06 | 73.07±2.92 | 75.79±4.53 | 74.81±3.39 | 72.52±2.83 | 73.19±3.85 | 74.25±1.68 | 77.01±4.82 | 77.04±5.12 | 71.42±2.97 |
| GSplat | 1e+04 | 16.29±3.61 | 12.61±0.45 | 12.00±0.15 | 12.34±0.47 | 11.77±0.26 | 11.58±0.09 | 11.36±0.11 | 11.27±0.05 | 11.18±0.06 |
| | 3e+04 | 15.90±3.96 | 13.41±0.83 | 12.41±0.57 | 12.27±0.41 | 11.90±0.45 | 11.93±0.24 | 11.56±0.09 | 11.38±0.05 | 11.34±0.13 |
| | 1e+05 | 18.00±3.26 | 15.10±1.73 | 12.81±0.66 | 13.20±0.29 | 12.89±0.84 | 12.53±0.56 | 11.96±0.18 | 11.79±0.14 | 11.62±0.06 |
| | 3e+05 | 20.01±3.40 | 16.17±1.71 | 14.63±1.47 | 13.92±0.55 | 13.03±0.73 | 13.47±0.77 | 12.52±0.39 | 12.23±0.32 | 11.89±0.19 |
| | 1e+06 | 21.35±3.69 | 16.99±2.03 | 15.82±1.27 | 15.11±0.82 | 14.14±0.84 | 13.77±0.71 | 13.43±0.18 | 13.06±0.57 | 12.91±0.56 |
| | 3e+06 | 21.88±3.59 | 17.80±1.61 | 16.62±0.92 | 15.89±0.49 | 14.38±0.54 | 14.07±0.53 | 14.13±0.18 | 13.43±0.18 | 13.25±0.56 |
| BACON | 1e+04 | 15.80±0.74 | 14.40±0.52 | 13.69±0.31 | 13.27±0.24 | 12.91±0.20 | 12.70±0.21 | 12.52±0.14 | 12.42±0.18 | 12.26±0.16 |
| | 3e+04 | 21.15±0.62 | 19.16±0.16 | 17.89±0.06 | 17.13±0.19 | 16.47±0.18 | 16.15±0.17 | 15.87±0.17 | 15.61±0.21 | 15.33±0.25 |
| | 1e+05 | 30.59±1.24 | 28.19±0.63 | 26.40±0.40 | 25.24±0.20 | 24.20±0.21 | 23.57±0.16 | 23.08±0.25 | 22.62±0.23 | 22.15±0.18 |
| | 3e+05 | 37.20±0.73 | 34.74±0.49 | 33.14±0.24 | 32.02±0.27 | 30.87±0.31 | 30.25±0.30 | 29.44±0.25 | 29.02±0.11 | 28.35±0.19 |
| | 1e+06 | 39.16±1.24 | 36.68±1.07 | 35.36±1.12 | 34.16±1.22 | 33.26±1.10 | 32.51±1.00 | 32.06±1.14 | 31.52±1.08 | 31.33±1.20 |
| | 3e+06 | 36.97±5.19 | 35.40±4.33 | 33.83±4.26 | 32.66±3.94 | 32.00±3.78 | 31.64±3.69 | 30.88±3.60 | 30.22±3.46 | 30.43±3.68 |
| Grid | 1e+04 | 28.69±0.70 | 25.63±0.39 | 23.67±0.09 | 22.30±0.10 | 21.26±0.04 | 20.43±0.08 | 19.67±0.05 | 19.15±0.04 | 18.76±0.04 |
| | 3e+04 | 31.06±0.73 | 28.02±0.35 | 26.07±0.07 | 24.74±0.09 | 23.65±0.05 | 22.89±0.06 | 22.18±0.05 | 21.62±0.04 | 21.09±0.04 |
| | 1e+05 | 33.82±0.70 | 30.78±0.39 | 28.84±0.08 | 27.49±0.09 | 26.40±0.05 | 25.63±0.06 | 24.95±0.04 | 24.36±0.04 | 23.82±0.05 |
| | 3e+05 | 36.99±0.72 | 33.91±0.38 | 31.93±0.08 | 30.62±0.09 | 29.51±0.06 | 28.76±0.07 | 28.07±0.05 | 27.49±0.04 | 26.94±0.04 |
| | 1e+06 | 72.86±25.66 | 59.01±1.36 | 57.58±0.79 | 55.06±0.86 | 54.53±0.74 | 53.73±0.87 | 52.96±0.91 | 52.23±1.14 | 51.67±0.90 |
| | 3e+06 | 144.20±0.50 | 144.13±0.13 | 144.08±0.05 | 144.06±0.09 | 144.01±0.06 | 144.06±0.06 | 144.07±0.06 | 144.12±0.06 | 144.14±0.03 |

Table 6: Comparison of model performance across bandwidths on 2D Spheres signal (SSIM).

| Model | Size | 0.1 | 0.2 | 0.3 | 0.4 | 0.5 | 0.6 | 0.7 | 0.8 | 0.9 |
|---|---|---|---|---|---|---|---|---|---|---|
| FFN | 1e+04 | 0.829±0.17 | 0.815±0.16 | 0.775±0.16 | 0.763±0.18 | 0.752±0.17 | 0.735±0.16 | 0.713±0.16 | 0.681±0.15 | 0.653±0.14 |
| | 3e+04 | 0.967±0.03 | 0.956±0.03 | 0.948±0.03 | 0.941±0.03 | 0.938±0.03 | 0.935±0.03 | 0.933±0.03 | 0.931±0.03 | 0.921±0.02 |
| | 1e+05 | 0.964±0.03 | 0.982±0.01 | 0.984±0.01 | 0.983±0.00 | 0.980±0.00 | 0.979±0.00 | 0.979±0.00 | 0.972±0.01 | 0.976±0.00 |
| | 3e+05 | 0.955±0.01 | 0.954±0.01 | 0.986±0.01 | 0.991±0.01 | 0.995±0.00 | 0.991±0.01 | 0.993±0.01 | 0.978±0.03 | 0.990±0.01 |
| | 1e+06 | 0.997±0.00 | 0.999±0.00 | 0.999±0.00 | 0.999±0.00 | 0.999±0.00 | 0.998±0.00 | 0.999±0.00 | 0.999±0.00 | 0.999±0.00 |
| | 3e+06 | 0.946±0.03 | 0.934±0.06 | 0.907±0.05 | 0.905±0.05 | 0.895±0.08 | 0.866±0.07 | 0.902±0.07 | 0.852±0.10 | 0.893±0.08 |
| SIREN | 1e+04 | 0.715±0.06 | 0.684±0.07 | 0.603±0.04 | 0.587±0.08 | 0.579±0.17 | 0.544±0.09 | 0.471±0.01 | 0.532±0.15 | 0.460±0.11 |
| | 3e+04 | 0.782±0.03 | 0.681±0.09 | 0.608±0.11 | 0.535±0.18 | 0.538±0.10 | 0.560±0.05 | 0.537±0.01 | 0.491±0.02 | 0.595±0.14 |
| | 1e+05 | 0.473±0.20 | 0.543±0.23 | 0.584±0.29 | 0.462±0.18 | 0.393±0.15 | 0.548±0.27 | 0.425±0.13 | 0.527±0.10 | 0.415±0.16 |
| | 3e+05 | 0.363±0.18 | 0.331±0.05 | 0.447±0.14 | 0.311±0.16 | 0.232±0.04 | 0.315±0.09 | 0.344±0.18 | 0.279±0.08 | 0.400±0.10 |
| | 1e+06 | 0.304±0.18 | 0.144±0.12 | 0.350±0.10 | 0.447±0.25 | 0.242±0.11 | 0.473±0.25 | 0.190±0.15 | 0.370±0.17 | 0.176±0.05 |
| | 3e+06 | 0.823±0.27 | 0.328±0.32 | 0.641±0.34 | 0.408±0.35 | 0.552±0.32 | 0.627±0.42 | 0.771±0.39 | 0.581±0.23 | 0.737±0.38 |
| WIRE | 1e+04 | 0.525±0.04 | 0.481±0.03 | 0.432±0.02 | 0.400±0.01 | 0.366±0.01 | 0.348±0.01 | 0.332±0.01 | 0.319±0.01 | 0.309±0.01 |
| | 3e+04 | 0.595±0.03 | 0.536±0.02 | 0.489±0.02 | 0.444±0.01 | 0.413±0.01 | 0.386±0.01 | 0.367±0.01 | 0.350±0.01 | 0.336±0.01 |
| | 1e+05 | 0.802±0.01 | 0.747±0.02 | 0.694±0.01 | 0.641±0.01 | 0.595±0.00 | 0.566±0.00 | 0.536±0.01 | 0.515±0.01 | 0.492±0.01 |
| | 3e+05 | 0.901±0.01 | 0.837±0.02 | 0.815±0.00 | 0.771±0.01 | 0.726±0.01 | 0.702±0.01 | 0.677±0.01 | 0.661±0.01 | 0.639±0.01 |
| | 1e+06 | 0.933±0.01 | 0.892±0.01 | 0.840±0.01 | 0.798±0.01 | 0.757±0.01 | 0.720±0.01 | 0.685±0.01 | 0.650±0.01 | 0.629±0.01 |
| | 3e+06 | 0.959±0.01 | 0.928±0.01 | 0.886±0.01 | 0.838±0.01 | 0.798±0.00 | 0.766±0.01 | 0.732±0.00 | 0.703±0.01 | 0.676±0.01 |
| GA-Planes | 1e+04 | 0.801±0.39 | 0.799±0.39 | 0.787±0.38 | 0.791±0.38 | 0.761±0.37 | 0.742±0.36 | 0.714±0.35 | 0.658±0.32 | 0.444±0.22 |
| | 3e+04 | 0.996±0.01 | 0.996±0.01 | 0.995±0.01 | 0.994±0.01 | 0.999±0.00 | 0.984±0.02 | 0.973±0.02 | 0.919±0.05 | 0.941±0.04 |
| | 1e+05 | 1.000±0.00 | 1.000±0.00 | 1.000±0.00 | 1.000±0.00 | 0.999±0.00 | 0.992±0.01 | 0.999±0.00 | 0.998±0.00 | 0.996±0.00 |
| | 3e+05 | 1.000±0.00 | 1.000±0.00 | 1.000±0.00 | 1.000±0.00 | 1.000±0.00 | 0.999±0.00 | 0.999±0.00 | 0.999±0.00 | 0.999±0.00 |
| | 1e+06 | 0.998±0.00 | 0.998±0.00 | 0.998±0.00 | 0.999±0.00 | 0.999±0.00 | 0.999±0.00 | 0.998±0.00 | 0.997±0.00 | 0.996±0.00 |
| | 3e+06 | 0.999±0.00 | 0.999±0.00 | 0.999±0.00 | 0.998±0.00 | 0.997±0.00 | 0.997±0.00 | 0.998±0.00 | 0.987±0.01 | 0.992±0.00 |
| Instant-NGP | 1e+04 | 0.795±0.08 | 0.679±0.11 | 0.654±0.06 | 0.604±0.08 | 0.388±0.08 | 0.410±0.08 | 0.310±0.04 | 0.296±0.08 | 0.284±0.05 |
| | 3e+04 | 0.996±0.01 | 0.999±0.00 | 0.991±0.01 | 0.922±0.07 | 0.951±0.02 | 0.859±0.10 | 0.882±0.03 | 0.866±0.03 | 0.824±0.06 |
| | 1e+05 | 0.998±0.00 | 1.000±0.00 | 1.000±0.00 | 1.000±0.00 | 1.000±0.00 | 1.000±0.00 | 1.000±0.00 | 0.999±0.00 | 0.997±0.00 |
| | 3e+05 | 0.998±0.00 | 0.999±0.00 | 1.000±0.00 | 1.000±0.00 | 1.000±0.00 | 1.000±0.00 | 1.000±0.00 | 1.000±0.00 | 1.000±0.00 |
| | 1e+06 | 0.999±0.00 | 1.000±0.00 | 1.000±0.00 | 1.000±0.00 | 1.000±0.00 | 1.000±0.00 | 1.000±0.00 | 1.000±0.00 | 1.000±0.00 |
| | 3e+06 | 1.000±0.00 | 1.000±0.00 | 1.000±0.00 | 1.000±0.00 | 1.000±0.00 | 1.000±0.00 | 1.000±0.00 | 1.000±0.00 | 1.000±0.00 |
| GSplat | 1e+04 | 0.743±0.17 | 0.792±0.09 | 0.655±0.15 | 0.474±0.14 | 0.262±0.25 | 0.308±0.24 | 0.035±0.00 | 0.031±0.00 | 0.032±0.01 |
| | 3e+04 | 0.815±0.14 | 0.796±0.06 | 0.740±0.09 | 0.556±0.15 | 0.322±0.17 | 0.311±0.15 | 0.107±0.13 | 0.036±0.00 | 0.059±0.04 |
| | 1e+05 | 0.859±0.06 | 0.609±0.27 | 0.605±0.04 | 0.551±0.09 | 0.338±0.09 | 0.385±0.07 | 0.165±0.11 | 0.164±0.12 | 0.102±0.07 |
| | 3e+05 | 0.866±0.06 | 0.708±0.04 | 0.570±0.05 | 0.515±0.04 | 0.444±0.07 | 0.393±0.02 | 0.322±0.06 | 0.256±0.08 | 0.208±0.13 |
| | 1e+06 | 0.831±0.07 | 0.667±0.09 | 0.586±0.05 | 0.519±0.03 | 0.460±0.05 | 0.444±0.04 | 0.354±0.01 | 0.355±0.06 | 0.297±0.04 |
| | 3e+06 | 0.836±0.06 | 0.659±0.06 | 0.615±0.08 | 0.553±0.03 | 0.446±0.04 | 0.441±0.07 | 0.430±0.02 | 0.360±0.03 | 0.330±0.02 |
| BACON | 1e+04 | 0.028±0.01 | 0.023±0.01 | 0.023±0.01 | 0.024±0.00 | 0.025±0.00 | 0.027±0.00 | 0.028±0.00 | 0.030±0.00 | 0.031±0.00 |
| | 3e+04 | 0.099±0.02 | 0.077±0.01 | 0.065±0.00 | 0.063±0.00 | 0.063±0.00 | 0.066±0.00 | 0.071±0.00 | 0.075±0.00 | 0.078±0.01 |
| | 1e+05 | 0.431±0.06 | 0.317±0.04 | 0.254±0.02 | 0.228±0.01 | 0.209±0.01 | 0.200±0.01 | 0.195±0.02 | 0.198±0.01 | 0.193±0.02 |
| | 3e+05 | 0.901±0.02 | 0.846±0.07 | 0.831±0.03 | 0.786±0.02 | 0.736±0.03 | 0.720±0.04 | 0.672±0.07 | 0.641±0.06 | 0.640±0.03 |
| | 1e+06 | 0.946±0.01 | 0.910±0.02 | 0.897±0.02 | 0.869±0.03 | 0.843±0.02 | 0.825±0.02 | 0.812±0.03 | 0.789±0.04 | 0.785±0.05 |
| | 3e+06 | 0.704±0.31 | 0.687±0.28 | 0.659±0.29 | 0.635±0.28 | 0.623±0.26 | 0.610±0.24 | 0.608±0.24 | 0.583±0.22 | 0.591±0.21 |
| Grid | 1e+04 | 0.960±0.01 | 0.921±0.01 | 0.878±0.00 | 0.836±0.00 | 0.796±0.01 | 0.765±0.00 | 0.734±0.00 | 0.703±0.00 | 0.679±0.00 |
| | 3e+04 | 0.974±0.00 | 0.949±0.00 | 0.921±0.00 | 0.893±0.00 | 0.864±0.00 | 0.841±0.00 | 0.820±0.00 | 0.796±0.00 | 0.775±0.00 |
| | 1e+05 | 0.987±0.00 | 0.975±0.00 | 0.961±0.00 | 0.947±0.00 | 0.933±0.00 | 0.921±0.00 | 0.908±0.00 | 0.895±0.00 | 0.883±0.00 |
| | 3e+05 | 0.994±0.00 | 0.988±0.00 | 0.982±0.00 | 0.975±0.00 | 0.969±0.00 | 0.963±0.00 | 0.957±0.00 | 0.951±0.00 | 0.944±0.00 |
| | 1e+06 | 0.999±0.00 | 0.998±0.00 | 0.998±0.00 | 0.996±0.00 | 0.996±0.00 | 0.995±0.00 | 0.994±0.00 | 0.993±0.00 | 0.992±0.00 |
| | 3e+06 | 1.000±0.00 | 1.000±0.00 | 1.000±0.00 | 1.000±0.00 | 1.000±0.00 | 1.000±0.00 | 1.000±0.00 | 1.000±0.00 | 1.000±0.00 |

Table 7: Comparison of model performance across bandwidths on 2D Spheres signal (LPIPS).

| Model | Size | 0.1 | 0.2 | 0.3 | 0.4 | 0.5 | 0.6 | 0.7 | 0.8 | 0.9 |
|---|---|---|---|---|---|---|---|---|---|---|
| FFN | 1e+04 | 0.315±0.10 | 0.325±0.09 | 0.348±0.08 | 0.354±0.08 | 0.355±0.08 | 0.365±0.08 | 0.377±0.07 | 0.387±0.07 | 0.397±0.07 |
| | 3e+04 | 0.225±0.08 | 0.244±0.07 | 0.258±0.05 | 0.255±0.05 | 0.245±0.04 | 0.235±0.04 | 0.225±0.03 | 0.213±0.03 | 0.206±0.02 |
| | 1e+05 | 0.181±0.08 | 0.134±0.06 | 0.117±0.05 | 0.130±0.02 | 0.142±0.02 | 0.143±0.02 | 0.132±0.02 | 0.130±0.01 | 0.116±0.02 |
| | 3e+05 | 0.237±0.00 | 0.208±0.01 | 0.102±0.05 | 0.071±0.04 | 0.048±0.02 | 0.052±0.01 | 0.049±0.01 | 0.047±0.01 | 0.037±0.00 |
| | 1e+06 | 0.056±0.05 | 0.020±0.00 | 0.012±0.00 | 0.011±0.00 | 0.015±0.00 | 0.012±0.00 | 0.012±0.00 | 0.011±0.00 | 0.009±0.00 |
| | 3e+06 | 0.218±0.09 | 0.200±0.15 | 0.260±0.12 | 0.250±0.12 | 0.231±0.12 | 0.251±0.12 | 0.209±0.10 | 0.219±0.11 | 0.186±0.10 |
| SIREN | 1e+04 | 0.427±0.03 | 0.433±0.03 | 0.454±0.01 | 0.458±0.02 | 0.445±0.08 | 0.479±0.02 | 0.501±0.01 | 0.471±0.07 | 0.513±0.03 |
| | 3e+04 | 0.389±0.02 | 0.410±0.03 | 0.427±0.03 | 0.444±0.04 | 0.448±0.03 | 0.445±0.01 | 0.450±0.00 | 0.458±0.01 | 0.419±0.05 |
| | 1e+05 | 0.479±0.08 | 0.430±0.08 | 0.425±0.09 | 0.471±0.06 | 0.494±0.06 | 0.425±0.13 | 0.471±0.05 | 0.442±0.04 | 0.499±0.06 |
| | 3e+05 | 0.529±0.05 | 0.525±0.02 | 0.489±0.04 | 0.534±0.06 | 0.555±0.03 | 0.533±0.03 | 0.528±0.06 | 0.552±0.04 | 0.495±0.05 |
| | 1e+06 | 0.540±0.06 | 0.589±0.04 | 0.516±0.03 | 0.480±0.08 | 0.550±0.05 | 0.476±0.08 | 0.602±0.10 | 0.529±0.07 | 0.579±0.04 |
| | 3e+06 | 0.336±0.10 | 0.515±0.14 | 0.352±0.15 | 0.486±0.19 | 0.415±0.26 | 0.352±0.25 | 0.302±0.28 | 0.280±0.09 | 0.277±0.28 |
| WIRE | 1e+04 | 0.484±0.01 | 0.488±0.00 | 0.499±0.01 | 0.507±0.00 | 0.525±0.00 | 0.537±0.00 | 0.550±0.00 | 0.561±0.00 | 0.569±0.00 |
| | 3e+04 | 0.434±0.01 | 0.444±0.00 | 0.455±0.00 | 0.463±0.00 | 0.470±0.00 | 0.478±0.00 | 0.485±0.00 | 0.490±0.00 | 0.493±0.01 |
| | 1e+05 | 0.342±0.01 | 0.361±0.01 | 0.378±0.00 | 0.394±0.00 | 0.407±0.00 | 0.414±0.00 | 0.422±0.00 | 0.423±0.00 | 0.420±0.00 |
| | 3e+05 | 0.325±0.01 | 0.368±0.01 | 0.359±0.00 | 0.373±0.01 | 0.391±0.00 | 0.393±0.00 | 0.396±0.00 | 0.391±0.01 | 0.386±0.01 |
| | 1e+06 | 0.301±0.03 | 0.337±0.01 | 0.354±0.01 | 0.354±0.00 | 0.356±0.00 | 0.360±0.00 | 0.366±0.00 | 0.367±0.00 | 0.366±0.00 |
| | 3e+06 | 0.235±0.02 | 0.280±0.01 | 0.317±0.00 | 0.327±0.00 | 0.332±0.00 | 0.337±0.00 | 0.343±0.00 | 0.343±0.00 | 0.343±0.00 |
| GA-Planes | 1e+04 | 0.084±0.10 | 0.092±0.11 | 0.109±0.12 | 0.131±0.12 | 0.172±0.13 | 0.203±0.12 | 0.211±0.13 | 0.274±0.11 | 0.329±0.12 |
| | 3e+04 | 0.018±0.02 | 0.019±0.03 | 0.019±0.03 | 0.022±0.04 | 0.004±0.00 | 0.039±0.04 | 0.061±0.04 | 0.128±0.05 | 0.114±0.07 |
| | 1e+05 | 0.002±0.00 | 0.001±0.00 | 0.001±0.00 | 0.002±0.00 | 0.004±0.00 | 0.008±0.01 | 0.008±0.01 | 0.007±0.01 | 0.009±0.01 |
| | 3e+05 | 0.003±0.00 | 0.004±0.00 | 0.002±0.00 | 0.001±0.00 | 0.002±0.00 | 0.002±0.00 | 0.002±0.00 | 0.002±0.00 | 0.003±0.00 |
| | 1e+06 | 0.020±0.01 | 0.017±0.01 | 0.012±0.01 | 0.007±0.00 | 0.006±0.00 | 0.006±0.00 | 0.008±0.01 | 0.009±0.01 | 0.010±0.01 |
| | 3e+06 | 0.010±0.01 | 0.015±0.02 | 0.004±0.00 | 0.008±0.00 | 0.011±0.01 | 0.024±0.02 | 0.028±0.03 | 0.046±0.06 | 0.032±0.02 |
| Instant-NGP | 1e+04 | 0.367±0.06 | 0.416±0.03 | 0.438±0.03 | 0.466±0.03 | 0.547±0.02 | 0.541±0.02 | 0.568±0.02 | 0.586±0.02 | 0.596±0.01 |
| | 3e+04 | 0.038±0.06 | 0.019±0.01 | 0.071±0.03 | 0.113±0.02 | 0.155±0.02 | 0.194±0.04 | 0.235±0.04 | 0.264±0.04 | 0.294±0.06 |
| | 1e+05 | 0.020±0.03 | 0.003±0.00 | 0.003±0.00 | 0.002±0.00 | 0.002±0.00 | 0.002±0.00 | 0.002±0.00 | 0.004±0.00 | 0.005±0.00 |
| | 3e+05 | 0.020±0.01 | 0.005±0.00 | 0.003±0.00 | 0.002±0.00 | 0.002±0.00 | 0.001±0.00 | 0.001±0.00 | 0.001±0.00 | 0.001±0.00 |
| | 1e+06 | 0.008±0.01 | 0.003±0.00 | 0.002±0.00 | 0.002±0.00 | 0.002±0.00 | 0.002±0.00 | 0.002±0.00 | 0.001±0.00 | 0.001±0.00 |
| | 3e+06 | 0.001±0.00 | 0.001±0.00 | 0.001±0.00 | 0.001±0.00 | 0.001±0.00 | 0.000±0.00 | 0.001±0.00 | 0.000±0.00 | 0.000±0.00 |
| GSplat | 1e+04 | 0.146±0.06 | 0.163±0.04 | 0.232±0.03 | 0.323±0.03 | 0.362±0.01 | 0.391±0.02 | 0.430±0.00 | 0.455±0.01 | 0.483±0.00 |
| | 3e+04 | 0.122±0.06 | 0.168±0.03 | 0.212±0.03 | 0.311±0.05 | 0.374±0.02 | 0.406±0.02 | 0.429±0.01 | 0.449±0.01 | 0.473±0.01 |
| | 1e+05 | 0.107±0.03 | 0.212±0.05 | 0.279±0.02 | 0.314±0.02 | 0.403±0.02 | 0.441±0.04 | 0.440±0.02 | 0.488±0.00 | 0.486±0.03 |
| | 3e+05 | 0.110±0.03 | 0.205±0.02 | 0.294±0.01 | 0.360±0.01 | 0.394±0.02 | 0.459±0.02 | 0.479±0.02 | 0.491±0.01 | 0.488±0.02 |
| | 1e+06 | 0.120±0.01 | 0.224±0.03 | 0.302±0.02 | 0.365±0.01 | 0.420±0.02 | 0.441±0.01 | 0.488±0.01 | 0.501±0.01 | 0.528±0.01 |
| | 3e+06 | 0.116±0.01 | 0.220±0.02 | 0.285±0.03 | 0.353±0.02 | 0.426±0.02 | 0.450±0.02 | 0.472±0.01 | 0.494±0.01 | 0.519±0.01 |
| BACON | 1e+04 | 0.626±0.03 | 0.634±0.01 | 0.651±0.01 | 0.658±0.01 | 0.673±0.01 | 0.684±0.01 | 0.693±0.01 | 0.700±0.00 | 0.709±0.00 |
| | 3e+04 | 0.580±0.01 | 0.598±0.00 | 0.611±0.00 | 0.624±0.01 | 0.636±0.01 | 0.648±0.01 | 0.653±0.01 | 0.651±0.00 | 0.652±0.00 |
| | 1e+05 | 0.451±0.01 | 0.468±0.00 | 0.478±0.00 | 0.486±0.00 | 0.493±0.01 | 0.496±0.01 | 0.495±0.01 | 0.491±0.00 | 0.489±0.00 |
| | 3e+05 | 0.270±0.01 | 0.312±0.01 | 0.321±0.01 | 0.332±0.01 | 0.346±0.01 | 0.354±0.01 | 0.362±0.01 | 0.355±0.01 | 0.350±0.01 |
| | 1e+06 | 0.277±0.02 | 0.294±0.01 | 0.275±0.02 | 0.292±0.02 | 0.315±0.02 | 0.324±0.02 | 0.327±0.02 | 0.323±0.02 | 0.319±0.02 |
| | 3e+06 | 0.346±0.07 | 0.375±0.07 | 0.383±0.07 | 0.388±0.06 | 0.391±0.06 | 0.386±0.06 | 0.388±0.05 | 0.380±0.04 | 0.364±0.05 |
| Grid | 1e+04 | 0.064±0.01 | 0.121±0.01 | 0.178±0.01 | 0.232±0.01 | 0.290±0.01 | 0.323±0.00 | 0.349±0.00 | 0.371±0.00 | 0.387±0.00 |
| | 3e+04 | 0.041±0.01 | 0.076±0.01 | 0.117±0.01 | 0.159±0.00 | 0.198±0.00 | 0.232±0.00 | 0.267±0.00 | 0.291±0.00 | 0.302±0.00 |
| | 1e+05 | 0.031±0.00 | 0.056±0.01 | 0.082±0.01 | 0.120±0.00 | 0.156±0.00 | 0.183±0.00 | 0.205±0.00 | 0.219±0.00 | 0.228±0.00 |
| | 3e+05 | 0.018±0.00 | 0.032±0.00 | 0.050±0.00 | 0.076±0.00 | 0.108±0.00 | 0.129±0.00 | 0.148±0.00 | 0.158±0.00 | 0.164±0.00 |
| | 1e+06 | 0.002±0.00 | 0.003±0.00 | 0.004±0.00 | 0.006±0.00 | 0.007±0.00 | 0.008±0.00 | 0.010±0.00 | 0.011±0.00 | 0.012±0.00 |
| | 3e+06 | 0.000±0.00 | 0.000±0.00 | 0.000±0.00 | 0.000±0.00 | 0.000±0.00 | 0.000±0.00 | 0.000±0.00 | 0.000±0.00 | 0.000±0.00 |

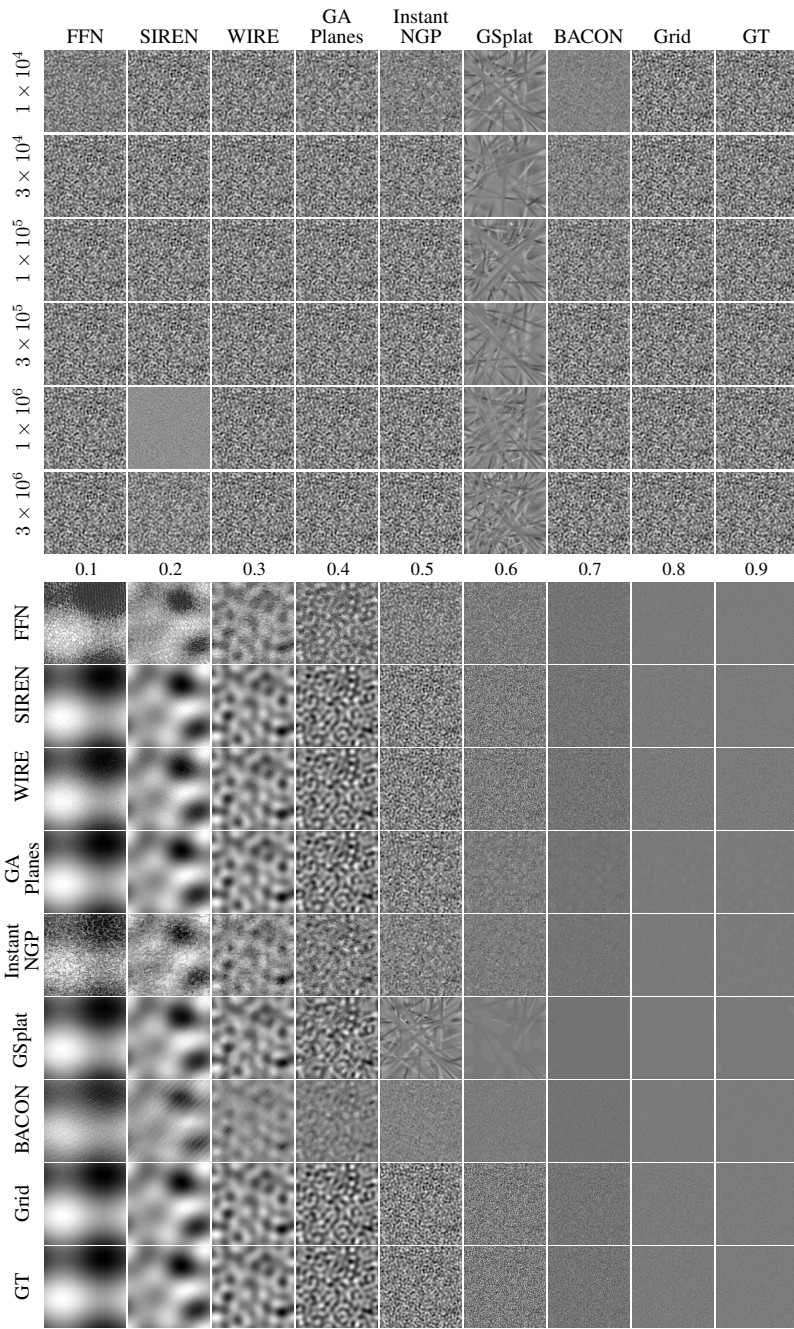

Figure 8: **2D Bandlimited Signal overfitting with bandwidth = 0.5 (top) and model size = $1 \times 10^4$ (bottom).** Sinusoid-based models (FFN, SIREN) and BACON which is also bandlimited exhibit characteristic wave-like artifacts, while GA-Planes introduces subtle blurring and axis-aligned artifacts. Instant-NGP struggles with severe noise, likely due to hash collisions. Grid remains stable under limited resource conditions but blurs high-bandwidth signals that exceed the Nyquist frequency. Detailed quantitative results are in Tables 8 to 10.

Table 8: Comparison of model performance across bandwidths on 2D Bandlimited signal (PSNR).

| Model | Size | 0.1 | 0.2 | 0.3 | 0.4 | 0.5 | 0.6 | 0.7 | 0.8 | 0.9 |
|---|---|---|---|---|---|---|---|---|---|---|
| FFN | 1e+04 | 19.08±2.30 | 19.19±1.57 | 20.61±1.29 | 21.61±1.07 | 21.65±0.20 | 20.95±0.32 | 20.17±0.19 | 19.32±0.13 | 16.78±0.26 |
| | 3e+04 | 32.08±0.65 | 32.75±0.81 | 33.37±1.33 | 33.96±1.07 | 33.36±1.04 | 30.14±0.58 | 23.84±0.12 | 20.12±0.18 | 16.94±0.28 |
| | 1e+05 | 42.70±0.75 | 43.59±0.53 | 44.38±0.17 | 44.87±0.49 | 44.66±0.47 | 43.01±0.30 | 34.75±0.35 | 23.00±0.17 | 17.42±0.25 |
| | 3e+05 | 49.36±0.32 | 50.24±0.16 | 50.99±0.29 | 51.46±0.36 | 51.36±0.33 | 50.62±0.28 | 44.87±0.18 | 30.36±0.26 | 18.89±0.33 |
| | 1e+06 | 55.11±0.54 | 55.88±0.66 | 56.21±0.65 | 56.84±0.85 | 56.53±0.76 | 56.07±0.35 | 52.97±0.54 | 39.58±0.58 | 21.03±0.26 |
| | 3e+06 | 62.51±0.40 | 64.03±0.53 | 64.87±0.82 | 64.99±0.49 | 63.51±1.55 | 61.60±1.44 | 58.36±1.03 | 45.62±1.23 | 24.46±1.00 |
| SIREN | 1e+04 | 34.38±1.76 | 34.39±1.73 | 34.41±1.22 | 32.44±2.55 | 30.15±2.63 | 23.85±1.56 | 20.14±0.18 | 19.23±0.11 | 16.76±0.26 |
| | 3e+04 | 31.51±8.00 | 31.18±7.79 | 33.74±8.87 | 32.14±8.16 | 32.08±7.90 | 26.42±5.96 | 20.86±2.26 | 18.96±0.93 | 16.25±1.00 |
| | 1e+05 | 22.74±2.79 | 25.56±6.82 | 30.24±6.84 | 36.87±7.11 | 32.89±8.40 | 28.48±7.35 | 24.38±3.35 | 18.99±1.10 | 16.35±0.43 |
| | 3e+05 | 20.57±3.73 | 16.82±3.97 | 19.52±3.26 | 22.29±4.34 | 20.62±5.48 | 21.18±3.88 | 18.30±2.50 | 16.14±2.12 | 15.13±1.51 |
| | 1e+06 | 13.75±2.49 | 14.15±4.69 | 14.44±6.43 | 11.70±3.04 | 16.19±5.78 | 12.08±2.68 | 16.25±5.91 | 12.97±3.67 | 11.65±2.61 |
| | 3e+06 | 26.38±12.15 | 24.49±12.31 | 23.17±6.49 | 26.11±9.92 | 18.24±6.08 | 21.69±8.17 | 25.29±9.31 | 16.06±4.99 | 23.62±8.77 |
| WIRE | 1e+04 | 28.43±0.69 | 28.67±0.65 | 28.97±0.84 | 28.98±0.54 | 27.45±0.61 | 23.53±0.42 | 20.45±0.17 | 19.12±0.13 | 16.64±0.27 |
| | 3e+04 | 32.86±0.62 | 33.03±0.72 | 33.31±0.79 | 33.39±0.56 | 32.93±0.68 | 30.44±0.60 | 23.85±0.18 | 19.93±0.11 | 16.86±0.26 |
| | 1e+05 | 38.84±0.43 | 38.90±0.57 | 39.11±0.58 | 39.26±0.34 | 39.13±0.50 | 38.58±0.46 | 32.50±0.33 | 22.13±0.21 | 17.27±0.28 |
| | 3e+05 | 43.38±0.60 | 43.59±0.59 | 43.76±0.27 | 43.60±0.61 | 43.60±0.52 | 43.21±0.63 | 39.88±0.41 | 26.12±0.28 | 17.82±0.31 |
| | 1e+06 | 44.52±0.60 | 44.62±0.43 | 45.00±0.86 | 44.62±0.74 | 44.89±1.15 | 44.68±0.88 | 42.27±0.41 | 32.99±0.31 | 19.12±0.35 |
| | 3e+06 | 44.47±0.86 | 46.32±3.47 | 48.89±3.25 | 44.48±0.91 | 45.18±3.09 | 46.11±4.46 | 41.93±0.74 | 33.17±0.17 | 20.13±0.31 |
| GA-Planes | 1e+04 | 53.60±21.79 | 49.22±18.49 | 38.66±11.96 | 30.44±6.51 | 22.06±2.25 | 19.76±0.64 | 19.65±0.13 | 19.21±0.13 | 16.80±0.28 |
| | 3e+04 | 65.42±1.06 | 57.76±2.99 | 50.28±2.41 | 43.28±1.38 | 31.07±2.33 | 22.90±0.96 | 20.71±0.25 | 19.62±0.16 | 16.98±0.28 |
| | 1e+05 | 65.24±1.24 | 57.96±2.00 | 53.29±3.28 | 50.21±1.92 | 43.73±1.65 | 32.81±1.86 | 24.16±0.75 | 20.92±0.22 | 17.62±0.26 |
| | 3e+05 | 64.50±0.59 | 58.51±0.97 | 54.04±1.48 | 53.45±1.04 | 51.37±0.69 | 49.90±1.80 | 33.64±1.29 | 24.63±0.53 | 19.43±0.25 |
| | 1e+06 | 64.06±0.52 | 59.82±0.79 | 55.59±1.88 | 56.19±0.82 | 54.90±1.00 | 54.73±1.06 | 49.25±1.23 | 37.18±1.76 | 23.15±3.15 |
| | 3e+06 | 58.83±1.77 | 55.57±1.32 | 51.90±1.84 | 51.40±1.63 | 53.10±1.33 | 54.06±2.33 | 51.73±1.29 | 42.64±2.27 | 26.81±4.49 |
| Instant-NGP | 1e+04 | 17.16±0.58 | 19.60±0.75 | 21.49±0.89 | 21.89±0.65 | 20.91±0.48 | 19.86±0.37 | 19.58±0.22 | 19.16±0.08 | 16.75±0.26 |
| | 3e+04 | 27.65±0.90 | 30.82±1.67 | 30.00±1.33 | 29.09±0.56 | 25.87±0.39 | 22.23±0.33 | 20.49±0.15 | 19.46±0.11 | 16.85±0.31 |
| | 1e+05 | 36.54±2.42 | 37.46±1.44 | 38.14±1.21 | 36.56±0.79 | 32.84±0.55 | 27.26±0.23 | 22.76±0.17 | 20.32±0.17 | 17.51±0.26 |
| | 3e+05 | 43.76±2.89 | 43.62±2.97 | 46.29±0.81 | 44.57±0.67 | 40.80±0.30 | 34.23±0.23 | 27.77±0.23 | 22.99±0.27 | 19.71±0.28 |
| | 1e+06 | 44.14±2.50 | 44.19±0.82 | 47.78±1.17 | 48.30±1.28 | 44.50±0.66 | 37.88±0.31 | 31.34±0.28 | 25.78±0.31 | 21.97±0.55 |
| | 3e+06 | 58.94±3.04 | 62.89±4.08 | 63.80±2.34 | 66.46±1.68 | 67.05±1.37 | 69.04±2.78 | 66.43±6.42 | 58.55±4.53 | 48.68±0.66 |
| GSplat | 1e+04 | 47.92±1.84 | 42.94±2.50 | 35.27±2.41 | 25.51±1.78 | 19.05±0.36 | 18.82±0.40 | 19.34±0.22 | 19.12±0.10 | 16.74±0.26 |
| | 3e+04 | 47.51±4.76 | 42.52±3.13 | 35.20±2.12 | 25.64±0.91 | 19.24±0.54 | 18.84±0.40 | 19.34±0.22 | 19.12±0.10 | 16.74±0.26 |
| | 1e+05 | 47.15±3.89 | 42.67±3.04 | 34.70±2.98 | 24.52±1.43 | 19.24±0.47 | 18.84±0.40 | 19.34±0.22 | 19.12±0.10 | 16.74±0.26 |
| | 3e+05 | 47.90±3.17 | 42.02±2.87 | 36.20±1.24 | 24.99±1.00 | 19.18±0.62 | 18.86±0.38 | 19.34±0.22 | 19.12±0.10 | 16.74±0.26 |
| | 1e+06 | 49.42±3.12 | 42.89±2.83 | 35.14±1.35 | 25.86±1.10 | 19.08±0.40 | 18.86±0.40 | 19.35±0.22 | 19.12±0.10 | 16.74±0.26 |
| | 3e+06 | 49.25±4.16 | 43.12±4.32 | 34.67±1.62 | 24.56±1.57 | 19.40±0.21 | 18.86±0.41 | 19.35±0.22 | 19.12±0.10 | 16.74±0.26 |
| BACON | 1e+04 | 18.29±0.49 | 18.31±0.86 | 18.80±0.81 | 19.38±0.66 | 19.42±0.58 | 19.71±0.45 | 20.00±0.27 | 19.52±0.09 | 16.85±0.26 |
| | 3e+04 | 22.08±0.72 | 21.92±0.56 | 22.19±0.89 | 22.19±0.73 | 21.99±0.47 | 21.71±0.32 | 21.44±0.17 | 20.52±0.13 | 17.12±0.26 |
| | 1e+05 | 28.03±0.28 | 28.33±0.44 | 28.03±0.38 | 27.87±0.75 | 27.50±0.64 | 27.08±0.48 | 25.67±0.52 | 23.48±0.06 | 17.98±0.26 |
| | 3e+05 | 37.90±0.57 | 38.33±0.83 | 38.66±0.84 | 39.47±0.61 | 39.14±0.85 | 38.58±0.48 | 36.23±1.02 | 31.18±0.62 | 20.33±0.28 |
| | 1e+06 | 48.97±2.79 | 49.24±2.93 | 49.61±3.39 | 49.04±3.71 | 47.75±3.98 | 48.08±4.72 | 46.96±3.04 | 42.99±3.07 | 25.95±0.80 |
| | 3e+06 | 45.44±12.56 | 45.35±12.38 | 46.37±14.85 | 46.20±13.29 | 47.03±13.80 | 46.17±14.00 | 45.52±15.32 | 39.21±10.41 | 25.17±6.25 |
| Grid | 1e+04 | 64.97±0.10 | 61.49±0.37 | 56.26±0.87 | 51.92±0.74 | 50.91±0.55 | 26.60±0.48 | 20.45±0.22 | 19.36±0.10 | 16.80±0.26 |
| | 3e+04 | 69.78±0.10 | 66.28±0.36 | 60.95±0.86 | 55.97±0.72 | 51.80±0.52 | 47.33±0.47 | 24.01±0.23 | 19.89±0.10 | 16.91±0.26 |
| | 1e+05 | 75.06±0.10 | 71.55±0.36 | 66.18±0.86 | 61.02±0.71 | 55.81±0.52 | 51.99±0.44 | 43.15±0.22 | 22.47±0.09 | 17.34±0.27 |
| | 3e+05 | 79.87±0.10 | 76.36±0.36 | 70.98±0.86 | 65.78±0.71 | 60.35±0.53 | 55.18±0.45 | 52.39±0.22 | 35.96±0.09 | 18.87±0.26 |
| | 1e+06 | 138.64±0.08 | 139.02±0.97 | 140.02±1.01 | 140.79±0.59 | 140.45±0.81 | 139.77±0.50 | 137.84±1.00 | 105.67±0.45 | 49.96±0.28 |
| | 3e+06 | 106.10±1.09 | 121.94±5.67 | 125.94±10.48 | 135.95±5.16 | 140.97±2.42 | 144.42±0.65 | 146.03±1.48 | 145.79±0.60 | 145.72±0.74 |

Table 9: Comparison of model performance across bandwidths on 2D Bandlimited signal (SSIM).

| Model | Size | 0.1 | 0.2 | 0.3 | 0.4 | 0.5 | 0.6 | 0.7 | 0.8 | 0.9 |
|---|---|---|---|---|---|---|---|---|---|---|
| FFN | 1e+04 | 0.503±0.11 | 0.564±0.09 | 0.618±0.10 | 0.664±0.05 | 0.631±0.02 | 0.526±0.03 | 0.299±0.03 | 0.132±0.02 | 0.053±0.01 |
| | 3e+04 | 0.702±0.03 | 0.740±0.03 | 0.783±0.05 | 0.836±0.03 | 0.871±0.03 | 0.865±0.02 | 0.683±0.02 | 0.313±0.03 | 0.102±0.01 |
| | 1e+05 | 0.946±0.01 | 0.957±0.00 | 0.967±0.00 | 0.976±0.00 | 0.984±0.00 | 0.988±0.00 | 0.969±0.00 | 0.693±0.02 | 0.234±0.01 |
| | 3e+05 | 0.987±0.00 | 0.990±0.00 | 0.992±0.00 | 0.995±0.00 | 0.996±0.00 | 0.998±0.00 | 0.997±0.00 | 0.954±0.00 | 0.540±0.02 |
| | 1e+06 | 0.996±0.00 | 0.997±0.00 | 0.998±0.00 | 0.998±0.00 | 0.999±0.00 | 0.999±0.00 | 1.000±0.00 | 0.994±0.00 | 0.751±0.02 |
| | 3e+06 | 0.999±0.00 | 1.000±0.00 | 1.000±0.00 | 1.000±0.00 | 1.000±0.00 | 1.000±0.00 | 1.000±0.00 | 0.999±0.00 | 0.897±0.03 |
| SIREN | 1e+04 | 0.904±0.03 | 0.907±0.03 | 0.919±0.02 | 0.912±0.03 | 0.889±0.04 | 0.699±0.07 | 0.229±0.01 | 0.095±0.00 | 0.044±0.00 |
| | 3e+04 | 0.756±0.22 | 0.752±0.22 | 0.810±0.24 | 0.791±0.21 | 0.847±0.21 | 0.799±0.16 | 0.468±0.03 | 0.144±0.02 | 0.057±0.01 |
| | 1e+05 | 0.596±0.14 | 0.625±0.24 | 0.809±0.10 | 0.924±0.06 | 0.867±0.17 | 0.869±0.12 | 0.798±0.10 | 0.253±0.04 | 0.083±0.02 |
| | 3e+05 | 0.416±0.23 | 0.212±0.22 | 0.381±0.21 | 0.560±0.23 | 0.464±0.30 | 0.666±0.13 | 0.565±0.19 | 0.202±0.16 | 0.116±0.04 |
| | 1e+06 | 0.082±0.05 | 0.110±0.11 | 0.152±0.25 | 0.049±0.06 | 0.239±0.24 | 0.070±0.11 | 0.341±0.32 | 0.163±0.18 | 0.047±0.04 |
| | 3e+06 | 0.494±0.36 | 0.426±0.36 | 0.366±0.21 | 0.524±0.31 | 0.245±0.24 | 0.445±0.39 | 0.553±0.38 | 0.196±0.34 | 0.594±0.46 |
| WIRE | 1e+04 | 0.644±0.03 | 0.660±0.03 | 0.689±0.03 | 0.725±0.02 | 0.737±0.02 | 0.648±0.01 | 0.343±0.01 | 0.135±0.00 | 0.055±0.00 |
| | 3e+04 | 0.726±0.03 | 0.738±0.03 | 0.764±0.03 | 0.801±0.02 | 0.846±0.02 | 0.866±0.01 | 0.671±0.01 | 0.271±0.01 | 0.090±0.01 |
| | 1e+05 | 0.883±0.01 | 0.887±0.01 | 0.901±0.01 | 0.921±0.01 | 0.946±0.01 | 0.969±0.00 | 0.951±0.00 | 0.607±0.02 | 0.195±0.01 |
| | 3e+05 | 0.959±0.00 | 0.960±0.01 | 0.968±0.00 | 0.972±0.00 | 0.982±0.00 | 0.991±0.00 | 0.993±0.00 | 0.872±0.01 | 0.342±0.02 |
| | 1e+06 | 0.984±0.00 | 0.985±0.00 | 0.988±0.00 | 0.989±0.00 | 0.993±0.00 | 0.997±0.00 | 0.997±0.00 | 0.980±0.00 | 0.588±0.02 |
| | 3e+06 | 0.982±0.00 | 0.982±0.01 | 0.991±0.00 | 0.988±0.00 | 0.992±0.00 | 0.996±0.00 | 0.997±0.00 | 0.981±0.00 | 0.700±0.01 |
| GA-Planes | 1e+04 | 0.968±0.06 | 0.977±0.04 | 0.966±0.06 | 0.931±0.08 | 0.717±0.08 | 0.416±0.04 | 0.218±0.03 | 0.114±0.02 | 0.065±0.01 |
| | 3e+04 | 1.000±0.00 | 0.999±0.00 | 0.999±0.00 | 0.991±0.00 | 0.912±0.02 | 0.639±0.04 | 0.391±0.02 | 0.233±0.02 | 0.132±0.02 |
| | 1e+05 | 1.000±0.00 | 1.000±0.00 | 0.999±0.00 | 0.997±0.00 | 0.988±0.00 | 0.932±0.02 | 0.719±0.04 | 0.497±0.02 | 0.325±0.03 |
| | 3e+05 | 1.000±0.00 | 1.000±0.00 | 0.999±0.00 | 0.998±0.00 | 0.997±0.00 | 0.998±0.00 | 0.966±0.01 | 0.815±0.02 | 0.633±0.02 |
| | 1e+06 | 1.000±0.00 | 1.000±0.00 | 0.999±0.00 | 0.999±0.00 | 0.999±0.00 | 0.999±0.00 | 0.999±0.00 | 0.991±0.00 | 0.759±0.31 |
| | 3e+06 | 0.999±0.00 | 0.998±0.00 | 0.998±0.00 | 0.996±0.00 | 0.998±0.00 | 0.999±0.00 | 0.999±0.00 | 0.997±0.00 | 0.864±0.21 |
| Instant-NGP | 1e+04 | 0.327±0.04 | 0.485±0.04 | 0.605±0.05 | 0.634±0.04 | 0.577±0.02 | 0.414±0.01 | 0.196±0.02 | 0.094±0.00 | 0.045±0.00 |
| | 3e+04 | 0.671±0.04 | 0.809±0.06 | 0.796±0.05 | 0.795±0.02 | 0.716±0.01 | 0.566±0.01 | 0.345±0.01 | 0.175±0.01 | 0.078±0.02 |
| | 1e+05 | 0.862±0.06 | 0.895±0.03 | 0.920±0.02 | 0.910±0.01 | 0.870±0.01 | 0.774±0.00 | 0.604±0.01 | 0.375±0.02 | 0.287±0.03 |
| | 3e+05 | 0.953±0.03 | 0.955±0.02 | 0.981±0.00 | 0.978±0.00 | 0.966±0.00 | 0.928±0.00 | 0.861±0.00 | 0.713±0.01 | 0.659±0.01 |
| | 1e+06 | 0.958±0.02 | 0.965±0.01 | 0.985±0.00 | 0.990±0.00 | 0.985±0.00 | 0.966±0.00 | 0.935±0.01 | 0.860±0.01 | 0.819±0.02 |
| | 3e+06 | 0.999±0.00 | 0.999±0.00 | 1.000±0.00 | 1.000±0.00 | 1.000±0.00 | 1.000±0.00 | 1.000±0.00 | 1.000±0.00 | 1.000±0.00 |
| GSplat | 1e+04 | 0.996±0.00 | 0.997±0.00 | 0.991±0.00 | 0.924±0.01 | 0.606±0.02 | 0.327±0.02 | 0.153±0.01 | 0.085±0.00 | 0.042±0.00 |
| | 3e+04 | 0.996±0.00 | 0.996±0.00 | 0.992±0.00 | 0.927±0.01 | 0.618±0.03 | 0.328±0.02 | 0.153±0.01 | 0.085±0.00 | 0.042±0.00 |
| | 1e+05 | 0.997±0.00 | 0.997±0.00 | 0.992±0.00 | 0.917±0.02 | 0.618±0.02 | 0.328±0.02 | 0.153±0.01 | 0.085±0.00 | 0.042±0.00 |
| | 3e+05 | 0.996±0.00 | 0.997±0.00 | 0.994±0.00 | 0.925±0.01 | 0.614±0.03 | 0.329±0.02 | 0.153±0.01 | 0.085±0.00 | 0.042±0.00 |
| | 1e+06 | 0.996±0.00 | 0.997±0.00 | 0.993±0.00 | 0.933±0.01 | 0.610±0.02 | 0.328±0.02 | 0.153±0.01 | 0.085±0.00 | 0.042±0.00 |
| | 3e+06 | 0.997±0.00 | 0.997±0.00 | 0.993±0.00 | 0.919±0.02 | 0.627±0.01 | 0.328±0.02 | 0.153±0.01 | 0.085±0.00 | 0.042±0.00 |
| BACON | 1e+04 | 0.099±0.01 | 0.118±0.02 | 0.172±0.03 | 0.254±0.03 | 0.336±0.02 | 0.351±0.02 | 0.299±0.01 | 0.211±0.01 | 0.082±0.00 |
| | 3e+04 | 0.176±0.03 | 0.188±0.02 | 0.229±0.03 | 0.287±0.03 | 0.373±0.01 | 0.466±0.01 | 0.499±0.01 | 0.432±0.01 | 0.174±0.00 |
| | 1e+05 | 0.416±0.01 | 0.462±0.03 | 0.471±0.02 | 0.513±0.03 | 0.608±0.02 | 0.734±0.01 | 0.789±0.02 | 0.750±0.01 | 0.396±0.00 |
| | 3e+05 | 0.858±0.02 | 0.879±0.01 | 0.897±0.01 | 0.926±0.01 | 0.947±0.01 | 0.971±0.00 | 0.980±0.01 | 0.962±0.01 | 0.713±0.00 |
| | 1e+06 | 0.983±0.01 | 0.985±0.01 | 0.986±0.01 | 0.986±0.01 | 0.988±0.01 | 0.993±0.01 | 0.998±0.00 | 0.997±0.00 | 0.934±0.01 |
| | 3e+06 | 0.804±0.24 | 0.807±0.23 | 0.776±0.27 | 0.833±0.20 | 0.882±0.14 | 0.915±0.10 | 0.934±0.08 | 0.956±0.05 | 0.739±0.29 |
| Grid | 1e+04 | 1.000±0.00 | 0.999±0.00 | 0.998±0.00 | 0.997±0.00 | 0.997±0.00 | 0.818±0.01 | 0.253±0.00 | 0.109±0.00 | 0.048±0.00 |
| | 3e+04 | 1.000±0.00 | 1.000±0.00 | 0.999±0.00 | 0.999±0.00 | 0.998±0.00 | 0.996±0.00 | 0.664±0.00 | 0.224±0.00 | 0.079±0.00 |
| | 1e+05 | 1.000±0.00 | 1.000±0.00 | 1.000±0.00 | 1.000±0.00 | 0.999±0.00 | 0.999±0.00 | 0.996±0.00 | 0.651±0.00 | 0.223±0.00 |
| | 3e+05 | 1.000±0.00 | 1.000±0.00 | 1.000±0.00 | 1.000±0.00 | 1.000±0.00 | 1.000±0.00 | 1.000±0.00 | 0.988±0.00 | 0.548±0.00 |
| | 1e+06 | 1.000±0.00 | 1.000±0.00 | 1.000±0.00 | 1.000±0.00 | 1.000±0.00 | 1.000±0.00 | 1.000±0.00 | 1.000±0.00 | 1.000±0.00 |
| | 3e+06 | 1.000±0.00 | 1.000±0.00 | 1.000±0.00 | 1.000±0.00 | 1.000±0.00 | 1.000±0.00 | 1.000±0.00 | 1.000±0.00 | 1.000±0.00 |

Table 10: Comparison of model performance across bandwidths on 2D Bandlimited signal (LPIPS).

| Model | Size | 0.1 | 0.2 | 0.3 | 0.4 | 0.5 | 0.6 | 0.7 | 0.8 | 0.9 |
|---|---|---|---|---|---|---|---|---|---|---|
| FFN | 1e+04 | 0.605±0.04 | 0.603±0.03 | 0.628±0.04 | 0.666±0.03 | 0.608±0.05 | 0.598±0.05 | 0.595±0.05 | 0.680±0.04 | 0.887±0.05 |
| | 3e+04 | 0.592±0.01 | 0.583±0.01 | 0.588±0.02 | 0.576±0.02 | 0.459±0.03 | 0.450±0.02 | 0.453±0.01 | 0.559±0.01 | 0.811±0.01 |
| | 1e+05 | 0.412±0.01 | 0.404±0.01 | 0.406±0.01 | 0.395±0.01 | 0.362±0.02 | 0.144±0.02 | 0.187±0.01 | 0.403±0.01 | 0.677±0.01 |
| | 3e+05 | 0.385±0.02 | 0.354±0.01 | 0.348±0.00 | 0.341±0.01 | 0.164±0.02 | 0.022±0.00 | 0.029±0.00 | 0.168±0.01 | 0.461±0.02 |
| | 1e+06 | 0.205±0.03 | 0.162±0.04 | 0.157±0.02 | 0.112±0.03 | 0.022±0.01 | 0.003±0.00 | 0.002±0.00 | 0.026±0.00 | 0.338±0.01 |
| | 3e+06 | 0.012±0.00 | 0.007±0.00 | 0.004±0.00 | 0.002±0.00 | 0.001±0.00 | 0.001±0.00 | 0.001±0.00 | 0.004±0.00 | 0.213±0.04 |
| SIREN | 1e+04 | 0.479±0.03 | 0.469±0.03 | 0.442±0.02 | 0.418±0.05 | 0.315±0.06 | 0.433±0.08 | 0.635±0.01 | 0.774±0.01 | 0.946±0.01 |
| | 3e+04 | 0.470±0.10 | 0.480±0.10 | 0.447±0.12 | 0.471±0.17 | 0.287±0.18 | 0.337±0.18 | 0.492±0.02 | 0.652±0.02 | 0.863±0.03 |
| | 1e+05 | 0.560±0.06 | 0.536±0.11 | 0.478±0.07 | 0.357±0.07 | 0.297±0.18 | 0.273±0.18 | 0.341±0.10 | 0.570±0.02 | 0.807±0.01 |
| | 3e+05 | 0.634±0.07 | 0.749±0.10 | 0.680±0.08 | 0.646±0.10 | 0.677±0.15 | 0.549±0.09 | 0.527±0.06 | 0.602±0.03 | 0.756±0.02 |
| | 1e+06 | 0.794±0.07 | 0.828±0.09 | 0.881±0.12 | 0.997±0.09 | 0.843±0.15 | 0.902±0.08 | 0.661±0.11 | 0.634±0.04 | 0.603±0.12 |
| | 3e+06 | 0.684±0.18 | 0.737±0.17 | 0.809±0.10 | 0.801±0.15 | 0.856±0.10 | 0.725±0.17 | 0.510±0.23 | 0.577±0.12 | 0.219±0.20 |
| WIRE | 1e+04 | 0.549±0.01 | 0.545±0.01 | 0.562±0.01 | 0.580±0.01 | 0.505±0.01 | 0.496±0.01 | 0.559±0.01 | 0.637±0.00 | 0.844±0.00 |
| | 3e+04 | 0.525±0.01 | 0.526±0.01 | 0.543±0.01 | 0.547±0.02 | 0.462±0.02 | 0.372±0.01 | 0.442±0.00 | 0.541±0.01 | 0.767±0.01 |
| | 1e+05 | 0.474±0.01 | 0.478±0.01 | 0.477±0.01 | 0.447±0.02 | 0.374±0.02 | 0.222±0.01 | 0.198±0.02 | 0.395±0.01 | 0.670±0.01 |
| | 3e+05 | 0.426±0.01 | 0.419±0.01 | 0.414±0.01 | 0.413±0.02 | 0.356±0.02 | 0.115±0.01 | 0.041±0.00 | 0.236±0.01 | 0.596±0.01 |
| | 1e+06 | 0.383±0.01 | 0.367±0.01 | 0.326±0.01 | 0.260±0.04 | 0.077±0.02 | 0.015±0.00 | 0.015±0.01 | 0.050±0.00 | 0.470±0.01 |
| | 3e+06 | 0.369±0.02 | 0.338±0.04 | 0.314±0.01 | 0.248±0.03 | 0.106±0.04 | 0.015±0.01 | 0.009±0.00 | 0.038±0.00 | 0.382±0.00 |
| GA-Planes | 1e+04 | 0.032±0.05 | 0.034±0.05 | 0.121±0.13 | 0.299±0.13 | 0.536±0.07 | 0.673±0.05 | 0.619±0.08 | 0.620±0.08 | 0.667±0.12 |
| | 3e+04 | 0.004±0.00 | 0.009±0.00 | 0.044±0.01 | 0.132±0.03 | 0.369±0.05 | 0.560±0.01 | 0.596±0.05 | 0.498±0.03 | 0.521±0.02 |
| | 1e+05 | 0.006±0.00 | 0.012±0.00 | 0.044±0.01 | 0.081±0.03 | 0.133±0.04 | 0.263±0.02 | 0.411±0.01 | 0.404±0.02 | 0.388±0.01 |
| | 3e+05 | 0.008±0.00 | 0.013±0.00 | 0.044±0.01 | 0.081±0.02 | 0.055±0.01 | 0.015±0.01 | 0.141±0.01 | 0.282±0.01 | 0.288±0.01 |
| | 1e+06 | 0.010±0.00 | 0.014±0.00 | 0.040±0.01 | 0.059±0.01 | 0.024±0.01 | 0.004±0.00 | 0.005±0.00 | 0.034±0.02 | 0.225±0.19 |
| | 3e+06 | 0.068±0.03 | 0.078±0.05 | 0.090±0.01 | 0.182±0.04 | 0.050±0.02 | 0.005±0.00 | 0.003±0.00 | 0.011±0.01 | 0.138±0.14 |
| Instant-NGP | 1e+04 | 0.719±0.01 | 0.681±0.01 | 0.690±0.02 | 0.729±0.01 | 0.668±0.01 | 0.653±0.01 | 0.653±0.02 | 0.695±0.02 | 0.872±0.01 |
| | 3e+04 | 0.616±0.01 | 0.570±0.03 | 0.609±0.02 | 0.627±0.01 | 0.591±0.01 | 0.606±0.01 | 0.609±0.01 | 0.606±0.01 | 0.619±0.09 |
| | 1e+05 | 0.528±0.04 | 0.508±0.03 | 0.508±0.03 | 0.545±0.01 | 0.503±0.01 | 0.527±0.01 | 0.511±0.00 | 0.555±0.01 | 0.404±0.02 |
| | 3e+05 | 0.474±0.03 | 0.474±0.05 | 0.434±0.01 | 0.436±0.02 | 0.466±0.02 | 0.473±0.01 | 0.443±0.01 | 0.458±0.01 | 0.277±0.01 |
| | 1e+06 | 0.519±0.03 | 0.529±0.02 | 0.464±0.04 | 0.402±0.03 | 0.415±0.02 | 0.424±0.02 | 0.398±0.01 | 0.365±0.01 | 0.204±0.02 |
| | 3e+06 | 0.072±0.07 | 0.018±0.02 | 0.012±0.01 | 0.003±0.00 | 0.000±0.00 | 0.000±0.00 | 0.000±0.00 | 0.000±0.00 | 0.001±0.00 |
| GSplat | 1e+04 | 0.023±0.01 | 0.038±0.01 | 0.102±0.02 | 0.290±0.03 | 0.622±0.01 | 0.733±0.01 | 0.689±0.00 | 0.722±0.00 | 0.895±0.00 |
| | 3e+04 | 0.008±0.00 | 0.055±0.03 | 0.112±0.02 | 0.293±0.03 | 0.607±0.03 | 0.734±0.01 | 0.689±0.00 | 0.722±0.00 | 0.895±0.00 |
| | 1e+05 | 0.011±0.00 | 0.031±0.01 | 0.097±0.02 | 0.293±0.03 | 0.607±0.01 | 0.737±0.01 | 0.692±0.00 | 0.723±0.00 | 0.895±0.00 |
| | 3e+05 | 0.007±0.00 | 0.036±0.01 | 0.073±0.01 | 0.284±0.02 | 0.611±0.01 | 0.734±0.00 | 0.691±0.00 | 0.722±0.00 | 0.895±0.00 |
| | 1e+06 | 0.010±0.00 | 0.023±0.00 | 0.093±0.02 | 0.268±0.02 | 0.617±0.01 | 0.733±0.01 | 0.690±0.00 | 0.724±0.00 | 0.895±0.00 |
| | 3e+06 | 0.006±0.00 | 0.038±0.00 | 0.099±0.02 | 0.281±0.03 | 0.596±0.01 | 0.734±0.00 | 0.693±0.00 | 0.724±0.00 | 0.895±0.00 |
| BACON | 1e+04 | 0.720±0.02 | 0.736±0.01 | 0.775±0.01 | 0.838±0.01 | 0.767±0.01 | 0.687±0.01 | 0.554±0.01 | 0.467±0.01 | 0.622±0.01 |
| | 3e+04 | 0.693±0.01 | 0.710±0.02 | 0.767±0.02 | 0.812±0.02 | 0.717±0.02 | 0.630±0.01 | 0.506±0.01 | 0.419±0.00 | 0.551±0.01 |
| | 1e+05 | 0.624±0.01 | 0.629±0.01 | 0.688±0.01 | 0.696±0.02 | 0.587±0.02 | 0.515±0.01 | 0.404±0.02 | 0.334±0.01 | 0.462±0.00 |
| | 3e+05 | 0.505±0.03 | 0.508±0.02 | 0.501±0.01 | 0.482±0.01 | 0.501±0.03 | 0.322±0.02 | 0.150±0.03 | 0.119±0.01 | 0.305±0.00 |
| | 1e+06 | 0.464±0.07 | 0.442±0.07 | 0.430±0.09 | 0.425±0.11 | 0.332±0.16 | 0.117±0.12 | 0.025±0.02 | 0.015±0.01 | 0.140±0.01 |
| | 3e+06 | 0.361±0.24 | 0.357±0.22 | 0.307±0.30 | 0.297±0.27 | 0.258±0.31 | 0.213±0.26 | 0.155±0.19 | 0.105±0.12 | 0.241±0.21 |
| Grid | 1e+04 | 0.023±0.00 | 0.047±0.01 | 0.061±0.00 | 0.038±0.00 | 0.014±0.00 | 0.293±0.00 | 0.602±0.00 | 0.716±0.00 | 0.895±0.00 |
| | 3e+04 | 0.005±0.00 | 0.018±0.00 | 0.042±0.01 | 0.075±0.00 | 0.031±0.00 | 0.018±0.00 | 0.381±0.00 | 0.549±0.00 | 0.754±0.00 |
| | 1e+05 | 0.000±0.00 | 0.001±0.00 | 0.002±0.00 | 0.005±0.00 | 0.012±0.00 | 0.011±0.00 | 0.017±0.00 | 0.346±0.00 | 0.594±0.00 |
| | 3e+05 | 0.000±0.00 | 0.000±0.00 | 0.001±0.00 | 0.002±0.00 | 0.008±0.00 | 0.017±0.00 | 0.003±0.00 | 0.036±0.00 | 0.474±0.00 |
| | 1e+06 | 0.000±0.00 | 0.000±0.00 | 0.000±0.00 | 0.000±0.00 | 0.000±0.00 | 0.000±0.00 | 0.000±0.00 | 0.000±0.00 | 0.001±0.00 |
| | 3e+06 | 0.000±0.00 | 0.000±0.00 | 0.000±0.00 | 0.000±0.00 | 0.000±0.00 | 0.000±0.00 | 0.000±0.00 | 0.000±0.00 | 0.000±0.00 |

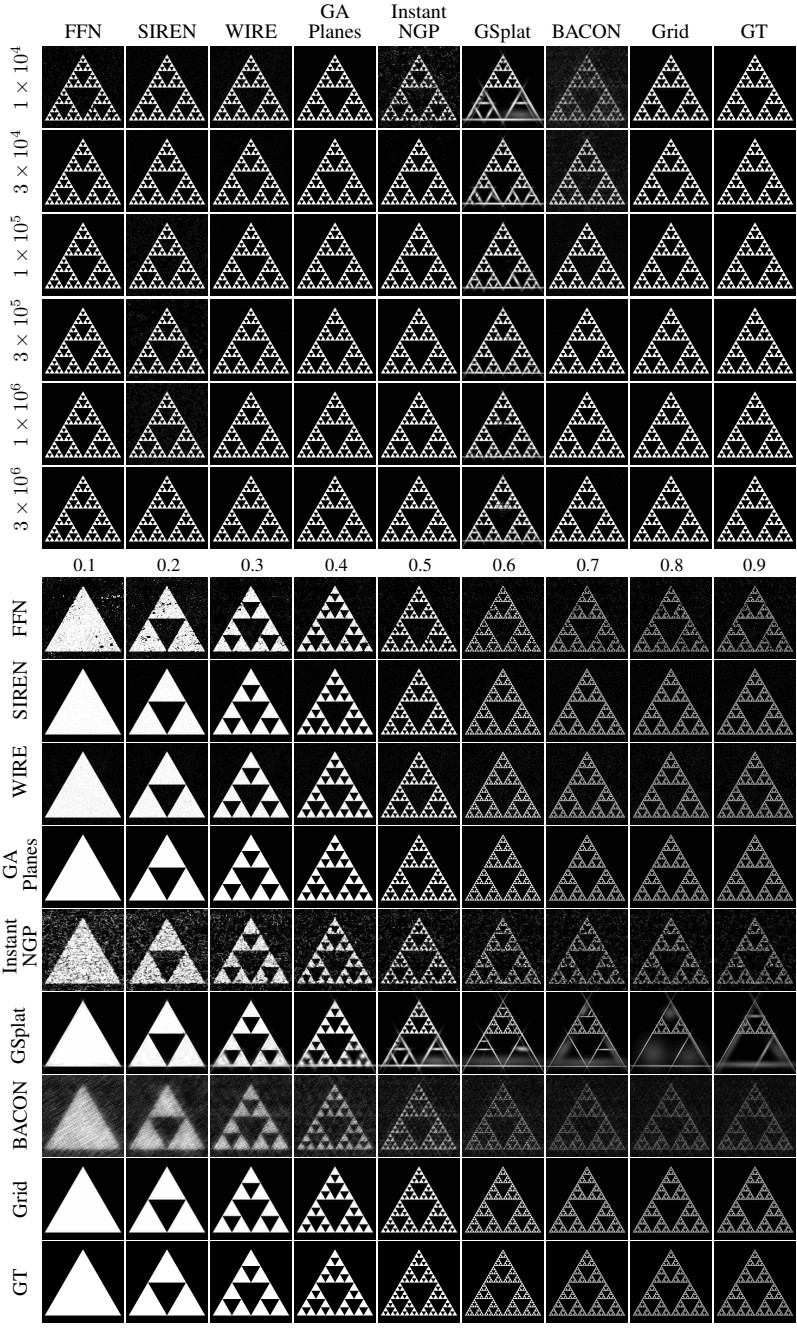

Figure 9: **2D Sierpinski overfitting with bandwidth = 0.5 (top) and model size = $1 \times 10^4$ (bottom).** In general, performance improves as model size increases. We note that for some signals including Sierpinski, Instant-NGP and BACON produce noisy outputs and GSplat fails to fit high frequency regions. Detailed quantitative results are in Tables 11 to 13.

Table 11: Comparison of model performance across bandwidths on 2D Sierpinski signal (PSNR).

| Model | Size | 0.1 | 0.2 | 0.3 | 0.4 | 0.5 | 0.6 | 0.7 | 0.8 | 0.9 |
|-------|------|-----|-----|-----|-----|-----|-----|-----|-----|-----|
| FFN | 1e+04 | 15.83 | 14.66 | 17.18 | 17.87 | 17.94 | 16.46 | 15.32 | 14.71 | 14.72 |
| | 3e+04 | 30.63 | 30.26 | 29.99 | 29.44 | 28.60 | 27.21 | 25.04 | 22.78 | 22.51 |
| | 1e+05 | 34.65 | 34.97 | 35.17 | 35.68 | 34.58 | 32.51 | 31.32 | 28.42 | 28.01 |
| | 3e+05 | 37.59 | 36.95 | 37.13 | 38.89 | 41.45 | 39.55 | 37.54 | 33.37 | 32.03 |
| | 1e+06 | 43.01 | 44.47 | 43.80 | 45.42 | 48.56 | 45.25 | 42.28 | 38.38 | 38.03 |
| | 3e+06 | 48.31 | 46.44 | 47.32 | 37.30 | 36.32 | 35.58 | 32.20 | 23.60 | 22.44 |
| SIREN | 1e+04 | 27.17 | 24.78 | 23.14 | 21.30 | 19.57 | 17.88 | 15.37 | 14.83 | 14.83 |
| | 3e+04 | 28.84 | 27.24 | 25.41 | 23.54 | 21.73 | 19.97 | 18.28 | 16.70 | 16.68 |
| | 1e+05 | 30.75 | 28.59 | 25.93 | 26.42 | 20.18 | 15.29 | 19.04 | 19.42 | 18.17 |
| | 3e+05 | 28.11 | 25.89 | 22.75 | 18.46 | 19.49 | 16.94 | 16.91 | 17.57 | 16.76 |
| | 1e+06 | 11.34 | 10.60 | 13.97 | 10.61 | 17.39 | 20.80 | 16.97 | 11.19 | 14.30 |
| | 3e+06 | 41.83 | 31.80 | 19.49 | 19.61 | 37.50 | 22.32 | 16.87 | 37.59 | 41.56 |
| WIRE | 1e+04 | 24.66 | 23.77 | 22.20 | 20.76 | 19.21 | 17.71 | 16.26 | 15.53 | 15.54 |
| | 3e+04 | 28.00 | 26.71 | 25.31 | 23.82 | 22.23 | 20.54 | 18.99 | 17.64 | 17.62 |
| | 1e+05 | 31.17 | 29.94 | 28.45 | 26.99 | 25.34 | 23.74 | 22.13 | 20.81 | 20.63 |
| | 3e+05 | 34.17 | 32.73 | 31.43 | 29.78 | 27.96 | 26.10 | 24.45 | 22.91 | 22.61 |
| | 1e+06 | 35.67 | 34.16 | 33.00 | 31.25 | 29.39 | 27.68 | 26.27 | 25.09 | 25.12 |
| | 3e+06 | 35.57 | 34.32 | 32.32 | 30.97 | 29.07 | 27.37 | 26.09 | 25.63 | 25.35 |
| GA-Planes | 1e+04 | 37.92 | 40.74 | 45.52 | 45.60 | 45.43 | 33.76 | 31.72 | 31.57 | 30.01 |
| | 3e+04 | 50.84 | 56.48 | 44.95 | 54.61 | 50.80 | 51.76 | 42.47 | 40.44 | 37.45 |
| | 1e+05 | 58.00 | 58.68 | 60.22 | 61.74 | 59.88 | 55.52 | 61.68 | 55.90 | 49.81 |
| | 3e+05 | 57.36 | 56.56 | 59.79 | 63.84 | 62.55 | 55.87 | 64.37 | 67.33 | 66.11 |
| | 1e+06 | 55.57 | 53.46 | 55.87 | 58.57 | 61.19 | 60.45 | 61.84 | 57.90 | 59.63 |
| | 3e+06 | 59.03 | 55.02 | 60.83 | 61.65 | 58.04 | 61.54 | 54.90 | 52.53 | 56.59 |
| Instant-NGP | 1e+04 | 12.42 | 12.67 | 13.91 | 14.22 | 14.34 | 13.85 | 13.47 | 13.53 | 13.56 |
| | 3e+04 | 31.41 | 30.44 | 32.35 | 27.43 | 24.74 | 21.25 | 18.60 | 17.48 | 17.65 |
| | 1e+05 | 47.49 | 62.31 | 62.52 | 58.86 | 58.83 | 44.88 | 36.01 | 32.44 | 30.51 |
| | 3e+05 | 62.43 | 63.73 | 63.76 | 59.03 | 60.38 | 53.40 | 56.50 | 59.19 | 68.86 |
| | 1e+06 | 65.92 | 66.45 | 65.04 | 70.98 | 64.84 | 55.37 | 54.05 | 52.31 | 52.33 |
| | 3e+06 | 71.07 | 66.84 | 69.65 | 69.95 | 79.64 | 73.71 | 80.05 | 84.06 | 80.65 |
| GSplat | 1e+04 | 25.71 | 24.04 | 20.03 | 17.74 | 14.21 | 13.38 | 13.25 | 12.88 | 12.79 |
| | 3e+04 | 26.12 | 23.77 | 21.39 | 18.84 | 15.92 | 14.65 | 13.93 | 13.49 | 13.48 |
| | 1e+05 | 25.70 | 24.19 | 21.16 | 19.14 | 17.07 | 15.23 | 14.38 | 14.41 | 14.07 |
| | 3e+05 | 24.92 | 23.91 | 21.07 | 19.32 | 17.99 | 16.33 | 15.00 | 14.80 | 14.71 |
| | 1e+06 | 24.74 | 24.66 | 20.44 | 19.65 | 17.99 | 16.57 | 15.49 | 15.06 | 14.87 |
| | 3e+06 | 24.74 | 24.05 | 21.73 | 19.43 | 17.32 | 16.76 | 15.51 | 14.87 | 14.96 |
| BACON | 1e+04 | 10.91 | 10.81 | 11.11 | 11.26 | 12.00 | 12.35 | 12.58 | 12.68 | 12.68 |
| | 3e+04 | 13.91 | 14.32 | 14.51 | 14.60 | 15.25 | 15.43 | 15.48 | 15.27 | 15.14 |
| | 1e+05 | 21.73 | 21.03 | 21.93 | 22.24 | 22.39 | 21.95 | 21.41 | 20.62 | 20.58 |
| | 3e+05 | 31.78 | 30.97 | 30.15 | 29.70 | 28.79 | 27.82 | 26.55 | 25.31 | 25.02 |
| | 1e+06 | 35.76 | 34.03 | 33.50 | 32.20 | 30.92 | 29.89 | 28.66 | 26.96 | 26.62 |
| | 3e+06 | 30.93 | 30.20 | 28.87 | 27.50 | 26.62 | 25.19 | 23.63 | 23.20 | 23.04 |
| Grid | 1e+04 | 25.93 | 24.17 | 22.42 | 20.65 | 18.86 | 17.38 | 14.97 | 14.63 | 14.63 |
| | 3e+04 | 28.69 | 26.86 | 25.19 | 23.44 | 21.80 | 19.93 | 18.29 | 16.54 | 16.53 |
| | 1e+05 | 31.30 | 29.70 | 27.87 | 26.17 | 24.50 | 22.85 | 21.32 | 19.98 | 19.83 |
| | 3e+05 | 34.01 | 32.58 | 30.92 | 29.31 | 27.56 | 25.92 | 24.41 | 23.12 | 22.89 |
| | 1e+06 | 59.98 | 54.19 | 53.85 | 53.21 | 51.98 | 50.82 | 49.25 | 48.63 | 47.61 |
| | 3e+06 | 137.76 | 139.00 | 140.24 | 141.46 | 142.64 | 143.74 | 144.64 | 145.19 | 145.20 |

Table 12: Comparison of model performance across bandwidths on 2D Sierpinski signal (SSIM).

| Model | Size | 0.1 | 0.2 | 0.3 | 0.4 | 0.5 | 0.6 | 0.7 | 0.8 | 0.9 |
|---|---|---|---|---|---|---|---|---|---|---|
| FFN | 1e+04 | 0.365 | 0.207 | 0.406 | 0.546 | 0.609 | 0.538 | 0.505 | 0.495 | 0.508 |
| | 3e+04 | 0.838 | 0.870 | 0.901 | 0.917 | 0.933 | 0.949 | 0.960 | 0.955 | 0.949 |
| | 1e+05 | 0.858 | 0.900 | 0.954 | 0.970 | 0.968 | 0.973 | 0.973 | 0.985 | 0.982 |
| | 3e+05 | 0.922 | 0.914 | 0.867 | 0.938 | 0.974 | 0.970 | 0.993 | 0.957 | 0.981 |
| | 1e+06 | 0.984 | 0.981 | 0.979 | 0.978 | 0.998 | 0.998 | 0.999 | 0.998 | 0.995 |
| | 3e+06 | 0.998 | 0.996 | 0.993 | 0.885 | 0.822 | 0.804 | 0.691 | 0.817 | 0.735 |
| SIREN | 1e+04 | 0.871 | 0.731 | 0.636 | 0.531 | 0.458 | 0.519 | 0.374 | 0.398 | 0.390 |
| | 3e+04 | 0.789 | 0.746 | 0.705 | 0.595 | 0.507 | 0.453 | 0.344 | 0.346 | 0.347 |
| | 1e+05 | 0.870 | 0.811 | 0.673 | 0.735 | 0.414 | 0.285 | 0.796 | 0.463 | 0.389 |
| | 3e+05 | 0.759 | 0.766 | 0.530 | 0.339 | 0.388 | 0.243 | 0.313 | 0.297 | 0.303 |
| | 1e+06 | 0.064 | 0.034 | 0.041 | 0.027 | 0.248 | 0.350 | 0.185 | 0.048 | 0.214 |
| | 3e+06 | 0.870 | 0.524 | 0.220 | 0.175 | 0.690 | 0.320 | 0.251 | 0.705 | 0.934 |
| WIRE | 1e+04 | 0.490 | 0.458 | 0.408 | 0.363 | 0.329 | 0.310 | 0.305 | 0.298 | 0.302 |
| | 3e+04 | 0.591 | 0.542 | 0.495 | 0.437 | 0.385 | 0.338 | 0.310 | 0.286 | 0.288 |
| | 1e+05 | 0.753 | 0.732 | 0.670 | 0.615 | 0.536 | 0.481 | 0.427 | 0.372 | 0.371 |
| | 3e+05 | 0.889 | 0.860 | 0.816 | 0.748 | 0.686 | 0.618 | 0.584 | 0.558 | 0.537 |
| | 1e+06 | 0.926 | 0.886 | 0.865 | 0.811 | 0.745 | 0.663 | 0.610 | 0.536 | 0.545 |
| | 3e+06 | 0.939 | 0.921 | 0.887 | 0.850 | 0.807 | 0.723 | 0.658 | 0.604 | 0.595 |
| GA-Planes | 1e+04 | 0.995 | 0.986 | 0.998 | 0.999 | 0.999 | 0.938 | 0.994 | 0.994 | 0.990 |
| | 3e+04 | 0.999 | 1.000 | 0.994 | 1.000 | 1.000 | 1.000 | 1.000 | 0.999 | 0.999 |
| | 1e+05 | 1.000 | 1.000 | 1.000 | 1.000 | 1.000 | 0.999 | 1.000 | 1.000 | 1.000 |
| | 3e+05 | 0.999 | 0.999 | 0.999 | 1.000 | 1.000 | 0.999 | 1.000 | 1.000 | 1.000 |
| | 1e+06 | 0.995 | 0.994 | 0.996 | 0.998 | 0.999 | 0.998 | 0.999 | 0.998 | 0.999 |
| | 3e+06 | 0.997 | 0.980 | 0.999 | 0.999 | 0.999 | 1.000 | 0.998 | 0.996 | 1.000 |
| Instant-NGP | 1e+04 | 0.183 | 0.183 | 0.211 | 0.228 | 0.269 | 0.269 | 0.249 | 0.257 | 0.361 |
| | 3e+04 | 0.757 | 0.836 | 0.687 | 0.866 | 0.746 | 0.816 | 0.831 | 0.564 | 0.834 |
| | 1e+05 | 0.984 | 0.999 | 0.999 | 1.000 | 1.000 | 1.000 | 0.995 | 0.989 | 0.981 |
| | 3e+05 | 0.998 | 0.999 | 1.000 | 0.998 | 1.000 | 1.000 | 1.000 | 1.000 | 1.000 |
| | 1e+06 | 0.999 | 0.999 | 1.000 | 1.000 | 0.997 | 1.000 | 1.000 | 1.000 | 1.000 |
| | 3e+06 | 1.000 | 0.998 | 1.000 | 1.000 | 1.000 | 1.000 | 1.000 | 1.000 | 1.000 |
| GSplat | 1e+04 | 0.937 | 0.864 | 0.769 | 0.717 | 0.650 | 0.602 | 0.585 | 0.433 | 0.598 |
| | 3e+04 | 0.940 | 0.877 | 0.802 | 0.745 | 0.719 | 0.690 | 0.675 | 0.624 | 0.621 |
| | 1e+05 | 0.935 | 0.875 | 0.775 | 0.736 | 0.724 | 0.675 | 0.640 | 0.619 | 0.650 |
| | 3e+05 | 0.922 | 0.866 | 0.771 | 0.737 | 0.731 | 0.711 | 0.690 | 0.687 | 0.688 |
| | 1e+06 | 0.923 | 0.873 | 0.755 | 0.725 | 0.712 | 0.697 | 0.706 | 0.678 | 0.693 |
| | 3e+06 | 0.922 | 0.872 | 0.794 | 0.724 | 0.703 | 0.713 | 0.669 | 0.678 | 0.665 |
| BACON | 1e+04 | 0.014 | 0.011 | 0.013 | 0.018 | 0.029 | 0.040 | 0.047 | 0.044 | 0.045 |
| | 3e+04 | 0.027 | 0.029 | 0.032 | 0.044 | 0.067 | 0.098 | 0.118 | 0.118 | 0.114 |
| | 1e+05 | 0.122 | 0.100 | 0.124 | 0.152 | 0.178 | 0.207 | 0.225 | 0.245 | 0.238 |
| | 3e+05 | 0.614 | 0.637 | 0.640 | 0.667 | 0.586 | 0.619 | 0.550 | 0.482 | 0.544 |
| | 1e+06 | 0.916 | 0.849 | 0.867 | 0.827 | 0.789 | 0.722 | 0.704 | 0.658 | 0.652 |
| | 3e+06 | 0.505 | 0.453 | 0.375 | 0.342 | 0.335 | 0.327 | 0.330 | 0.330 | 0.326 |
| Grid | 1e+04 | 0.929 | 0.897 | 0.855 | 0.808 | 0.773 | 0.760 | 0.729 | 0.723 | 0.724 |
| | 3e+04 | 0.956 | 0.934 | 0.908 | 0.871 | 0.838 | 0.824 | 0.827 | 0.801 | 0.801 |
| | 1e+05 | 0.977 | 0.968 | 0.953 | 0.934 | 0.913 | 0.902 | 0.909 | 0.904 | 0.903 |
| | 3e+05 | 0.989 | 0.985 | 0.978 | 0.968 | 0.957 | 0.949 | 0.954 | 0.956 | 0.955 |
| | 1e+06 | 0.999 | 0.995 | 0.995 | 0.994 | 0.993 | 0.993 | 0.994 | 0.994 | 0.994 |
| | 3e+06 | 1.000 | 1.000 | 1.000 | 1.000 | 1.000 | 1.000 | 1.000 | 1.000 | 1.000 |

Table 13: Comparison of model performance across bandwidths on 2D Sierpinski signal (LPIPS).

| Model | Size | 0.1 | 0.2 | 0.3 | 0.4 | 0.5 | 0.6 | 0.7 | 0.8 | 0.9 |
|---|---|---|---|---|---|---|---|---|---|---|
| FFN | 1e+04 | 0.472 | 0.491 | 0.467 | 0.460 | 0.456 | 0.489 | 0.503 | 0.499 | 0.501 |
| | 3e+04 | 0.391 | 0.351 | 0.296 | 0.252 | 0.218 | 0.209 | 0.218 | 0.218 | 0.225 |
| | 1e+05 | 0.366 | 0.331 | 0.245 | 0.153 | 0.134 | 0.127 | 0.152 | 0.125 | 0.139 |
| | 3e+05 | 0.282 | 0.292 | 0.262 | 0.226 | 0.123 | 0.088 | 0.089 | 0.166 | 0.157 |
| | 1e+06 | 0.219 | 0.203 | 0.202 | 0.131 | 0.021 | 0.018 | 0.013 | 0.030 | 0.034 |
| | 3e+06 | 0.096 | 0.066 | 0.055 | 0.297 | 0.323 | 0.333 | 0.382 | 0.361 | 0.396 |
| SIREN | 1e+04 | 0.390 | 0.440 | 0.460 | 0.483 | 0.507 | 0.510 | 0.552 | 0.539 | 0.543 |
| | 3e+04 | 0.360 | 0.386 | 0.398 | 0.418 | 0.436 | 0.460 | 0.487 | 0.482 | 0.483 |
| | 1e+05 | 0.312 | 0.331 | 0.437 | 0.342 | 0.503 | 0.557 | 0.377 | 0.409 | 0.496 |
| | 3e+05 | 0.397 | 0.414 | 0.494 | 0.544 | 0.537 | 0.560 | 0.550 | 0.537 | 0.546 |
| | 1e+06 | 0.703 | 0.737 | 0.655 | 0.722 | 0.567 | 0.490 | 0.594 | 0.662 | 0.601 |
| | 3e+06 | 0.317 | 0.460 | 0.558 | 0.548 | 0.314 | 0.492 | 0.547 | 0.304 | 0.246 |
| WIRE | 1e+04 | 0.503 | 0.503 | 0.516 | 0.526 | 0.547 | 0.566 | 0.568 | 0.561 | 0.562 |
| | 3e+04 | 0.448 | 0.444 | 0.446 | 0.457 | 0.467 | 0.489 | 0.510 | 0.515 | 0.514 |
| | 1e+05 | 0.378 | 0.368 | 0.373 | 0.376 | 0.385 | 0.389 | 0.412 | 0.421 | 0.428 |
| | 3e+05 | 0.336 | 0.340 | 0.348 | 0.370 | 0.373 | 0.388 | 0.396 | 0.401 | 0.406 |
| | 1e+06 | 0.335 | 0.345 | 0.326 | 0.332 | 0.339 | 0.353 | 0.370 | 0.373 | 0.373 |
| | 3e+06 | 0.290 | 0.289 | 0.292 | 0.300 | 0.306 | 0.321 | 0.339 | 0.339 | 0.341 |
| GA-Planes | 1e+04 | 0.102 | 0.094 | 0.032 | 0.011 | 0.025 | 0.103 | 0.043 | 0.048 | 0.054 |
| | 3e+04 | 0.018 | 0.003 | 0.027 | 0.002 | 0.001 | 0.001 | 0.002 | 0.001 | 0.002 |
| | 1e+05 | 0.003 | 0.012 | 0.003 | 0.002 | 0.001 | 0.001 | 0.001 | 0.001 | 0.001 |
| | 3e+05 | 0.031 | 0.038 | 0.015 | 0.003 | 0.002 | 0.003 | 0.001 | 0.000 | 0.001 |
| | 1e+06 | 0.086 | 0.092 | 0.045 | 0.016 | 0.008 | 0.013 | 0.008 | 0.012 | 0.009 |
| | 3e+06 | 0.012 | 0.004 | 0.009 | 0.004 | 0.006 | 0.002 | 0.005 | 0.018 | 0.001 |
| Instant-NGP | 1e+04 | 0.669 | 0.651 | 0.629 | 0.630 | 0.624 | 0.630 | 0.621 | 0.611 | 0.585 |
| | 3e+04 | 0.382 | 0.354 | 0.233 | 0.282 | 0.405 | 0.353 | 0.361 | 0.388 | 0.327 |
| | 1e+05 | 0.195 | 0.007 | 0.006 | 0.002 | 0.001 | 0.002 | 0.047 | 0.076 | 0.093 |
| | 3e+05 | 0.017 | 0.008 | 0.005 | 0.004 | 0.001 | 0.006 | 0.003 | 0.001 | 0.001 |
| | 1e+06 | 0.006 | 0.003 | 0.003 | 0.001 | 0.002 | 0.001 | 0.001 | 0.001 | 0.001 |
| | 3e+06 | 0.001 | 0.001 | 0.001 | 0.002 | 0.000 | 0.001 | 0.000 | 0.000 | 0.000 |
| GSplat | 1e+04 | 0.126 | 0.182 | 0.251 | 0.263 | 0.302 | 0.317 | 0.311 | 0.318 | 0.311 |
| | 3e+04 | 0.138 | 0.181 | 0.224 | 0.255 | 0.287 | 0.309 | 0.312 | 0.315 | 0.313 |
| | 1e+05 | 0.132 | 0.181 | 0.236 | 0.258 | 0.273 | 0.311 | 0.325 | 0.324 | 0.318 |
| | 3e+05 | 0.138 | 0.170 | 0.234 | 0.252 | 0.265 | 0.300 | 0.317 | 0.313 | 0.309 |
| | 1e+06 | 0.140 | 0.168 | 0.230 | 0.250 | 0.275 | 0.303 | 0.303 | 0.301 | 0.301 |
| | 3e+06 | 0.134 | 0.158 | 0.223 | 0.256 | 0.282 | 0.294 | 0.319 | 0.300 | 0.305 |
| BACON | 1e+04 | 0.714 | 0.710 | 0.703 | 0.716 | 0.696 | 0.696 | 0.684 | 0.675 | 0.672 |
| | 3e+04 | 0.660 | 0.664 | 0.664 | 0.666 | 0.662 | 0.653 | 0.634 | 0.614 | 0.613 |
| | 1e+05 | 0.562 | 0.568 | 0.549 | 0.527 | 0.498 | 0.479 | 0.470 | 0.453 | 0.455 |
| | 3e+05 | 0.413 | 0.437 | 0.424 | 0.365 | 0.352 | 0.341 | 0.355 | 0.357 | 0.352 |
| | 1e+06 | 0.335 | 0.352 | 0.320 | 0.325 | 0.317 | 0.319 | 0.326 | 0.332 | 0.332 |
| | 3e+06 | 0.461 | 0.469 | 0.469 | 0.450 | 0.436 | 0.422 | 0.419 | 0.424 | 0.421 |
| Grid | 1e+04 | 0.119 | 0.164 | 0.207 | 0.230 | 0.253 | 0.265 | 0.281 | 0.281 | 0.283 |
| | 3e+04 | 0.069 | 0.098 | 0.125 | 0.149 | 0.147 | 0.180 | 0.184 | 0.202 | 0.203 |
| | 1e+05 | 0.053 | 0.073 | 0.094 | 0.108 | 0.106 | 0.111 | 0.125 | 0.124 | 0.125 |
| | 3e+05 | 0.038 | 0.053 | 0.065 | 0.073 | 0.071 | 0.078 | 0.080 | 0.075 | 0.076 |
| | 1e+06 | 0.002 | 0.011 | 0.011 | 0.011 | 0.010 | 0.009 | 0.009 | 0.008 | 0.009 |
| | 3e+06 | 0.000 | 0.000 | 0.000 | 0.000 | 0.000 | 0.000 | 0.000 | 0.000 | 0.000 |

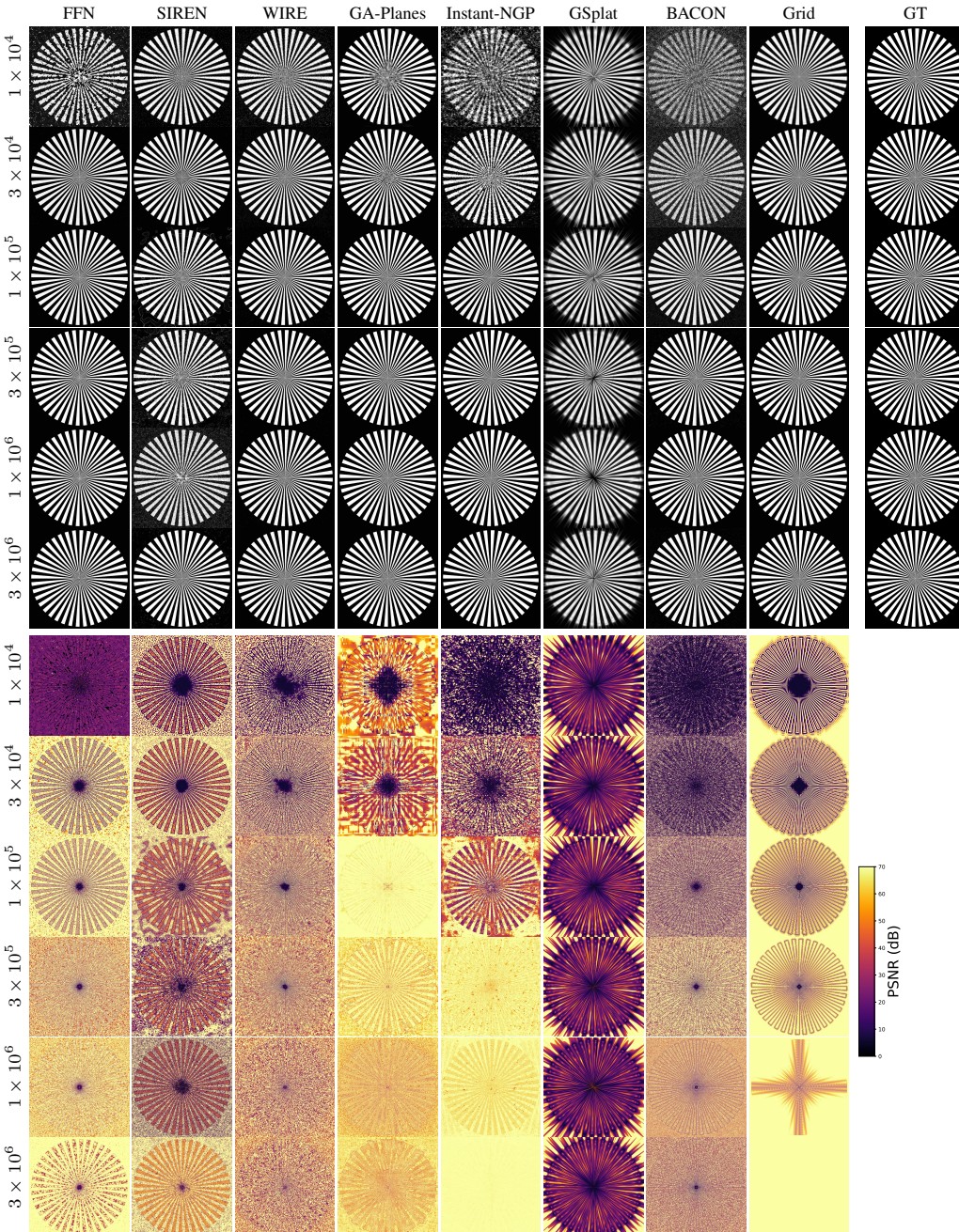

Figure 10: **2D Star Target overfitting outputs (top) and error maps (bottom).** As the signal transitions to higher frequencies toward the center of the Star Target, most models incur larger reconstruction errors (visualized in the lower figure via pixelwise PSNR). The error maps visually reveal the implicit biases of each model, illustrating which regions (e.g., background, constant regions, edges) are best captured by each model at each level of compression. Detailed quantitative results are in Tables 14 to 16.

Table 14: Comparison of model performance across bandwidths on 2D Star Target signal (PSNR).

| Model | Size | 0.1 | 0.2 | 0.3 | 0.4 | 0.5 | 0.6 | 0.7 | 0.8 | 0.9 |
|---|---|---|---|---|---|---|---|---|---|---|
| FFN | 1e+04 | 15.83 | 14.66 | 17.18 | 17.87 | 17.94 | 16.46 | 15.32 | 14.71 | 14.72 |
| | 3e+04 | 30.63 | 30.26 | 29.99 | 29.44 | 28.60 | 27.21 | 25.04 | 22.78 | 22.51 |
| | 1e+05 | 34.65 | 34.97 | 35.17 | 35.68 | 34.58 | 32.51 | 31.32 | 28.42 | 28.01 |
| | 3e+05 | 37.59 | 36.95 | 37.13 | 38.89 | 41.45 | 39.55 | 37.54 | 33.37 | 32.03 |
| | 1e+06 | 43.01 | 44.47 | 43.80 | 45.42 | 48.56 | 45.25 | 42.28 | 38.38 | 38.03 |
| | 3e+06 | 48.31 | 46.44 | 47.32 | 37.30 | 36.32 | 35.58 | 32.20 | 23.60 | 22.44 |
| SIREN | 1e+04 | 27.17 | 24.78 | 23.14 | 21.30 | 19.57 | 17.88 | 15.37 | 14.83 | 14.83 |
| | 3e+04 | 28.84 | 27.24 | 25.41 | 23.54 | 21.73 | 19.97 | 18.28 | 16.70 | 16.68 |
| | 1e+05 | 30.75 | 28.59 | 25.93 | 26.42 | 20.18 | 15.29 | 19.04 | 19.42 | 18.17 |
| | 3e+05 | 28.11 | 25.89 | 22.75 | 18.46 | 19.49 | 16.94 | 16.91 | 17.57 | 16.76 |
| | 1e+06 | 11.34 | 10.60 | 13.97 | 10.61 | 17.39 | 20.80 | 16.97 | 11.19 | 14.30 |
| | 3e+06 | 41.83 | 31.80 | 19.49 | 19.61 | 37.50 | 22.32 | 16.87 | 37.59 | 41.56 |
| WIRE | 1e+04 | 24.66 | 23.77 | 22.20 | 20.76 | 19.21 | 17.71 | 16.26 | 15.53 | 15.54 |
| | 3e+04 | 28.00 | 26.71 | 25.31 | 23.82 | 22.23 | 20.54 | 18.99 | 17.64 | 17.62 |
| | 1e+05 | 31.17 | 29.94 | 28.45 | 26.99 | 25.34 | 23.74 | 22.13 | 20.81 | 20.63 |
| | 3e+05 | 34.17 | 32.73 | 31.43 | 29.78 | 27.96 | 26.10 | 24.45 | 22.91 | 22.61 |
| | 1e+06 | 35.67 | 34.16 | 33.00 | 31.25 | 29.39 | 27.68 | 26.27 | 25.09 | 25.12 |
| | 3e+06 | 35.57 | 34.32 | 32.32 | 30.97 | 29.07 | 27.37 | 26.09 | 25.63 | 25.35 |
| GA-Planes | 1e+04 | 37.92 | 40.74 | 45.52 | 45.60 | 45.43 | 33.76 | 31.72 | 31.57 | 30.01 |
| | 3e+04 | 50.84 | 56.48 | 44.95 | 54.61 | 50.80 | 51.76 | 42.47 | 40.44 | 37.45 |
| | 1e+05 | 58.00 | 58.68 | 60.22 | 61.74 | 59.88 | 55.52 | 61.68 | 55.90 | 49.81 |
| | 3e+05 | 57.36 | 56.56 | 59.79 | 63.84 | 62.55 | 55.87 | 64.37 | 67.33 | 66.11 |
| | 1e+06 | 55.57 | 53.46 | 55.87 | 58.57 | 61.19 | 60.45 | 61.84 | 57.90 | 59.63 |
| | 3e+06 | 59.03 | 55.02 | 60.83 | 61.65 | 58.04 | 61.54 | 54.90 | 52.53 | 56.59 |
| Instant-NGP | 1e+04 | 12.42 | 12.67 | 13.91 | 14.22 | 14.34 | 13.85 | 13.47 | 13.53 | 13.56 |
| | 3e+04 | 31.41 | 30.44 | 32.35 | 27.43 | 24.74 | 21.25 | 18.60 | 17.48 | 17.65 |
| | 1e+05 | 47.49 | 62.31 | 62.52 | 58.86 | 58.83 | 44.88 | 36.01 | 32.44 | 30.51 |
| | 3e+05 | 62.43 | 63.73 | 63.76 | 59.03 | 60.38 | 53.40 | 56.50 | 59.19 | 68.86 |
| | 1e+06 | 65.92 | 66.45 | 65.04 | 70.98 | 64.84 | 55.37 | 54.05 | 52.31 | 52.33 |
| | 3e+06 | 71.07 | 66.84 | 69.65 | 69.95 | 79.64 | 73.71 | 80.05 | 84.06 | 80.65 |
| GSplat | 1e+04 | 25.71 | 24.04 | 20.03 | 17.74 | 14.21 | 13.38 | 13.25 | 12.88 | 12.79 |
| | 3e+04 | 26.12 | 23.77 | 21.39 | 18.84 | 15.92 | 14.65 | 13.93 | 13.49 | 13.48 |
| | 1e+05 | 25.70 | 24.19 | 21.16 | 19.14 | 17.07 | 15.23 | 14.38 | 14.41 | 14.07 |
| | 3e+05 | 24.92 | 23.91 | 21.07 | 19.32 | 17.99 | 16.33 | 15.00 | 14.80 | 14.71 |
| | 1e+06 | 24.74 | 24.66 | 20.44 | 19.65 | 17.99 | 16.57 | 15.49 | 15.06 | 14.87 |
| | 3e+06 | 24.74 | 24.05 | 21.73 | 19.43 | 17.32 | 16.76 | 15.51 | 14.87 | 14.96 |
| BACON | 1e+04 | 10.91 | 10.81 | 11.11 | 11.26 | 12.00 | 12.35 | 12.58 | 12.68 | 12.68 |
| | 3e+04 | 13.91 | 14.32 | 14.51 | 14.60 | 15.25 | 15.43 | 15.48 | 15.27 | 15.14 |
| | 1e+05 | 21.73 | 21.03 | 21.93 | 22.24 | 22.39 | 21.95 | 21.41 | 20.62 | 20.58 |
| | 3e+05 | 31.78 | 30.97 | 30.15 | 29.70 | 28.79 | 27.82 | 26.55 | 25.31 | 25.02 |
| | 1e+06 | 35.76 | 34.03 | 33.50 | 32.20 | 30.92 | 29.89 | 28.66 | 26.96 | 26.62 |
| | 3e+06 | 30.93 | 30.20 | 28.87 | 27.50 | 26.62 | 25.19 | 23.63 | 23.20 | 23.04 |
| Grid | 1e+04 | 25.93 | 24.17 | 22.42 | 20.65 | 18.86 | 17.38 | 14.97 | 14.63 | 14.63 |
| | 3e+04 | 28.69 | 26.86 | 25.19 | 23.44 | 21.80 | 19.93 | 18.29 | 16.54 | 16.53 |
| | 1e+05 | 31.30 | 29.70 | 27.87 | 26.17 | 24.50 | 22.85 | 21.32 | 19.98 | 19.83 |
| | 3e+05 | 34.01 | 32.58 | 30.92 | 29.31 | 27.56 | 25.92 | 24.41 | 23.12 | 22.89 |
| | 1e+06 | 59.98 | 54.19 | 53.85 | 53.21 | 51.98 | 50.82 | 49.25 | 48.63 | 47.61 |
| | 3e+06 | 137.76 | 139.00 | 140.24 | 141.46 | 142.64 | 143.74 | 144.64 | 145.19 | 145.20 |

Table 15: Comparison of model performance across bandwidths on 2D Star Target signal (SSIM).

| Model | Size | 0.1 | 0.2 | 0.3 | 0.4 | 0.5 | 0.6 | 0.7 | 0.8 | 0.9 |
|---|---|---|---|---|---|---|---|---|---|---|
| FFN | 1e+04 | 0.365 | 0.207 | 0.406 | 0.546 | 0.609 | 0.538 | 0.505 | 0.495 | 0.508 |
| | 3e+04 | 0.838 | 0.870 | 0.901 | 0.917 | 0.933 | 0.949 | 0.960 | 0.955 | 0.949 |
| | 1e+05 | 0.858 | 0.900 | 0.954 | 0.970 | 0.968 | 0.973 | 0.973 | 0.985 | 0.982 |
| | 3e+05 | 0.922 | 0.914 | 0.867 | 0.938 | 0.974 | 0.970 | 0.993 | 0.957 | 0.981 |
| | 1e+06 | 0.984 | 0.981 | 0.979 | 0.978 | 0.998 | 0.998 | 0.999 | 0.998 | 0.995 |
| | 3e+06 | 0.998 | 0.996 | 0.993 | 0.885 | 0.822 | 0.804 | 0.691 | 0.817 | 0.735 |
| SIREN | 1e+04 | 0.871 | 0.731 | 0.636 | 0.531 | 0.458 | 0.519 | 0.374 | 0.398 | 0.390 |
| | 3e+04 | 0.789 | 0.746 | 0.705 | 0.595 | 0.507 | 0.453 | 0.344 | 0.346 | 0.347 |
| | 1e+05 | 0.870 | 0.811 | 0.673 | 0.735 | 0.414 | 0.285 | 0.796 | 0.463 | 0.389 |
| | 3e+05 | 0.759 | 0.766 | 0.530 | 0.339 | 0.388 | 0.243 | 0.313 | 0.297 | 0.303 |
| | 1e+06 | 0.064 | 0.034 | 0.041 | 0.027 | 0.248 | 0.350 | 0.185 | 0.048 | 0.214 |
| | 3e+06 | 0.870 | 0.524 | 0.220 | 0.175 | 0.690 | 0.320 | 0.251 | 0.705 | 0.934 |
| WIRE | 1e+04 | 0.490 | 0.458 | 0.408 | 0.363 | 0.329 | 0.310 | 0.305 | 0.298 | 0.302 |
| | 3e+04 | 0.591 | 0.542 | 0.495 | 0.437 | 0.385 | 0.338 | 0.310 | 0.286 | 0.288 |
| | 1e+05 | 0.753 | 0.732 | 0.670 | 0.615 | 0.536 | 0.481 | 0.427 | 0.372 | 0.371 |
| | 3e+05 | 0.889 | 0.860 | 0.816 | 0.748 | 0.686 | 0.618 | 0.584 | 0.558 | 0.537 |
| | 1e+06 | 0.926 | 0.886 | 0.865 | 0.811 | 0.745 | 0.663 | 0.610 | 0.536 | 0.545 |
| | 3e+06 | 0.939 | 0.921 | 0.887 | 0.850 | 0.807 | 0.723 | 0.658 | 0.604 | 0.595 |
| GA-Planes | 1e+04 | 0.995 | 0.986 | 0.998 | 0.999 | 0.999 | 0.938 | 0.994 | 0.994 | 0.990 |
| | 3e+04 | 0.999 | 1.000 | 0.994 | 1.000 | 1.000 | 1.000 | 1.000 | 0.999 | 0.999 |
| | 1e+05 | 1.000 | 1.000 | 1.000 | 1.000 | 1.000 | 0.999 | 1.000 | 1.000 | 1.000 |
| | 3e+05 | 0.999 | 0.999 | 0.999 | 1.000 | 1.000 | 0.999 | 1.000 | 1.000 | 1.000 |
| | 1e+06 | 0.995 | 0.994 | 0.996 | 0.998 | 0.999 | 0.998 | 0.999 | 0.998 | 0.999 |
| | 3e+06 | 0.997 | 0.980 | 0.999 | 0.999 | 0.999 | 1.000 | 0.998 | 0.996 | 1.000 |
| Instant-NGP | 1e+04 | 0.183 | 0.183 | 0.211 | 0.228 | 0.269 | 0.269 | 0.249 | 0.257 | 0.361 |
| | 3e+04 | 0.757 | 0.836 | 0.687 | 0.866 | 0.746 | 0.816 | 0.831 | 0.564 | 0.834 |
| | 1e+05 | 0.984 | 0.999 | 0.999 | 1.000 | 1.000 | 1.000 | 0.995 | 0.989 | 0.981 |
| | 3e+05 | 0.998 | 0.999 | 1.000 | 0.998 | 1.000 | 1.000 | 1.000 | 1.000 | 1.000 |
| | 1e+06 | 0.999 | 0.999 | 1.000 | 1.000 | 0.997 | 1.000 | 1.000 | 1.000 | 1.000 |
| | 3e+06 | 1.000 | 0.998 | 1.000 | 1.000 | 1.000 | 1.000 | 1.000 | 1.000 | 1.000 |
| GSplat | 1e+04 | 0.937 | 0.864 | 0.769 | 0.717 | 0.650 | 0.602 | 0.585 | 0.433 | 0.598 |
| | 3e+04 | 0.940 | 0.877 | 0.802 | 0.745 | 0.719 | 0.690 | 0.675 | 0.624 | 0.621 |
| | 1e+05 | 0.935 | 0.875 | 0.775 | 0.736 | 0.724 | 0.675 | 0.640 | 0.619 | 0.650 |
| | 3e+05 | 0.922 | 0.866 | 0.771 | 0.737 | 0.731 | 0.711 | 0.690 | 0.687 | 0.688 |
| | 1e+06 | 0.923 | 0.873 | 0.755 | 0.725 | 0.712 | 0.697 | 0.706 | 0.678 | 0.693 |
| | 3e+06 | 0.922 | 0.872 | 0.794 | 0.724 | 0.703 | 0.713 | 0.669 | 0.678 | 0.665 |
| BACON | 1e+04 | 0.014 | 0.011 | 0.013 | 0.018 | 0.029 | 0.040 | 0.047 | 0.044 | 0.045 |
| | 3e+04 | 0.027 | 0.029 | 0.032 | 0.044 | 0.067 | 0.098 | 0.118 | 0.118 | 0.114 |
| | 1e+05 | 0.122 | 0.100 | 0.124 | 0.152 | 0.178 | 0.207 | 0.225 | 0.245 | 0.238 |
| | 3e+05 | 0.614 | 0.637 | 0.640 | 0.667 | 0.586 | 0.619 | 0.550 | 0.482 | 0.544 |
| | 1e+06 | 0.916 | 0.849 | 0.867 | 0.827 | 0.789 | 0.722 | 0.704 | 0.658 | 0.652 |
| | 3e+06 | 0.505 | 0.453 | 0.375 | 0.342 | 0.335 | 0.327 | 0.330 | 0.330 | 0.326 |
| Grid | 1e+04 | 0.929 | 0.897 | 0.855 | 0.808 | 0.773 | 0.760 | 0.729 | 0.723 | 0.724 |
| | 3e+04 | 0.956 | 0.934 | 0.908 | 0.871 | 0.838 | 0.824 | 0.827 | 0.801 | 0.801 |
| | 1e+05 | 0.977 | 0.968 | 0.953 | 0.934 | 0.913 | 0.902 | 0.909 | 0.904 | 0.903 |
| | 3e+05 | 0.989 | 0.985 | 0.978 | 0.968 | 0.957 | 0.949 | 0.954 | 0.956 | 0.955 |
| | 1e+06 | 0.999 | 0.995 | 0.995 | 0.994 | 0.993 | 0.993 | 0.994 | 0.994 | 0.994 |
| | 3e+06 | 1.000 | 1.000 | 1.000 | 1.000 | 1.000 | 1.000 | 1.000 | 1.000 | 1.000 |

Table 16: Comparison of model performance across bandwidths on 2D Star Target signal (LPIPS).

| Model | Size | 0.1 | 0.2 | 0.3 | 0.4 | 0.5 | 0.6 | 0.7 | 0.8 | 0.9 |
|---|---|---|---|---|---|---|---|---|---|---|
| FFN | 1e+04 | 0.472 | 0.491 | 0.467 | 0.460 | 0.456 | 0.489 | 0.503 | 0.499 | 0.501 |
| | 3e+04 | 0.391 | 0.351 | 0.296 | 0.252 | 0.218 | 0.209 | 0.218 | 0.218 | 0.225 |
| | 1e+05 | 0.366 | 0.331 | 0.245 | 0.153 | 0.134 | 0.127 | 0.152 | 0.125 | 0.139 |
| | 3e+05 | 0.282 | 0.292 | 0.262 | 0.226 | 0.123 | 0.088 | 0.089 | 0.166 | 0.157 |
| | 1e+06 | 0.219 | 0.203 | 0.202 | 0.131 | 0.021 | 0.018 | 0.013 | 0.030 | 0.034 |
| | 3e+06 | 0.096 | 0.066 | 0.055 | 0.297 | 0.323 | 0.333 | 0.382 | 0.361 | 0.396 |
| SIREN | 1e+04 | 0.390 | 0.440 | 0.460 | 0.483 | 0.507 | 0.510 | 0.552 | 0.539 | 0.543 |
| | 3e+04 | 0.360 | 0.386 | 0.398 | 0.418 | 0.436 | 0.460 | 0.487 | 0.482 | 0.483 |
| | 1e+05 | 0.312 | 0.331 | 0.437 | 0.342 | 0.503 | 0.557 | 0.377 | 0.409 | 0.496 |
| | 3e+05 | 0.397 | 0.414 | 0.494 | 0.544 | 0.537 | 0.560 | 0.550 | 0.537 | 0.546 |
| | 1e+06 | 0.703 | 0.737 | 0.655 | 0.722 | 0.567 | 0.490 | 0.594 | 0.662 | 0.601 |
| | 3e+06 | 0.317 | 0.460 | 0.558 | 0.548 | 0.314 | 0.492 | 0.547 | 0.304 | 0.246 |
| WIRE | 1e+04 | 0.503 | 0.503 | 0.516 | 0.526 | 0.547 | 0.566 | 0.568 | 0.561 | 0.562 |
| | 3e+04 | 0.448 | 0.444 | 0.446 | 0.457 | 0.467 | 0.489 | 0.510 | 0.515 | 0.514 |
| | 1e+05 | 0.378 | 0.368 | 0.373 | 0.376 | 0.385 | 0.389 | 0.412 | 0.421 | 0.428 |
| | 3e+05 | 0.336 | 0.340 | 0.348 | 0.370 | 0.373 | 0.388 | 0.396 | 0.401 | 0.406 |
| | 1e+06 | 0.335 | 0.345 | 0.326 | 0.332 | 0.339 | 0.353 | 0.370 | 0.373 | 0.373 |
| | 3e+06 | 0.290 | 0.289 | 0.292 | 0.300 | 0.306 | 0.321 | 0.339 | 0.339 | 0.341 |
| GA-Planes | 1e+04 | 0.102 | 0.094 | 0.032 | 0.011 | 0.025 | 0.103 | 0.043 | 0.048 | 0.054 |
| | 3e+04 | 0.018 | 0.003 | 0.027 | 0.002 | 0.001 | 0.001 | 0.001 | 0.001 | 0.002 |
| | 1e+05 | 0.003 | 0.012 | 0.003 | 0.002 | 0.001 | 0.001 | 0.001 | 0.001 | 0.001 |
| | 3e+05 | 0.031 | 0.038 | 0.015 | 0.003 | 0.002 | 0.003 | 0.001 | 0.000 | 0.001 |
| | 1e+06 | 0.086 | 0.092 | 0.045 | 0.016 | 0.008 | 0.013 | 0.008 | 0.012 | 0.009 |
| | 3e+06 | 0.012 | 0.004 | 0.009 | 0.004 | 0.006 | 0.002 | 0.005 | 0.018 | 0.001 |
| Instant-NGP | 1e+04 | 0.669 | 0.651 | 0.629 | 0.630 | 0.624 | 0.630 | 0.621 | 0.611 | 0.585 |
| | 3e+04 | 0.382 | 0.354 | 0.233 | 0.282 | 0.405 | 0.353 | 0.361 | 0.388 | 0.327 |
| | 1e+05 | 0.195 | 0.007 | 0.006 | 0.002 | 0.001 | 0.002 | 0.047 | 0.076 | 0.093 |
| | 3e+05 | 0.017 | 0.008 | 0.005 | 0.004 | 0.001 | 0.006 | 0.003 | 0.001 | 0.001 |
| | 1e+06 | 0.006 | 0.003 | 0.003 | 0.001 | 0.002 | 0.001 | 0.001 | 0.001 | 0.001 |
| | 3e+06 | 0.001 | 0.001 | 0.001 | 0.002 | 0.000 | 0.001 | 0.000 | 0.000 | 0.000 |
| GSplat | 1e+04 | 0.126 | 0.182 | 0.251 | 0.263 | 0.302 | 0.317 | 0.311 | 0.318 | 0.311 |
| | 3e+04 | 0.138 | 0.181 | 0.224 | 0.255 | 0.287 | 0.309 | 0.312 | 0.315 | 0.313 |
| | 1e+05 | 0.132 | 0.181 | 0.236 | 0.258 | 0.273 | 0.311 | 0.325 | 0.324 | 0.318 |
| | 3e+05 | 0.138 | 0.170 | 0.234 | 0.252 | 0.265 | 0.300 | 0.317 | 0.313 | 0.309 |
| | 1e+06 | 0.140 | 0.168 | 0.230 | 0.250 | 0.275 | 0.303 | 0.303 | 0.301 | 0.301 |
| | 3e+06 | 0.134 | 0.158 | 0.223 | 0.256 | 0.282 | 0.294 | 0.319 | 0.300 | 0.305 |
| BACON | 1e+04 | 0.714 | 0.710 | 0.703 | 0.716 | 0.696 | 0.696 | 0.684 | 0.675 | 0.672 |
| | 3e+04 | 0.660 | 0.664 | 0.664 | 0.666 | 0.662 | 0.653 | 0.634 | 0.614 | 0.613 |
| | 1e+05 | 0.562 | 0.568 | 0.549 | 0.527 | 0.498 | 0.479 | 0.470 | 0.453 | 0.455 |
| | 3e+05 | 0.413 | 0.437 | 0.424 | 0.365 | 0.352 | 0.341 | 0.355 | 0.357 | 0.352 |
| | 1e+06 | 0.335 | 0.352 | 0.320 | 0.325 | 0.317 | 0.319 | 0.326 | 0.332 | 0.332 |
| | 3e+06 | 0.461 | 0.469 | 0.469 | 0.450 | 0.436 | 0.422 | 0.419 | 0.424 | 0.421 |
| Grid | 1e+04 | 0.119 | 0.164 | 0.207 | 0.230 | 0.253 | 0.265 | 0.281 | 0.281 | 0.283 |
| | 3e+04 | 0.069 | 0.098 | 0.125 | 0.149 | 0.147 | 0.180 | 0.184 | 0.202 | 0.203 |
| | 1e+05 | 0.053 | 0.073 | 0.094 | 0.108 | 0.106 | 0.111 | 0.125 | 0.124 | 0.125 |
| | 3e+05 | 0.038 | 0.053 | 0.065 | 0.073 | 0.071 | 0.078 | 0.080 | 0.075 | 0.076 |
| | 1e+06 | 0.002 | 0.011 | 0.011 | 0.011 | 0.010 | 0.009 | 0.009 | 0.008 | 0.009 |
| | 3e+06 | 0.000 | 0.000 | 0.000 | 0.000 | 0.000 | 0.000 | 0.000 | 0.000 | 0.000 |

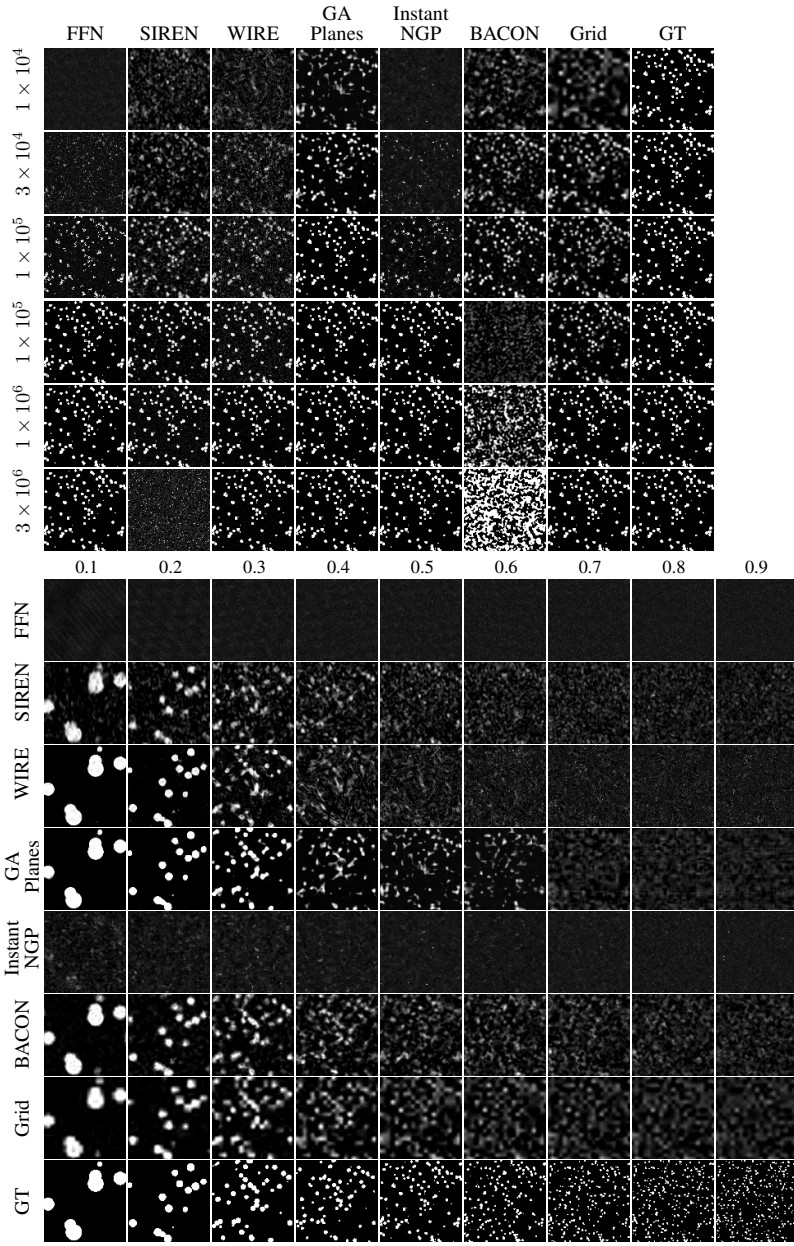

Figure 11: **3D Spheres overfitting with bandwidth = 0.5 (top) and model size = $1 \times 10^4$ (bottom).** FFN and Instant-NGP struggle to fit the 3D Spheres in the most compressed settings, while BACON suffers instability at large model sizes. Detailed quantitative results are in Tables 17 and 18.

Table 17: Comparison of model performance across bandwidths on 3D Spheres signal (PSNR).

| Model | Size | 0.1 | 0.2 | 0.3 | 0.4 | 0.5 | 0.6 | 0.7 | 0.8 | 0.9 |
|---|---|---|---|---|---|---|---|---|---|---|
| FFN | 1e+04 | 11.28±0.16 | 10.96±0.09 | 10.87±0.04 | 10.84±0.04 | 10.81±0.04 | 10.81±0.04 | 10.80±0.04 | 10.79±0.03 | 10.78±0.04 |
| | 3e+04 | 12.04±0.41 | 11.49±0.13 | 11.36±0.15 | 11.33±0.17 | 11.28±0.17 | 11.28±0.16 | 11.26±0.17 | 11.28±0.18 | 11.28±0.15 |
| | 1e+05 | 15.31±0.56 | 14.06±0.21 | 13.84±0.21 | 13.77±0.21 | 13.77±0.17 | 13.72±0.16 | 13.69±0.16 | 13.67±0.17 | 13.67±0.17 |
| | 3e+05 | 27.95±1.37 | 24.35±1.39 | 23.86±1.54 | 23.79±1.21 | 23.59±1.29 | 23.51±1.29 | 23.57±1.29 | 23.45±1.40 | 23.21±1.34 |
| | 1e+06 | 56.65±7.55 | 51.80±3.51 | 51.58±4.73 | 51.17±6.68 | 51.88±4.47 | 54.78±5.87 | 52.55±5.13 | 52.25±4.95 | 53.11±4.82 |
| | 3e+06 | 48.55±1.76 | 45.95±2.24 | 45.46±2.92 | 45.07±2.77 | 46.12±3.61 | 47.10±2.37 | 44.93±2.74 | 46.46±1.93 | 46.39±1.26 |
| SIREN | 1e+04 | 18.52±1.36 | 14.53±0.29 | 13.29±0.14 | 12.35±0.08 | 11.78±0.05 | 11.41±0.03 | 11.19±0.02 | 11.04±0.02 | 10.94±0.01 |
| | 3e+04 | 24.47±1.16 | 19.23±0.95 | 16.40±0.67 | 14.44±0.24 | 13.35±0.14 | 12.68±0.14 | 12.06±0.17 | 11.69±0.10 | 11.48±0.07 |
| | 1e+05 | 23.63±1.45 | 20.64±4.96 | 17.99±1.48 | 17.03±2.49 | 15.29±1.00 | 13.37±1.26 | 13.64±0.56 | 13.34±0.23 | 12.70±0.30 |
| | 3e+05 | 28.16±6.20 | 24.56±6.61 | 19.63±7.44 | 18.37±7.32 | 15.25±3.04 | 17.66±4.96 | 13.91±2.37 | 12.58±2.67 | 13.17±0.91 |
| | 1e+06 | 18.62±9.74 | 17.92±6.43 | 17.02±8.05 | 15.98±8.76 | 17.21±5.82 | 11.66±5.05 | 13.99±6.15 | 13.76±5.31 | 15.43±6.33 |
| | 3e+06 | 24.20±15.49 | 23.59±16.42 | 19.62±17.23 | 30.14±20.37 | 24.75±15.26 | 23.99±15.32 | 18.45±15.82 | 16.10±15.27 | 8.00±2.55 |
| WIRE | 1e+04 | 29.30±1.73 | 21.42±0.95 | 14.36±0.55 | 12.36±0.11 | 11.56±0.10 | 11.21±0.06 | 11.07±0.03 | 10.99±0.02 | 10.94±0.01 |
| | 3e+04 | 34.12±1.42 | 27.27±1.57 | 18.04±0.85 | 14.43±0.34 | 12.66±0.09 | 12.03±0.04 | 11.77±0.03 | 11.65±0.03 | 11.56±0.03 |
| | 1e+05 | 37.94±2.67 | 24.59±1.11 | 18.21±0.23 | 16.06±0.18 | 14.97±0.12 | 14.41±0.07 | 14.08±0.06 | 13.89±0.07 | 13.74±0.06 |
| | 3e+05 | 28.14±0.82 | 24.37±0.16 | 22.40±0.08 | 21.22±0.08 | 20.38±0.04 | 19.85±0.05 | 19.44±0.04 | 19.16±0.03 | 18.93±0.05 |
| | 1e+06 | 36.82±0.44 | 34.15±0.06 | 32.60±0.10 | 31.48±0.06 | 30.66±0.06 | 30.07±0.07 | 29.64±0.09 | 29.21±0.06 | 28.84±0.06 |
| | 3e+06 | 54.01±1.78 | 40.08±0.60 | 37.61±0.24 | 36.05±0.11 | 34.80±0.40 | 33.89±0.21 | 33.36±0.38 | 32.33±0.17 | 31.96±0.30 |
| GA-Planes | 1e+04 | 29.73±4.47 | 25.61±2.63 | 18.84±0.92 | 14.68±0.14 | 12.89±0.07 | 12.05±0.08 | 11.49±0.20 | 11.34±0.14 | 11.17±0.12 |
| | 3e+04 | 50.54±3.32 | 39.19±0.91 | 32.39±0.43 | 23.80±0.74 | 18.01±0.75 | 15.14±0.20 | 13.61±0.17 | 12.92±0.16 | 12.30±0.18 |
| | 1e+05 | 55.76±3.68 | 45.43±1.21 | 39.55±1.23 | 34.04±2.19 | 28.88±0.60 | 24.58±0.38 | 21.18±0.30 | 18.68±0.26 | 16.64±0.22 |
| | 3e+05 | 61.34±6.58 | 50.62±1.60 | 46.14±2.05 | 40.44±1.96 | 36.73±1.60 | 31.88±1.68 | 28.82±0.75 | 27.08±1.28 | 23.95±1.06 |
| | 1e+06 | 58.66±4.04 | 50.89±3.37 | 47.70±2.11 | 42.94±1.25 | 39.24±1.68 | 39.17±4.14 | 36.72±7.91 | 34.39±5.63 | 30.10±1.58 |
| | 3e+06 | 63.69±5.67 | 50.67±3.92 | 45.88±3.08 | 42.06±3.35 | 39.83±3.88 | 36.16±4.00 | 38.94±3.64 | 38.86±4.07 | 37.49±1.92 |
| Instant-NGP | 1e+04 | 11.53±0.17 | 11.16±0.05 | 11.02±0.02 | 10.93±0.01 | 10.89±0.02 | 10.85±0.01 | 10.83±0.01 | 10.84±0.01 | 10.80±0.01 |
| | 3e+04 | 12.40±0.22 | 11.87±0.08 | 11.66±0.04 | 11.52±0.04 | 11.41±0.02 | 11.34±0.07 | 11.26±0.03 | 11.28±0.04 | 11.26±0.04 |
| | 1e+05 | 17.11±0.46 | 15.70±0.18 | 14.60±0.22 | 14.21±0.13 | 13.70±0.11 | 13.42±0.18 | 13.27±0.18 | 13.15±0.13 | 13.04±0.20 |
| | 3e+05 | 42.01±4.30 | 36.21±3.20 | 31.58±1.17 | 28.21±2.30 | 27.88±2.21 | 26.02±1.90 | 22.67±0.77 | 22.15±1.06 | 21.75±0.88 |
| | 1e+06 | 67.78±2.60 | 62.04±6.10 | 54.50±4.88 | 51.39±9.90 | 48.47±4.51 | 51.89±9.87 | 45.62±2.42 | 45.53±4.02 | 45.47±6.80 |
| | 3e+06 | 77.61±6.19 | 78.78±2.45 | 75.74±2.83 | 79.04±3.45 | 78.57±3.13 | 77.86±3.53 | 78.67±2.85 | 80.42±3.97 | 79.98±6.18 |
| BACON | 1e+04 | 21.45±0.28 | 17.48±0.08 | 15.18±0.05 | 13.42±0.05 | 12.40±0.04 | 11.80±0.02 | 11.44±0.02 | 11.21±0.01 | 11.07±0.01 |
| | 3e+04 | 23.59±0.27 | 19.58±0.12 | 17.28±0.03 | 15.72±0.02 | 14.35±0.04 | 13.30±0.02 | 12.53±0.02 | 12.03±0.01 | 11.71±0.01 |
| | 1e+05 | 25.60±0.30 | 21.60±0.16 | 19.30±0.05 | 17.67±0.06 | 16.60±0.02 | 15.53±0.05 | 14.55±0.04 | 13.81±0.03 | 13.23±0.02 |
| | 3e+05 | 17.25±8.15 | 10.34±0.04 | 10.25±0.06 | 10.22±0.06 | 10.18±0.07 | 10.17±0.06 | 10.15±0.07 | 10.14±0.07 | 10.13±0.06 |
| | 1e+06 | 6.76±0.10 | 6.72±0.14 | 6.75±0.14 | 6.77±0.13 | 6.78±0.12 | 6.81±0.15 | 6.82±0.13 | 6.82±0.14 | 6.84±0.14 |
| | 3e+06 | 3.23±0.03 | 3.23±0.02 | 3.22±0.03 | 3.22±0.03 | 3.22±0.02 | 3.24±0.02 | 3.24±0.03 | 3.24±0.02 | 3.23±0.06 |
| Grid | 1e+04 | 20.10±0.27 | 16.61±0.08 | 13.95±0.10 | 12.48±0.03 | 11.74±0.03 | 11.35±0.01 | 11.13±0.01 | 11.00±0.00 | 10.91±0.00 |
| | 3e+04 | 21.96±0.26 | 18.51±0.06 | 16.55±0.03 | 14.72±0.01 | 13.32±0.02 | 12.45±0.01 | 11.90±0.01 | 11.56±0.00 | 11.34±0.01 |
| | 1e+05 | 23.89±0.26 | 20.49±0.05 | 18.52±0.02 | 17.27±0.02 | 16.02±0.02 | 14.80±0.02 | 13.79±0.02 | 13.07±0.01 | 12.54±0.01 |
| | 3e+05 | 26.29±0.26 | 22.87±0.02 | 20.98±0.02 | 19.63±0.01 | 18.65±0.02 | 17.82±0.01 | 16.92±0.02 | 16.04±0.02 | 15.15±0.01 |
| | 1e+06 | 38.68±0.70 | 35.51±0.33 | 33.64±0.24 | 32.30±0.24 | 31.27±0.15 | 30.15±0.06 | 30.16±0.03 | 29.94±0.08 | 29.27±0.05 |
| | 3e+06 | 147.03±0.18 | 146.03±0.06 | 145.34±0.02 | 144.79±0.02 | 144.29±0.01 | 143.84±0.03 | 143.43±0.02 | 143.05±0.02 | 142.68±0.02 |

Table 18: Comparison of model performance across bandwidths on 3D Spheres signal (IoU).

| Model | Size | 0.1 | 0.2 | 0.3 | 0.4 | 0.5 | 0.6 | 0.7 | 0.8 | 0.9 |
|---|---|---|---|---|---|---|---|---|---|---|
| FFN | 1e+04 | 0.04±0.05 | 0.03±0.03 | 0.03±0.03 | 0.03±0.03 | 0.03±0.03 | 0.03±0.03 | 0.03±0.03 | 0.03±0.03 | 0.03±0.03 |
| | 3e+04 | 0.24±0.06 | 0.18±0.03 | 0.18±0.03 | 0.18±0.03 | 0.18±0.03 | 0.18±0.03 | 0.18±0.03 | 0.18±0.04 | 0.18±0.04 |
| | 1e+05 | 0.61±0.04 | 0.52±0.02 | 0.51±0.03 | 0.51±0.03 | 0.51±0.02 | 0.51±0.02 | 0.50±0.02 | 0.49±0.02 | 0.50±0.02 |
| | 3e+05 | 0.98±0.01 | 0.96±0.01 | 0.95±0.02 | 0.95±0.01 | 0.95±0.01 | 0.95±0.02 | 0.95±0.02 | 0.95±0.02 | 0.94±0.02 |
| | 1e+06 | 1.00±0.00 | 1.00±0.00 | 1.00±0.00 | 1.00±0.00 | 1.00±0.00 | 1.00±0.00 | 1.00±0.00 | 1.00±0.00 | 1.00±0.00 |
| | 3e+06 | 1.00±0.00 | 1.00±0.00 | 1.00±0.00 | 1.00±0.00 | 1.00±0.00 | 1.00±0.00 | 1.00±0.00 | 1.00±0.00 | 1.00±0.00 |
| SIREN | 1e+04 | 0.69±0.07 | 0.49±0.03 | 0.40±0.01 | 0.34±0.01 | 0.30±0.01 | 0.25±0.00 | 0.21±0.01 | 0.17±0.01 | 0.13±0.01 |
| | 3e+04 | 0.89±0.02 | 0.75±0.03 | 0.61±0.06 | 0.49±0.04 | 0.42±0.02 | 0.38±0.01 | 0.34±0.02 | 0.29±0.01 | 0.27±0.01 |
| | 1e+05 | 0.87±0.02 | 0.74±0.21 | 0.69±0.08 | 0.61±0.12 | 0.52±0.07 | 0.40±0.08 | 0.44±0.04 | 0.42±0.02 | 0.38±0.02 |
| | 3e+05 | 0.87±0.20 | 0.85±0.11 | 0.65±0.28 | 0.59±0.29 | 0.51±0.19 | 0.60±0.31 | 0.43±0.13 | 0.36±0.16 | 0.39±0.06 |
| | 1e+06 | 0.55±0.40 | 0.61±0.37 | 0.53±0.38 | 0.43±0.43 | 0.55±0.37 | 0.25±0.29 | 0.37±0.38 | 0.41±0.34 | 0.54±0.37 |
| | 3e+06 | 0.63±0.45 | 0.55±0.41 | 0.45±0.45 | 0.63±0.45 | 0.60±0.44 | 0.63±0.45 | 0.38±0.39 | 0.36±0.37 | 0.13±0.09 |
| WIRE | 1e+04 | 0.96±0.01 | 0.83±0.04 | 0.47±0.05 | 0.34±0.01 | 0.27±0.01 | 0.21±0.02 | 0.18±0.01 | 0.15±0.01 | 0.14±0.01 |
| | 3e+04 | 0.99±0.00 | 0.94±0.02 | 0.67±0.05 | 0.44±0.02 | 0.35±0.01 | 0.31±0.00 | 0.29±0.00 | 0.27±0.00 | 0.27±0.00 |
| | 1e+05 | 1.00±0.00 | 0.90±0.02 | 0.62±0.01 | 0.50±0.01 | 0.45±0.00 | 0.43±0.00 | 0.41±0.00 | 0.41±0.00 | 0.40±0.00 |
| | 3e+05 | 0.99±0.00 | 0.96±0.00 | 0.88±0.00 | 0.80±0.01 | 0.74±0.00 | 0.70±0.00 | 0.67±0.00 | 0.65±0.00 | 0.64±0.00 |
| | 1e+06 | 1.00±0.00 | 1.00±0.00 | 1.00±0.00 | 1.00±0.00 | 1.00±0.00 | 1.00±0.00 | 1.00±0.00 | 1.00±0.00 | 1.00±0.00 |
| | 3e+06 | 1.00±0.00 | 1.00±0.00 | 1.00±0.00 | 1.00±0.00 | 1.00±0.00 | 1.00±0.00 | 1.00±0.00 | 1.00±0.00 | 1.00±0.00 |
| GA-Planes | 1e+04 | 0.95±0.03 | 0.93±0.06 | 0.76±0.07 | 0.56±0.01 | 0.42±0.01 | 0.33±0.01 | 0.24±0.04 | 0.22±0.03 | 0.18±0.03 |
| | 3e+04 | 1.00±0.00 | 1.00±0.00 | 0.99±0.00 | 0.93±0.02 | 0.78±0.07 | 0.63±0.03 | 0.51±0.02 | 0.45±0.03 | 0.35±0.03 |
| | 1e+05 | 1.00±0.00 | 1.00±0.00 | 1.00±0.00 | 1.00±0.00 | 0.99±0.00 | 0.96±0.00 | 0.92±0.00 | 0.85±0.01 | 0.75±0.02 |
| | 3e+05 | 1.00±0.00 | 1.00±0.00 | 1.00±0.00 | 1.00±0.00 | 1.00±0.00 | 0.99±0.00 | 0.98±0.00 | 0.98±0.01 | 0.95±0.01 |
| | 1e+06 | 1.00±0.00 | 1.00±0.00 | 1.00±0.00 | 1.00±0.00 | 1.00±0.00 | 1.00±0.00 | 0.99±0.00 | 0.99±0.01 | 0.99±0.00 |
| | 3e+06 | 1.00±0.00 | 1.00±0.00 | 1.00±0.00 | 1.00±0.00 | 1.00±0.00 | 1.00±0.01 | 1.00±0.00 | 1.00±0.00 | 1.00±0.00 |
| Instant-NGP | 1e+04 | 0.14±0.01 | 0.13±0.01 | 0.11±0.01 | 0.09±0.01 | 0.07±0.01 | 0.05±0.01 | 0.05±0.01 | 0.06±0.01 | 0.04±0.02 |
| | 3e+04 | 0.29±0.01 | 0.26±0.00 | 0.24±0.01 | 0.23±0.01 | 0.20±0.02 | 0.20±0.02 | 0.18±0.01 | 0.18±0.01 | 0.20±0.01 |
| | 1e+05 | 0.67±0.02 | 0.63±0.02 | 0.55±0.03 | 0.53±0.02 | 0.48±0.01 | 0.45±0.03 | 0.44±0.03 | 0.43±0.02 | 0.41±0.02 |
| | 3e+05 | 1.00±0.00 | 1.00±0.00 | 0.99±0.00 | 0.98±0.01 | 0.98±0.01 | 0.97±0.01 | 0.94±0.01 | 0.92±0.01 | 0.92±0.02 |
| | 1e+06 | 1.00±0.00 | 1.00±0.00 | 1.00±0.00 | 1.00±0.00 | 1.00±0.00 | 1.00±0.00 | 1.00±0.00 | 1.00±0.00 | 1.00±0.00 |
| | 3e+06 | 1.00±0.00 | 1.00±0.00 | 1.00±0.00 | 1.00±0.00 | 1.00±0.00 | 1.00±0.00 | 1.00±0.00 | 1.00±0.00 | 1.00±0.00 |
| BACON | 1e+04 | 0.77±0.01 | 0.62±0.02 | 0.52±0.01 | 0.42±0.00 | 0.36±0.01 | 0.30±0.01 | 0.25±0.00 | 0.21±0.00 | 0.17±0.00 |
| | 3e+04 | 0.82±0.01 | 0.69±0.02 | 0.60±0.01 | 0.52±0.01 | 0.46±0.01 | 0.41±0.01 | 0.37±0.00 | 0.33±0.01 | 0.30±0.00 |
| | 1e+05 | 0.88±0.01 | 0.76±0.01 | 0.66±0.01 | 0.59±0.01 | 0.55±0.02 | 0.49±0.01 | 0.46±0.02 | 0.43±0.02 | 0.40±0.00 |
| | 3e+05 | 0.42±0.41 | 0.08±0.00 | 0.07±0.01 | 0.07±0.01 | 0.07±0.00 | 0.07±0.00 | 0.07±0.00 | 0.06±0.00 | 0.06±0.00 |
| | 1e+06 | 0.08±0.00 | 0.08±0.00 | 0.08±0.00 | 0.08±0.00 | 0.08±0.00 | 0.08±0.00 | 0.08±0.00 | 0.08±0.00 | 0.08±0.00 |
| | 3e+06 | 0.08±0.00 | 0.08±0.00 | 0.08±0.00 | 0.08±0.00 | 0.08±0.00 | 0.08±0.00 | 0.09±0.00 | 0.09±0.00 | 0.09±0.00 |
| Grid | 1e+04 | 0.72±0.01 | 0.56±0.00 | 0.44±0.00 | 0.36±0.00 | 0.29±0.00 | 0.23±0.00 | 0.17±0.00 | 0.13±0.00 | 0.10±0.00 |
| | 3e+04 | 0.79±0.00 | 0.66±0.00 | 0.57±0.00 | 0.48±0.00 | 0.41±0.00 | 0.36±0.00 | 0.31±0.00 | 0.27±0.00 | 0.23±0.00 |
| | 1e+05 | 0.85±0.00 | 0.74±0.00 | 0.66±0.00 | 0.60±0.00 | 0.54±0.00 | 0.48±0.00 | 0.43±0.00 | 0.40±0.00 | 0.36±0.00 |
| | 3e+05 | 0.90±0.00 | 0.81±0.00 | 0.75±0.00 | 0.69±0.00 | 0.65±0.00 | 0.61±0.00 | 0.57±0.00 | 0.53±0.00 | 0.49±0.00 |
| | 1e+06 | 1.00±0.00 | 1.00±0.00 | 1.00±0.00 | 1.00±0.00 | 0.99±0.00 | 0.99±0.00 | 0.99±0.00 | 0.98±0.00 | 0.98±0.00 |
| | 3e+06 | 1.00±0.00 | 1.00±0.00 | 1.00±0.00 | 1.00±0.00 | 1.00±0.00 | 1.00±0.00 | 1.00±0.00 | 1.00±0.00 | 1.00±0.00 |

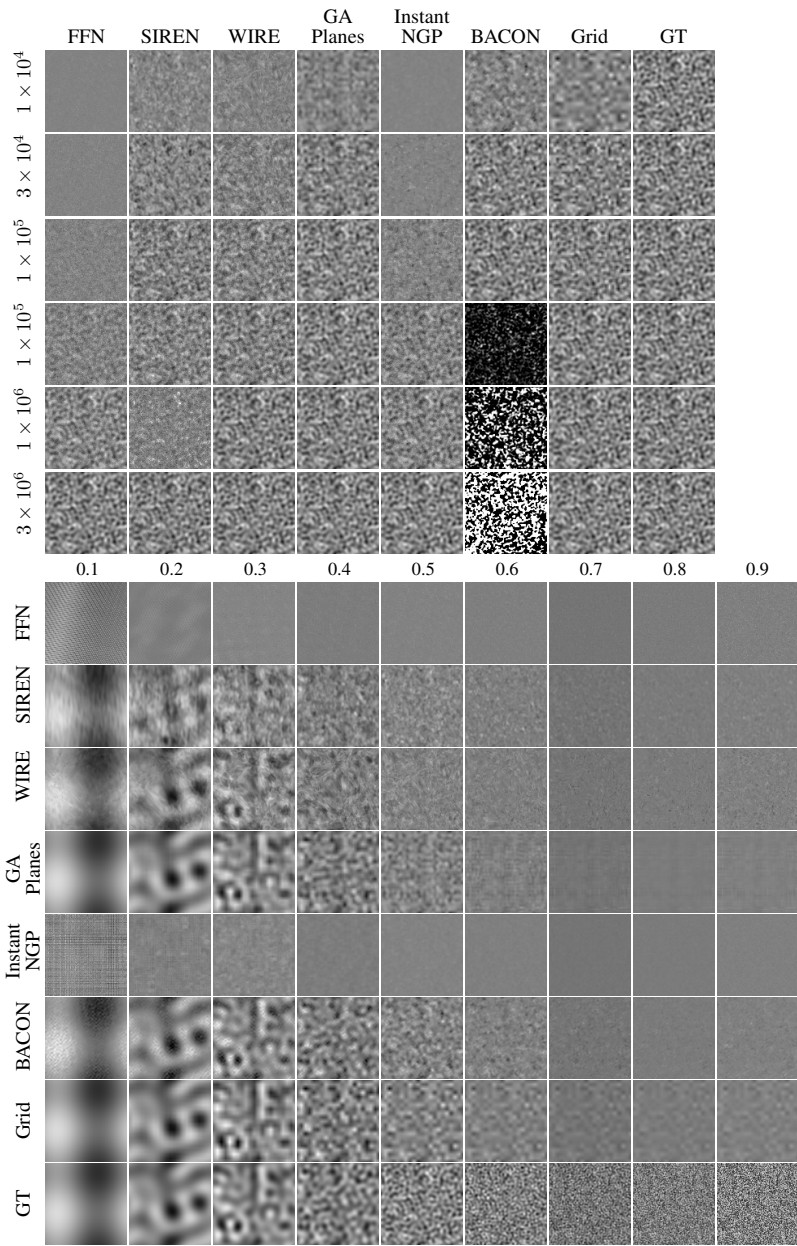

Figure 12: **3D Bandlimited overfitting with bandwidth = 0.5 (top) and model size = $1 \times 10^4$ (bottom).** FFN, Instant-NGP, and BACON show similar behavior as on the 3D Spheres, while other models exhibit similar behavior as with 2D Bandlimited images. Detailed quantitative results are in Table 19.

Table 19: Comparison of model performance across bandwidths on 3D Bandlimited signal (PSNR).

| Model | Size | 0.1 | 0.2 | 0.3 | 0.4 | 0.5 | 0.6 | 0.7 | 0.8 | 0.9 |
|---|---|---|---|---|---|---|---|---|---|---|
| FFN | 1e+04 | 14.43±0.36 | 17.17±0.54 | 18.25±0.31 | 18.86±0.18 | 19.35±0.30 | 19.48±0.26 | 19.24±0.36 | 17.87±0.25 | 13.71±0.06 |
| | 3e+04 | 18.17±2.05 | 17.52±0.69 | 18.38±0.32 | 18.99±0.20 | 19.48±0.30 | 19.60±0.26 | 19.36±0.36 | 18.00±0.26 | 13.86±0.05 |
| | 1e+05 | 23.48±1.73 | 19.27±0.60 | 19.10±0.30 | 19.61±0.17 | 20.07±0.27 | 20.19±0.25 | 19.95±0.38 | 18.60±0.24 | 14.51±0.06 |
| | 3e+05 | 27.29±1.56 | 21.74±0.54 | 21.11±0.28 | 21.56±0.16 | 21.97±0.28 | 22.08±0.24 | 21.87±0.39 | 20.59±0.23 | 16.70±0.05 |
| | 1e+06 | 31.04±2.59 | 27.77±0.47 | 27.44±0.43 | 27.74±0.27 | 27.90±0.35 | 27.92±0.34 | 27.94±0.53 | 26.81±0.31 | 23.25±0.17 |
| | 3e+06 | 45.64±1.64 | 42.28±1.80 | 42.37±2.21 | 42.76±2.34 | 43.04±1.35 | 44.14±1.90 | 44.24±1.38 | 41.69±1.67 | 38.87±1.70 |
| SIREN | 1e+04 | 29.07±3.49 | 20.64±1.51 | 20.76±0.35 | 20.50±0.37 | 20.16±0.31 | 19.79±0.23 | 19.31±0.35 | 17.86±0.25 | 13.69±0.06 |
| | 3e+04 | 31.30±2.39 | 25.36±1.23 | 24.76±1.18 | 23.49±0.66 | 22.02±0.46 | 20.62±0.14 | 19.62±0.36 | 17.96±0.25 | 13.76±0.06 |
| | 1e+05 | 27.10±2.81 | 27.66±1.07 | 26.59±1.80 | 26.06±2.04 | 25.01±0.46 | 20.10±3.63 | 20.70±0.37 | 18.32±0.25 | 14.04±0.07 |
| | 3e+05 | 19.60±4.18 | 21.99±5.50 | 23.48±2.94 | 18.91±2.79 | 23.09±5.29 | 17.96±4.38 | 19.20±1.69 | 17.51±1.05 | 14.91±0.51 |
| | 1e+06 | 11.76±3.50 | 13.07±4.41 | 16.16±3.66 | 15.82±5.18 | 13.64±3.91 | 14.07±5.02 | 15.48±4.27 | 15.97±6.05 | 15.72±3.10 |
| | 3e+06 | 16.99±8.20 | 27.96±6.69 | 21.25±8.81 | 26.10±17.85 | 19.84±11.55 | 20.11±9.76 | 21.35±5.78 | 22.90±10.93 | 18.19±9.03 |
| WIRE | 1e+04 | 30.60±1.03 | 26.92±1.47 | 22.92±0.61 | 20.84±0.33 | 20.12±0.34 | 19.77±0.22 | 19.32±0.36 | 17.88±0.25 | 13.73±0.06 |
| | 3e+04 | 32.82±1.36 | 31.08±1.12 | 27.80±0.49 | 24.35±0.46 | 21.81±0.41 | 20.60±0.17 | 19.77±0.35 | 18.16±0.25 | 14.00±0.06 |
| | 1e+05 | 33.17±0.81 | 32.70±0.89 | 30.86±0.50 | 28.08±0.51 | 24.86±0.55 | 22.60±0.17 | 21.09±0.34 | 19.21±0.23 | 15.15±0.04 |
| | 3e+05 | 31.04±0.42 | 31.02±0.27 | 30.00±0.37 | 28.85±0.40 | 27.45±0.36 | 25.95±0.14 | 24.41±0.42 | 22.46±0.12 | 18.49±0.04 |
| | 1e+06 | 37.73±0.15 | 38.11±0.21 | 37.88±0.21 | 37.54±0.22 | 36.79±0.17 | 35.56±0.11 | 34.07±0.29 | 31.96±0.27 | 27.96±0.10 |
| | 3e+06 | 48.82±0.18 | 49.04±0.39 | 48.90±0.36 | 48.95±0.32 | 48.41±0.39 | 47.48±0.23 | 45.18±0.45 | 39.64±0.53 | 30.13±0.14 |
| GA-Planes | 1e+04 | 60.93±1.64 | 48.29±1.84 | 34.71±1.40 | 25.54±0.58 | 21.30±0.27 | 20.06±0.41 | 19.40±0.33 | 17.91±0.23 | 13.72±0.06 |
| | 3e+04 | 64.84±1.40 | 54.33±1.88 | 48.15±2.34 | 38.04±1.06 | 26.27±0.38 | 21.43±0.50 | 19.90±0.33 | 18.07±0.21 | 13.87±0.10 |
| | 1e+05 | 69.25±0.71 | 60.70±1.74 | 56.05±0.79 | 42.46±0.92 | 32.72±1.47 | 24.81±1.07 | 21.29±0.28 | 18.58±0.22 | 14.43±0.28 |
| | 3e+05 | 70.20±0.49 | 60.67±1.33 | 57.12±0.65 | 45.84±1.33 | 39.14±1.86 | 30.90±1.34 | 23.97±0.51 | 19.74±0.38 | 16.16±0.26 |
| | 1e+06 | 66.04±1.38 | 61.22±0.93 | 56.97±0.43 | 46.29±0.41 | 39.58±1.11 | 30.86±1.01 | 24.36±1.12 | 19.63±0.29 | 16.83±1.47 |
| | 3e+06 | 70.06±2.09 | 66.13±1.95 | 66.32±0.49 | 60.80±0.34 | 52.68±1.11 | 46.37±1.04 | 39.16±1.46 | 34.63±4.22 | 34.55±1.70 |
| Instant-NGP | 1e+04 | 14.26±0.40 | 17.18±0.54 | 18.31±0.29 | 18.88±0.17 | 19.36±0.30 | 19.47±0.27 | 19.22±0.35 | 17.84±0.25 | 13.68±0.07 |
| | 3e+04 | 15.26±0.23 | 17.79±0.54 | 18.78±0.30 | 19.27±0.17 | 19.60±0.28 | 19.60±0.27 | 19.29±0.37 | 17.91±0.28 | 13.81±0.06 |
| | 1e+05 | 16.91±0.61 | 19.42±0.55 | 20.35±0.28 | 20.62±0.15 | 20.70±0.29 | 20.40±0.30 | 19.92±0.40 | 18.40±0.27 | 14.58±0.15 |
| | 3e+05 | 21.01±1.37 | 23.30±0.68 | 23.94±0.28 | 24.37±0.16 | 24.09±0.25 | 23.20±0.31 | 22.44±0.49 | 20.86±0.37 | 17.09±0.22 |
| | 1e+06 | 24.12±0.80 | 26.16±0.65 | 26.99±0.43 | 27.25±0.37 | 27.00±0.24 | 26.67±0.56 | 25.83±0.49 | 23.90±0.61 | 20.58±0.33 |
| | 3e+06 | 72.69±8.18 | 70.06±7.32 | 75.34±9.42 | 73.68±13.05 | 80.09±6.94 | 77.25±4.82 | 66.84±9.87 | 77.76±9.41 | 88.87±27.66 |
| BACON | 1e+04 | 31.33±0.37 | 34.33±0.58 | 31.59±0.41 | 26.50±0.16 | 21.90±0.30 | 20.13±0.24 | 19.37±0.34 | 17.87±0.25 | 13.70±0.06 |
| | 3e+04 | 37.93±4.03 | 36.79±0.49 | 37.49±1.04 | 34.73±1.44 | 28.19±0.21 | 22.03±0.23 | 20.07±0.35 | 18.07±0.25 | 13.84±0.06 |
| | 1e+05 | 38.90±1.56 | 41.91±0.82 | 41.23±1.26 | 39.80±0.94 | 36.28±0.44 | 26.80±0.12 | 22.79±0.28 | 18.88±0.23 | 14.32±0.07 |
| | 3e+05 | 14.34±15.20 | 7.37±0.45 | 7.36±0.60 | 7.54±0.43 | 7.40±0.42 | 7.51±0.28 | 10.08±4.54 | 9.96±5.41 | 6.87±0.10 |
| | 1e+06 | 6.60±0.13 | 7.10±0.23 | 7.12±0.26 | 7.25±0.13 | 7.18±0.18 | 7.26±0.07 | 7.38±0.21 | 7.12±0.10 | 6.62±0.09 |
| | 3e+06 | 5.37±0.08 | 5.83±0.06 | 5.89±0.04 | 5.94±0.02 | 5.95±0.02 | 5.97±0.02 | 5.96±0.02 | 5.88±0.02 | 5.46±0.02 |
| Grid | 1e+04 | 61.83±0.55 | 47.48±0.61 | 35.09±0.34 | 25.62±0.20 | 21.11±0.30 | 19.98±0.26 | 19.39±0.35 | 17.90±0.25 | 13.71±0.06 |
| | 3e+04 | 69.04±0.50 | 55.04±0.62 | 43.65±0.32 | 34.39±0.15 | 25.78±0.32 | 21.40±0.26 | 19.82±0.35 | 18.06±0.25 | 13.80±0.06 |
| | 1e+05 | 76.45±0.50 | 62.62±0.61 | 51.82±0.31 | 43.41±0.17 | 34.67±0.31 | 26.52±0.26 | 21.55±0.35 | 18.62±0.25 | 14.11±0.06 |
| | 3e+05 | 85.84±0.52 | 71.93±0.62 | 60.91±0.32 | 52.33±0.17 | 43.70±0.33 | 35.04±0.25 | 26.79±0.36 | 20.56±0.24 | 15.15±0.06 |
| | 1e+06 | 138.54±0.27 | 139.36±0.52 | 137.70±0.52 | 132.37±0.59 | 126.20±0.57 | 100.44±0.78 | 70.52±0.59 | 34.49±0.23 | 24.57±0.07 |
| | 3e+06 | 141.99±0.35 | 144.01±0.76 | 144.28±1.06 | 144.86±0.68 | 144.67±0.68 | 144.95±0.31 | 145.37±1.02 | 144.11±0.40 | 142.26±0.19 |

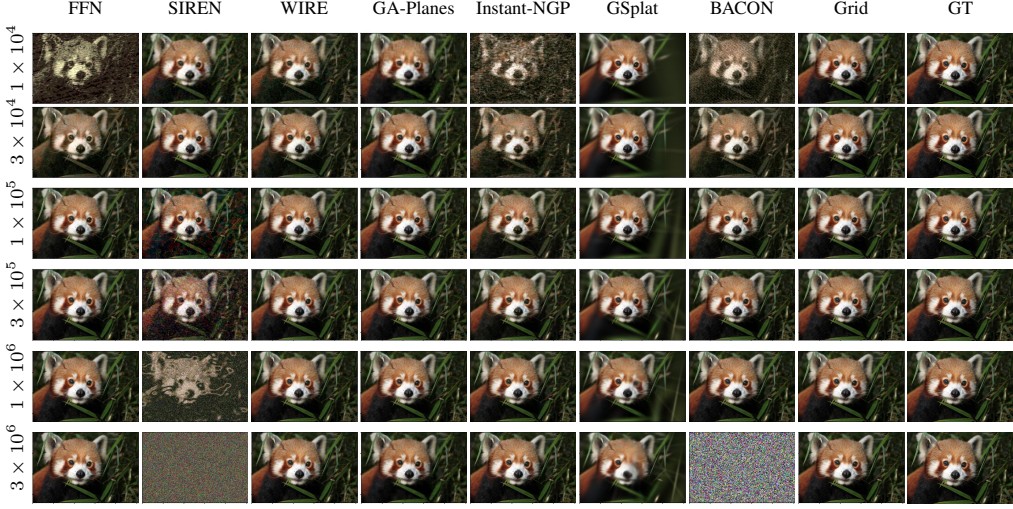

Figure 13: **DIV2K Overfitting.** FFN and Instant-NGP struggle to overfit colors with small model sizes. Similar to synthetic signals, SIREN and BACON exhibit noisy artifacts and BACON exhibits unstable performance in the over-parameterized regime. Detailed quantitative results are in Tables 20 and 21.

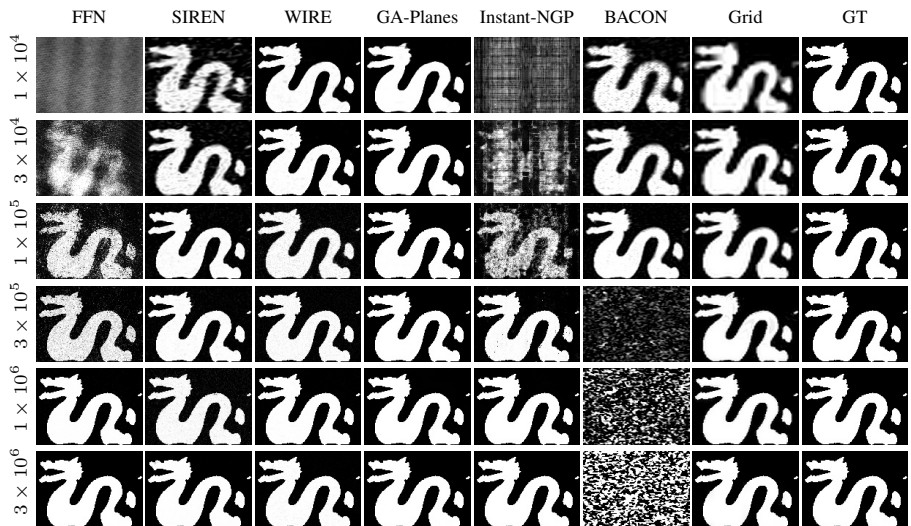

Figure 14: **3D Dragon Occupancy Overfitting.** All models except BACON effectively learn the signal when model size matches or exceeds the inherent signal size of $1 \times 10^6$. At small model sizes (high compression), WIRE and GA-Planes produce much sharper representations than the Grid, which is blurry when underparameterized. Detailed quantitative results are in Table 20.

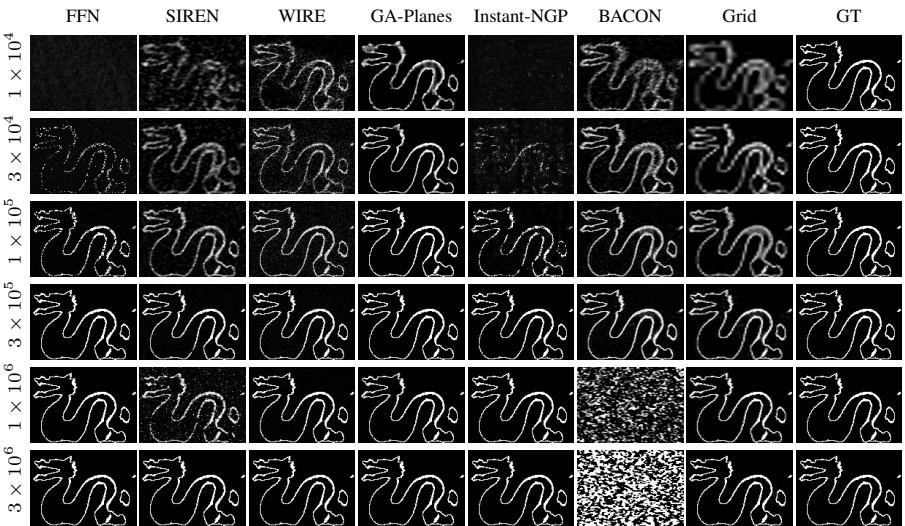

Figure 15: **3D Dragon Surface Overfitting.** Results mirror those for the solid 3D Dragon occupancy overfitting task. At small model sizes, GA-Planes produces the sharpest representation. Detailed quantitative results are in Table 20.

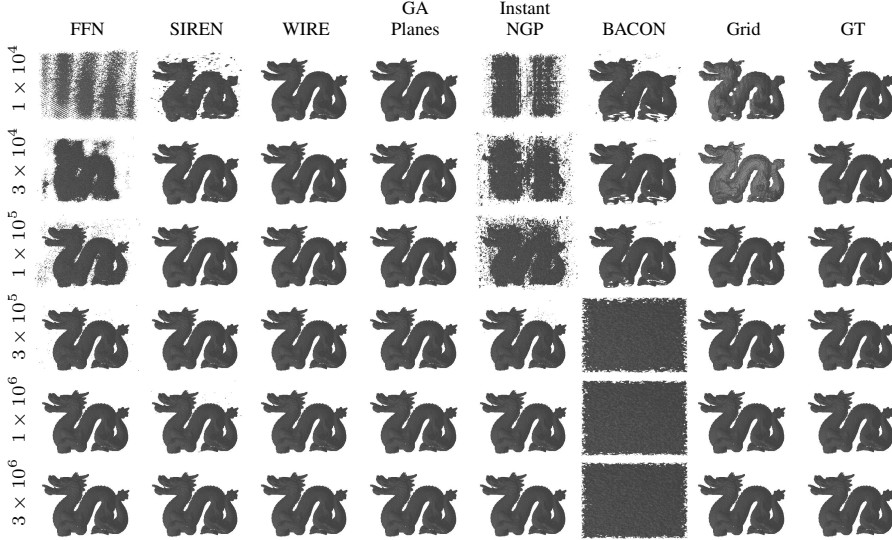

Figure 16: **3D Dragon Occupancy Overfitting Render.** Detailed quantitative results are in Table 20.

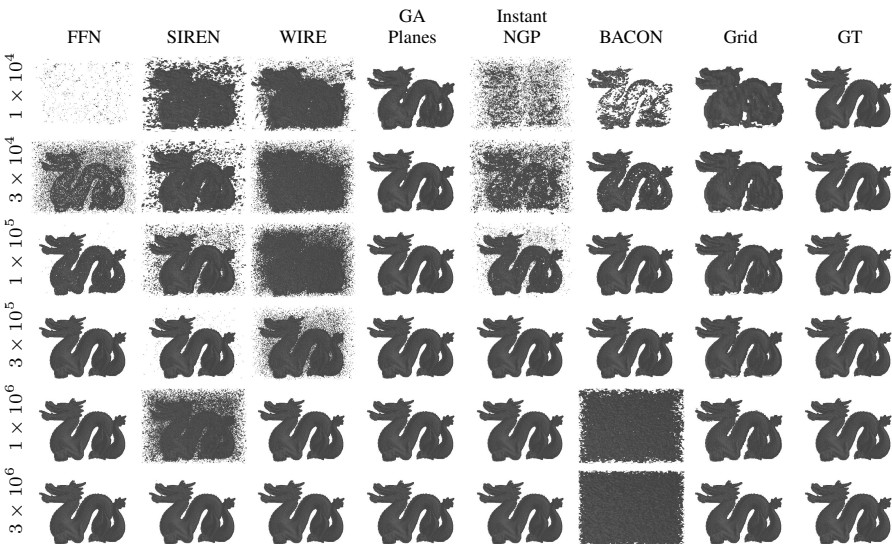

Figure 17: **3D Dragon Surface Overfitting Render.** For most models, results parallel results at dragon occupancy fitting. However, WIRE and BACON notably exhibit different performance in these two tasks, both struggling to fit the dragon surface at small model sizes. Detailed quantitative results are in Table 20.

Table 20: Comparison of PSNR across model sizes on overfitting tasks.

| Task | Model | 1e+04 | 3e+04 | 1e+05 | 3e+05 | 1e+06 | 3e+06 |
|---|---|---|---|---|---|---|---|
| DIV2K | FFN | 15.99±1.29 | 22.00±2.26 | 28.88±3.28 | 35.95±3.34 | 44.08±3.67 | 48.26±2.86 |
| | SIREN | 21.98±3.26 | 23.93±3.27 | 22.05±2.90 | 20.35±2.42 | 15.87±4.03 | 11.22±3.20 |
| | WIRE | 21.58±2.75 | 24.08±2.90 | 27.67±2.93 | 31.62±2.92 | 36.09±2.26 | 39.06±1.96 |
| | GA-Planes | 22.05±3.53 | 24.96±3.88 | 29.81±4.13 | 35.20±3.38 | 38.75±2.23 | 29.32±2.41 |
| | Instant-NGP | 15.71±1.24 | 20.18±2.40 | 24.73±3.43 | 32.42±3.44 | 39.21±3.56 | 63.21±4.25 |
| | GSplat | 20.97±3.56 | 21.46±3.55 | 21.90±3.46 | 22.07±3.41 | 21.81±3.19 | 21.62±3.24 |
| | BACON | 17.57±1.64 | 21.65±2.54 | 27.19±3.27 | 32.17±3.63 | 34.59±4.14 | 10.35±11.38 |
| | Grid | 24.06±3.97 | 26.95±4.21 | 33.01±4.65 | 148.70±0.85 | 154.35±0.84 | 156.03±0.69 |
| 3D Dragon Occupancy | FFN | 8.56 | 10.86 | 15.55 | 19.79 | 42.43 | 62.75 |
| | SIREN | 16.72 | 23.08 | 26.95 | 30.24 | 25.89 | 45.08 |
| | WIRE | 23.54 | 28.01 | 26.41 | 24.56 | 34.53 | 46.17 |
| | GA-Planes | 24.23 | 30.40 | 41.45 | 45.90 | 48.43 | 47.42 |
| | Instant-NGP | 8.91 | 9.64 | 12.11 | 25.22 | 40.71 | 76.17 |
| | BACON | 18.28 | 20.39 | 22.63 | 7.63 | 5.64 | 3.13 |
| | Grid | 17.61 | 19.40 | 21.36 | 23.76 | 28.47 | 42.63 |
| 3D Dragon Surface | FFN | 13.42 | 14.67 | 19.27 | 35.04 | 73.36 | 47.47 |
| | SIREN | 14.60 | 15.92 | 18.25 | 28.28 | 16.27 | 38.71 |
| | WIRE | 15.68 | 16.42 | 18.81 | 23.61 | 33.29 | 37.08 |
| | GA-Planes | 17.99 | 22.62 | 38.86 | 40.86 | 39.44 | 43.11 |
| | Instant-NGP | 13.62 | 14.44 | 18.81 | 45.37 | 78.03 | 77.92 |
| | BACON | 15.83 | 17.77 | 20.48 | 22.66 | 7.60 | 3.20 |
| | Grid | 14.87 | 15.83 | 17.51 | 20.10 | 25.35 | 40.33 |

Table 21: Comparison of metrics across model sizes on overfitting task (DIV2K).

| Task | Metric | Model | 1e+04 | 3e+04 | 1e+05 | 3e+05 | 1e+06 | 3e+06 |
|---|---|---|---|---|---|---|---|---|
| DIV2K | SSIM | FFN | 0.251±0.048 | 0.579±0.086 | 0.839±0.030 | 0.954±0.014 | 0.991±0.005 | 0.997±0.001 |
| | | SIREN | 0.508±0.108 | 0.637±0.070 | 0.625±0.124 | 0.549±0.167 | 0.281±0.243 | 0.104±0.149 |
| | | WIRE | 0.485±0.056 | 0.636±0.042 | 0.798±0.035 | 0.906±0.020 | 0.959±0.013 | 0.975±0.011 |
| | | GA-Planes | 0.545±0.134 | 0.708±0.101 | 0.876±0.050 | 0.958±0.016 | 0.982±0.004 | 0.883±0.035 |
| | | Instant-NGP | 0.200±0.041 | 0.430±0.050 | 0.691±0.057 | 0.924±0.024 | 0.979±0.009 | 1.000±0.000 |
| | | GSplat | 0.489±0.165 | 0.525±0.154 | 0.553±0.138 | 0.561±0.136 | 0.549±0.132 | 0.538±0.137 |
| | | BACON | 0.253±0.037 | 0.507±0.057 | 0.793±0.030 | 0.935±0.014 | 0.967±0.014 | 0.195±0.384 |
| | | Grid | 0.692±0.122 | 0.845±0.074 | 0.962±0.022 | 1.000±0.000 | 1.000±0.000 | 1.000±0.000 |
| | LPIPS | FFN | 0.71±0.02 | 0.56±0.04 | 0.26±0.04 | 0.08±0.04 | 0.01±0.01 | 0.00±0.00 |
| | | SIREN | 0.58±0.05 | 0.47±0.05 | 0.48±0.11 | 0.50±0.13 | 0.65±0.16 | 0.76±0.09 |
| | | WIRE | 0.59±0.02 | 0.50±0.03 | 0.29±0.05 | 0.15±0.06 | 0.08±0.06 | 0.06±0.05 |
| | | GA-Planes | 0.56±0.06 | 0.44±0.07 | 0.22±0.05 | 0.07±0.03 | 0.03±0.01 | 0.22±0.07 |
| | | Instant-NGP | 0.72±0.03 | 0.63±0.04 | 0.45±0.06 | 0.14±0.05 | 0.04±0.02 | 0.00±0.00 |
| | | GSplat | 0.59±0.08 | 0.56±0.08 | 0.54±0.07 | 0.53±0.07 | 0.53±0.06 | 0.55±0.06 |
| | | BACON | 0.68±0.02 | 0.59±0.02 | 0.33±0.04 | 0.12±0.04 | 0.07±0.03 | 0.67±0.31 |
| | | Grid | 0.38±0.07 | 0.23±0.06 | 0.06±0.02 | 0.00±0.00 | 0.00±0.00 | 0.00±0.00 |

## 5.7 Inverse Problems: Full Results

**Extended Results for Generalization and Inverse Problems.** We present detailed visualizations of each model's predictions on the computed tomography, image denoising ($\epsilon = 0.05, 0.1$), and super-resolution tasks for both images and volumetric data. See Figure 18 (CT reconstruction), Figure 19 (PSNR curves for DIV2K image denoising), Figure 20 (DIV2K denoising, $\epsilon = 0.05$), Figure 21 (DIV2K denoising, $\epsilon = 0.1$), Figure 22 (DIV2K super-resolution), Figure 23 (3D Dragon occupancy super-resolution), and Figures 24 and 26 (3D Dragon surface super-resolution).

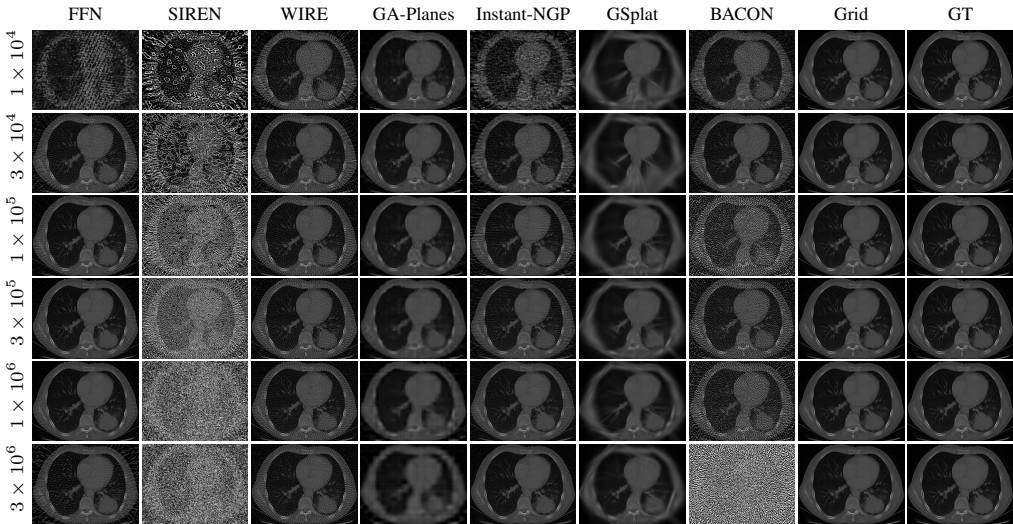

Figure 18: **CT Reconstruction.** All models except SIREN and BACON successfully reconstruct the CT image at medium to large model sizes. However, the qualitative performance of GA-Planes degrades in the over-parameterized regime, perhaps due to lack of explicit regularization for this underdetermined inverse problem. No model outperforms the TV-regularized Grid at any model size in the compressive regime. Detailed quantitative results are in Tables 22 to 24.

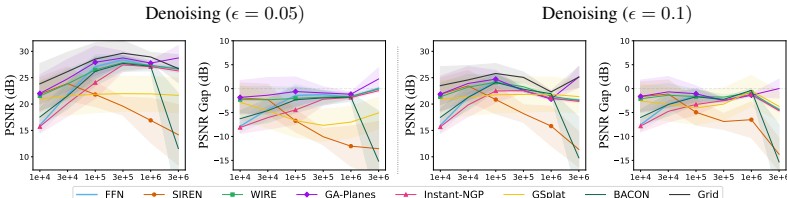

Figure 19: **Denoising Evaluation.** Denoising performance of various models on the DIV2K dataset, evaluated using 10 images. Grid with TV regularization consistently outperforms other methods at both noise levels.

Table 22: Comparison of model performance across model sizes on inverse problems (PSNR).

| Task | Model | 1e+04 | 3e+04 | 1e+05 | 3e+05 | 1e+06 | 3e+06 |
|---|---|---|---|---|---|---|---|
| CT | FFN | 12.29±2.64 | 17.16±2.10 | 23.17±1.55 | 26.81±0.94 | 29.75±0.92 | 28.50±2.14 |
| | SIREN | 12.20±0.63 | 10.96±3.09 | 8.09±0.64 | 8.52±0.72 | 7.26±0.50 | 6.82±0.95 |
| | WIRE | 18.75±1.00 | 20.69±0.76 | 21.08±0.46 | 21.52±0.32 | 21.86±0.26 | 22.27±0.14 |
| | GA-Planes | 31.49±2.64 | 33.76±3.02 | 33.15±3.16 | 32.41±2.88 | 31.91±3.49 | 29.89±2.37 |
| | Instant-NGP | 12.34±2.41 | 19.03±2.73 | 18.94±1.74 | 16.64±2.47 | 18.32±2.32 | 22.01±2.02 |
| | GSplat | 27.59±2.74 | 27.57±2.53 | 28.36±2.89 | 27.92±2.87 | 28.20±3.38 | 27.00±3.28 |
| | BACON | 16.37±1.43 | 16.30±2.30 | 12.07±0.72 | 13.27±2.56 | 9.87±2.61 | 5.11±0.25 |
| | Grid | 38.31±4.91 | 40.35±4.48 | 41.13±4.39 | 40.71±4.76 | 37.76±2.46 | 31.82±1.62 |
| DIV2K Denoising $\epsilon = 0.05$ | FFN | 15.96±1.29 | 21.83±2.17 | 27.11±2.28 | 28.39±0.81 | 27.16±0.20 | 26.81±0.21 |
| | SIREN | 21.89±3.14 | 23.89±3.17 | 21.81±2.31 | 19.59±1.71 | 16.91±4.21 | 14.17±5.72 |
| | WIRE | 21.54±2.72 | 23.82±2.77 | 26.49±2.29 | 27.86±1.38 | 27.35±0.43 | 26.63±0.22 |
| | GA-Planes | 22.01±3.51 | 24.78±3.68 | 27.93±3.05 | 28.75±1.27 | 27.77±0.45 | 28.74±2.33 |
| | Instant-NGP | 15.72±1.27 | 20.10±2.41 | 24.07±3.00 | 27.43±1.27 | 27.10±0.24 | 26.29±0.29 |
| | GSplat | 20.97±3.50 | 21.54±3.64 | 21.86±3.48 | 21.97±3.34 | 21.93±3.30 | 21.65±3.29 |
| | BACON | 17.54±1.64 | 21.55±2.51 | 26.21±2.73 | 27.71±1.49 | 27.16±0.87 | 11.57±10.50 |
| | Grid | 23.84±3.84 | 26.16±3.68 | 28.53±3.07 | 29.64±2.37 | 28.93±0.87 | 26.73±2.64 |
| DIV2K Denoising $\epsilon = 0.1$ | FFN | 15.94±1.29 | 21.34±1.93 | 24.35±1.38 | 22.56±0.26 | 21.02±0.21 | 20.81±0.29 |
| | SIREN | 21.83±3.12 | 23.45±2.91 | 20.85±2.70 | 18.19±1.68 | 15.83±3.75 | 11.40±3.35 |
| | WIRE | 21.38±2.64 | 23.20±2.44 | 24.14±1.42 | 23.29±0.50 | 21.48±0.19 | 20.67±0.24 |
| | GA-Planes | 21.87±3.42 | 23.90±3.19 | 24.73±1.65 | 22.59±0.33 | 20.95±0.29 | 25.14±1.96 |
| | Instant-NGP | 15.66±1.24 | 19.83±2.22 | 22.49±2.19 | 22.56±0.38 | 21.10±0.22 | 20.50±0.29 |
| | GSplat | 20.92±3.53 | 21.32±3.50 | 21.76±3.44 | 21.83±3.34 | 21.84±3.24 | 21.42±3.12 |
| | BACON | 17.47±1.60 | 21.25±2.38 | 24.10±1.64 | 22.81±0.42 | 21.96±0.42 | 9.80±7.80 |
| | Grid | 23.50±3.61 | 24.58±2.71 | 25.80±1.88 | 25.06±1.02 | 22.32±0.25 | 25.12±1.96 |
| DIV2K Super Resolution | FFN | 15.16±1.64 | 19.51±2.97 | 21.98±3.94 | 22.51±4.09 | 22.72±4.10 | 22.78±4.11 |
| | SIREN | 19.74±3.52 | 20.78±3.69 | 18.75±2.45 | 16.97±1.95 | 13.92±2.26 | 9.95±2.12 |
| | WIRE | 19.45±3.19 | 20.76±3.48 | 21.89±3.81 | 22.41±3.99 | 22.54±4.01 | 22.61±4.01 |
| | GA-Planes | 19.71±3.69 | 20.83±3.92 | 21.77±4.10 | 22.24±4.13 | 22.34±4.06 | 21.92±3.57 |
| | Instant-NGP | 14.39±1.46 | 17.77±2.85 | 17.71±3.32 | 14.03±2.91 | 12.91±2.37 | 12.29±1.24 |
| | GSplat | 19.07±3.55 | 19.40±3.59 | 19.69±3.58 | 19.84±3.62 | 19.66±3.42 | 19.51±3.36 |
| | BACON | 16.50±2.07 | 19.38±3.07 | 21.66±3.79 | 22.34±4.01 | 22.43±4.05 | 7.68±6.29 |
| | Grid | 20.81±3.97 | 21.86±4.12 | 22.35±3.98 | 22.38±3.91 | 22.31±4.03 | 21.06±3.70 |

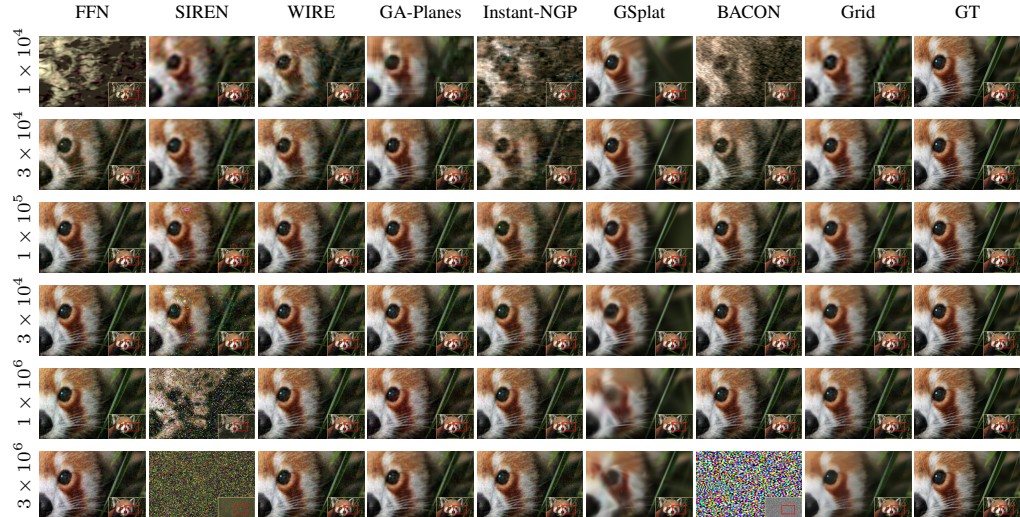

Figure 20: **DIV2K Denoising, $\epsilon = 0.05$.** WIRE and GA-Planes produce sharper reconstructions but exhibit characteristic noisy and grid-like artifacts, respectively, at the smallest model size. SIREN and Grid produce reasonable but characteristically blurry results in the underparameterized regime, and SIREN and BACON exhibit unstable performance when overparameterized. Instant-NGP and FFN also perform well at denoising, except under the strongest compression. GSplat produces sharp details in some regions but misses details in others. Detailed quantitative results are in Tables 22 to 24.

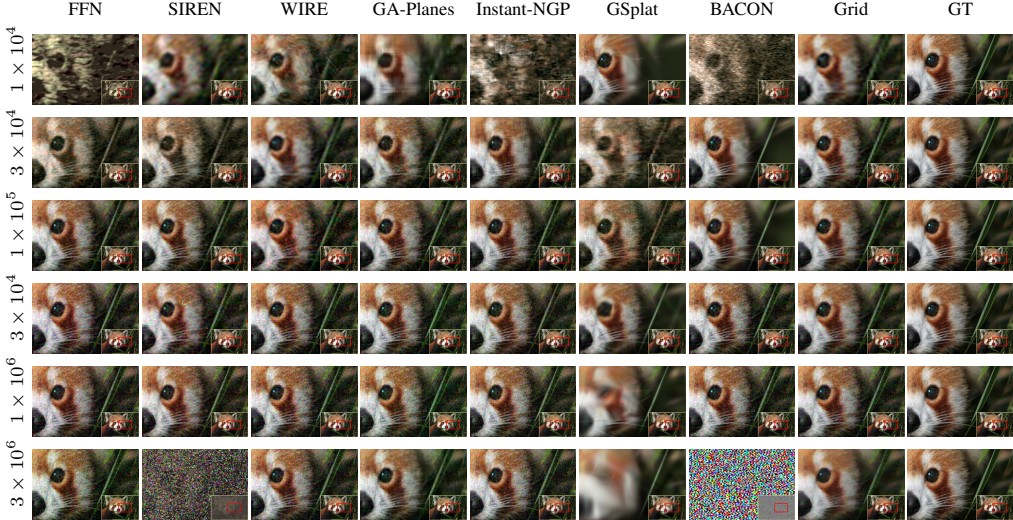

Figure 21: **DIV2K Denoising, $\epsilon = 0.1$.** The overall trends remain similar to the $\epsilon = 0.05$ case, but due to the increased noise level, all models struggle to effectively remove noise in the reconstructed outputs. Detailed quantitative results are in Tables 22 to 24.

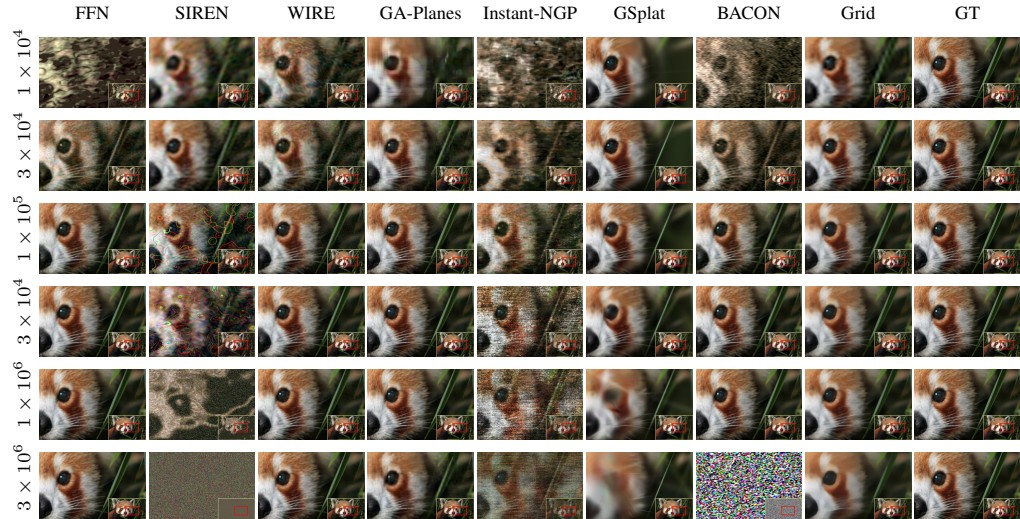

Figure 22: **DIV2K Super-Resolution.** All methods except Instant-NGP perform reasonably well at super-resolution, though only SIREN, WIRE, GA-Planes, GSplat, and Grid perform well under extreme compression. Detailed quantitative results are in Tables 22 to 24.

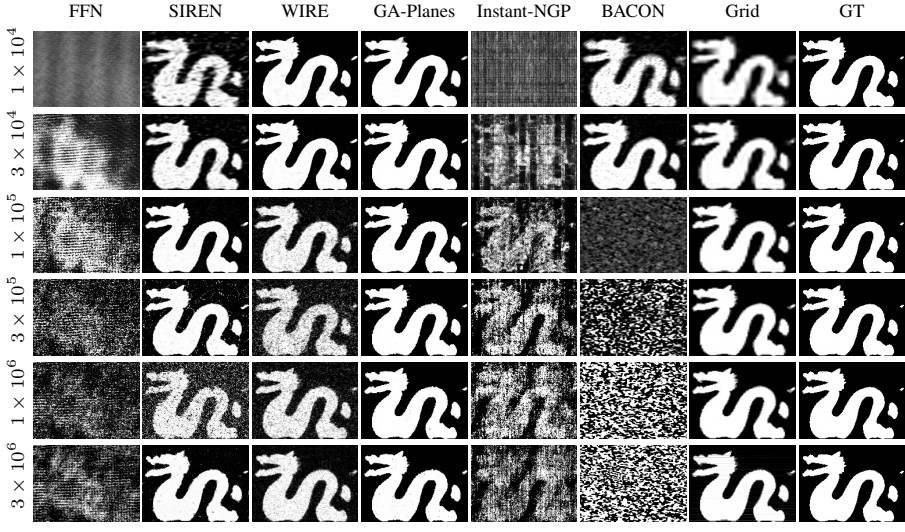

Figure 23: **3D Dragon Occupancy Super-Resolution.** SIREN, WIRE, GA-Planes, and Grid all produce reasonable results, with GA-Planes' the most consistent across model sizes. BACON struggles to represent the shape at large model sizes. Detailed quantitative results are in Table 25.

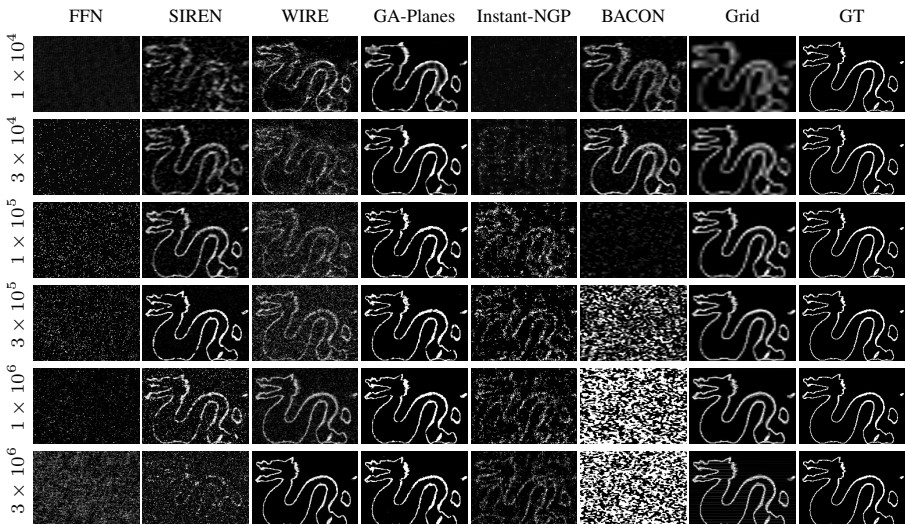

Figure 24: **3D Dragon Surface Super-Resolution.** For 3D super-resolution, only SIREN, WIRE, GA-Planes, BACON, and Grid successfully capture surface details at some model sizes. GA-Planes consistently achieves high-quality results across model sizes, while WIRE and Grid struggle in under-parameterized conditions and SIREN and BACON show nonmonotonic performance with model size. Detailed quantitative results are in Table 25.

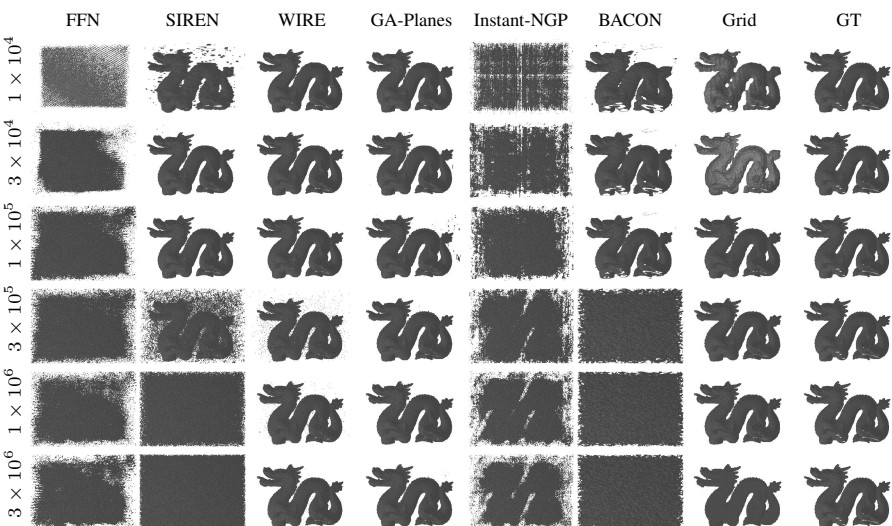

Figure 25: **3D Dragon Occupancy Super Resolution Render.** Detailed quantitative results are in Table 25.

Table 23: Comparison of model performance across model sizes on inverse problems (SSIM).

| Task | Model | 1e+04 | 3e+04 | 1e+05 | 3e+05 | 1e+06 | 3e+06 |
|------|-------|-------|-------|-------|-------|-------|-------|
| CT | FFN | 0.102±0.04 | 0.118±0.06 | 0.288±0.07 | 0.444±0.06 | 0.627±0.04 | 0.615±0.10 |
| | SIREN | 0.110±0.02 | 0.076±0.05 | 0.025±0.01 | 0.026±0.01 | 0.017±0.01 | 0.013±0.01 |
| | WIRE | 0.196±0.02 | 0.227±0.03 | 0.202±0.02 | 0.206±0.02 | 0.212±0.02 | 0.223±0.02 |
| | GA-Planes | 0.831±0.08 | 0.877±0.07 | 0.872±0.07 | 0.868±0.08 | 0.848±0.10 | 0.777±0.08 |
| | Instant-NGP | 0.062±0.03 | 0.193±0.09 | 0.163±0.05 | 0.100±0.06 | 0.131±0.07 | 0.239±0.09 |
| | GSplat | 0.803±0.08 | 0.802±0.08 | 0.814±0.08 | 0.818±0.08 | 0.816±0.09 | 0.809±0.09 |
| | BACON | 0.081±0.03 | 0.086±0.05 | 0.035±0.01 | 0.050±0.03 | 0.026±0.02 | 0.005±0.00 |
| | Grid | 0.940±0.07 | 0.954±0.06 | 0.961±0.04 | 0.958±0.04 | 0.928±0.04 | 0.784±0.06 |
| DIV2K Denoising $\epsilon = 0.05$ | FFN | 0.249±0.05 | 0.566±0.08 | 0.764±0.04 | 0.775±0.08 | 0.722±0.11 | 0.712±0.12 |
| | SIREN | 0.507±0.10 | 0.632±0.07 | 0.602±0.12 | 0.508±0.16 | 0.299±0.25 | 0.221±0.26 |
| | WIRE | 0.482±0.06 | 0.618±0.04 | 0.737±0.05 | 0.770±0.07 | 0.735±0.10 | 0.705±0.12 |
| | GA-Planes | 0.541±0.13 | 0.698±0.09 | 0.819±0.04 | 0.800±0.06 | 0.754±0.10 | 0.843±0.02 |
| | Instant-NGP | 0.201±0.04 | 0.417±0.05 | 0.661±0.05 | 0.766±0.08 | 0.724±0.11 | 0.688±0.13 |
| | GSplat | 0.490±0.16 | 0.526±0.15 | 0.550±0.14 | 0.557±0.13 | 0.555±0.13 | 0.537±0.14 |
| | BACON | 0.251±0.04 | 0.501±0.06 | 0.738±0.03 | 0.769±0.07 | 0.735±0.09 | 0.230±0.35 |
| | Grid | 0.664±0.12 | 0.786±0.06 | 0.856±0.03 | 0.873±0.02 | 0.799±0.07 | 0.780±0.05 |
| DIV2K Denoising $\epsilon = 0.1$ | FFN | 0.246±0.05 | 0.532±0.08 | 0.635±0.09 | 0.554±0.14 | 0.495±0.15 | 0.487±0.15 |
| | SIREN | 0.501±0.10 | 0.603±0.05 | 0.537±0.11 | 0.426±0.15 | 0.218±0.12 | 0.117±0.18 |
| | WIRE | 0.470±0.05 | 0.578±0.04 | 0.619±0.08 | 0.582±0.13 | 0.511±0.15 | 0.480±0.15 |
| | GA-Planes | 0.527±0.12 | 0.640±0.06 | 0.666±0.07 | 0.553±0.14 | 0.495±0.15 | 0.691±0.08 |
| | Instant-NGP | 0.197±0.04 | 0.398±0.04 | 0.569±0.04 | 0.560±0.14 | 0.496±0.16 | 0.473±0.16 |
| | GSplat | 0.485±0.16 | 0.513±0.15 | 0.540±0.14 | 0.546±0.13 | 0.548±0.14 | 0.530±0.13 |
| | BACON | 0.247±0.04 | 0.480±0.05 | 0.625±0.08 | 0.560±0.13 | 0.526±0.14 | 0.159±0.26 |
| | Grid | 0.631±0.11 | 0.668±0.03 | 0.709±0.05 | 0.655±0.10 | 0.536±0.14 | 0.669±0.05 |
| DIV2K Super Resolution | FFN | 0.277±0.12 | 0.392±0.14 | 0.500±0.15 | 0.555±0.15 | 0.591±0.15 | 0.612±0.15 |
| | SIREN | 0.430±0.20 | 0.456±0.19 | 0.403±0.13 | 0.340±0.10 | 0.125±0.09 | 0.040±0.04 |
| | WIRE | 0.395±0.16 | 0.427±0.15 | 0.488±0.15 | 0.547±0.15 | 0.578±0.14 | 0.593±0.14 |
| | GA-Planes | 0.445±0.20 | 0.474±0.19 | 0.531±0.18 | 0.577±0.16 | 0.594±0.16 | 0.555±0.16 |
| | Instant-NGP | 0.156±0.07 | 0.282±0.13 | 0.274±0.15 | 0.129±0.09 | 0.074±0.04 | 0.060±0.02 |
| | GSplat | 0.433±0.20 | 0.444±0.20 | 0.451±0.20 | 0.453±0.20 | 0.449±0.20 | 0.445±0.20 |
| | BACON | 0.214±0.08 | 0.343±0.13 | 0.486±0.15 | 0.546±0.15 | 0.557±0.15 | 0.108±0.21 |
| | Grid | 0.476±0.20 | 0.525±0.19 | 0.554±0.18 | 0.559±0.17 | 0.568±0.17 | 0.493±0.19 |

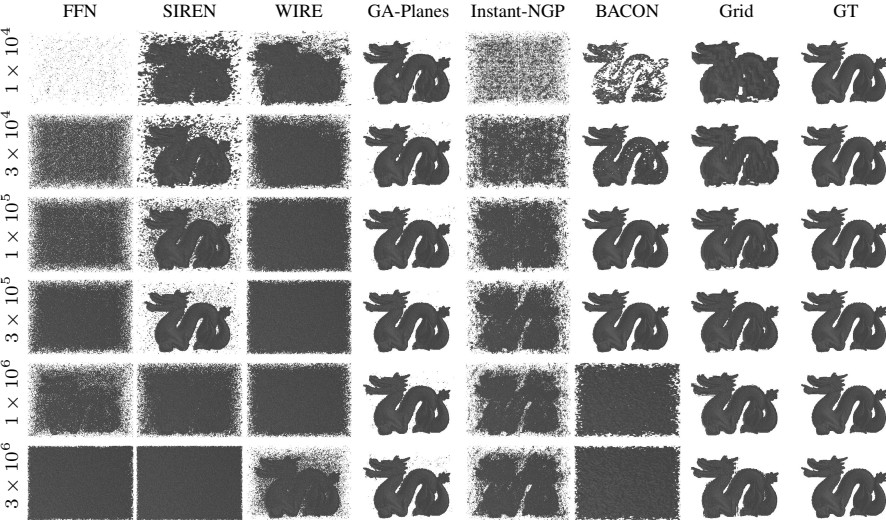

Figure 26: **3D Dragon Surface Super Resolution Render.** Detailed quantitative results are in Table 25.

Table 24: Comparison of model performance across model sizes on inverse problems (LPIPS).

| Task | Model | 1e+04 | 3e+04 | 1e+05 | 3e+05 | 1e+06 | 3e+06 |
|---|---|---|---|---|---|---|---|
| CT | FFN | 0.705±0.03 | 0.661±0.05 | 0.580±0.06 | 0.525±0.06 | 0.464±0.05 | 0.467±0.04 |
| | SIREN | 0.668±0.02 | 0.688±0.02 | 0.786±0.04 | 0.785±0.04 | 0.849±0.03 | 0.858±0.04 |
| | WIRE | 0.634±0.02 | 0.626±0.03 | 0.623±0.04 | 0.612±0.04 | 0.606±0.04 | 0.603±0.04 |
| | GA-Planes | 0.403±0.06 | 0.330±0.05 | 0.343±0.06 | 0.361±0.08 | 0.410±0.08 | 0.482±0.06 |
| | Instant-NGP | 0.859±0.05 | 0.714±0.06 | 0.710±0.06 | 0.731±0.08 | 0.697±0.08 | 0.634±0.10 |
| | GSplat | 0.397±0.10 | 0.397±0.10 | 0.381±0.10 | 0.384±0.09 | 0.384±0.09 | 0.395±0.09 |
| | BACON | 0.666±0.03 | 0.657±0.04 | 0.694±0.03 | 0.690±0.05 | 0.734±0.04 | 0.795±0.03 |
| | Grid | 0.212±0.10 | 0.180±0.09 | 0.168±0.07 | 0.168±0.06 | 0.265±0.06 | 0.411±0.07 |
| DIV2K Denoising $\epsilon = 0.05$ | FFN | 0.707±0.02 | 0.569±0.03 | 0.325±0.05 | 0.244±0.09 | 0.248±0.10 | 0.249±0.10 |
| | SIREN | 0.590±0.05 | 0.478±0.05 | 0.499±0.10 | 0.513±0.12 | 0.620±0.18 | 0.670±0.23 |
| | WIRE | 0.592±0.02 | 0.504±0.03 | 0.341±0.06 | 0.266±0.08 | 0.258±0.10 | 0.263±0.10 |
| | GA-Planes | 0.576±0.04 | 0.467±0.03 | 0.313±0.04 | 0.246±0.09 | 0.243±0.10 | 0.287±0.08 |
| | Instant-NGP | 0.723±0.03 | 0.641±0.04 | 0.477±0.06 | 0.293±0.09 | 0.255±0.10 | 0.255±0.10 |
| | GSplat | 0.593±0.08 | 0.563±0.08 | 0.542±0.07 | 0.534±0.07 | 0.533±0.06 | 0.550±0.06 |
| | BACON | 0.675±0.01 | 0.593±0.02 | 0.375±0.05 | 0.270±0.08 | 0.262±0.09 | 0.647±0.28 |
| | Grid | 0.407±0.06 | 0.310±0.03 | 0.213±0.04 | 0.195±0.06 | 0.210±0.08 | 0.258±0.05 |
| DIV2K Denoising $\epsilon = 0.1$ | FFN | 0.708±0.02 | 0.586±0.03 | 0.419±0.07 | 0.404±0.11 | 0.417±0.12 | 0.417±0.12 |
| | SIREN | 0.593±0.04 | 0.499±0.05 | 0.535±0.08 | 0.551±0.11 | 0.677±0.10 | 0.760±0.13 |
| | WIRE | 0.597±0.02 | 0.524±0.04 | 0.424±0.08 | 0.399±0.10 | 0.419±0.11 | 0.428±0.12 |
| | GA-Planes | 0.594±0.03 | 0.521±0.01 | 0.419±0.07 | 0.407±0.11 | 0.421±0.12 | 0.381±0.09 |
| | Instant-NGP | 0.727±0.03 | 0.650±0.03 | 0.539±0.06 | 0.432±0.10 | 0.421±0.12 | 0.423±0.12 |
| | GSplat | 0.597±0.08 | 0.574±0.07 | 0.553±0.06 | 0.548±0.06 | 0.543±0.06 | 0.558±0.05 |
| | BACON | 0.678±0.02 | 0.600±0.02 | 0.443±0.07 | 0.411±0.10 | 0.412±0.11 | 0.696±0.23 |
| | Grid | 0.461±0.05 | 0.420±0.05 | 0.347±0.07 | 0.347±0.09 | 0.393±0.11 | 0.354±0.07 |
| DIV2K Super Resolution | FFN | 0.772±0.04 | 0.688±0.04 | 0.586±0.05 | 0.505±0.05 | 0.450±0.05 | 0.421±0.07 |
| | SIREN | 0.783±0.06 | 0.712±0.06 | 0.677±0.06 | 0.696±0.06 | 0.761±0.07 | 0.818±0.04 |
| | WIRE | 0.717±0.04 | 0.666±0.04 | 0.601±0.04 | 0.536±0.05 | 0.487±0.06 | 0.465±0.06 |
| | GA-Planes | 0.668±0.06 | 0.622±0.06 | 0.551±0.06 | 0.484±0.06 | 0.447±0.06 | 0.536±0.06 |
| | Instant-NGP | 0.778±0.03 | 0.710±0.03 | 0.685±0.04 | 0.739±0.05 | 0.767±0.06 | 0.762±0.04 |
| | GSplat | 0.696±0.08 | 0.686±0.08 | 0.677±0.08 | 0.677±0.08 | 0.682±0.08 | 0.686±0.08 |
| | BACON | 0.735±0.02 | 0.701±0.03 | 0.613±0.04 | 0.538±0.06 | 0.517±0.06 | 0.764±0.15 |
| | Grid | 0.670±0.09 | 0.577±0.08 | 0.524±0.08 | 0.511±0.09 | 0.504±0.08 | 0.584±0.09 |

Table 25: Comparison of IoU across model sizes on 3D Dragon Super Resolution tasks.

| Task | Model | 1e+04 | 3e+04 | 1e+05 | 3e+05 | 1e+06 | 3e+06 |
|---|---|---|---|---|---|---|---|
| 3D Dragon Occupancy | FFN | 0.28 | 0.37 | 0.28 | 0.24 | 0.29 | 0.35 |
| | SIREN | 0.75 | 0.91 | 0.93 | 0.82 | 0.43 | 0.22 |
| | WIRE | 0.92 | 0.94 | 0.93 | 0.67 | 0.75 | 0.81 |
| | GA-Planes | 0.93 | 0.95 | 0.95 | 0.95 | 0.95 | 0.95 |
| | Instant-NGP | 0.21 | 0.30 | 0.43 | 0.45 | 0.43 | 0.37 |
| | BACON | 0.79 | 0.84 | 0.88 | 0.16 | 0.16 | 0.16 |
| | Grid | 0.77 | 0.82 | 0.86 | 0.88 | 0.91 | 0.90 |
| 3D Dragon Surface | FFN | 0.00 | 0.03 | 0.05 | 0.07 | 0.12 | 0.08 |
| | SIREN | 0.28 | 0.36 | 0.45 | 0.57 | 0.27 | 0.04 |
| | WIRE | 0.30 | 0.24 | 0.17 | 0.16 | 0.26 | 0.47 |
| | GA-Planes | 0.44 | 0.55 | 0.59 | 0.60 | 0.59 | 0.60 |
| | Instant-NGP | 0.02 | 0.06 | 0.12 | 0.17 | 0.17 | 0.22 |
| | BACON | 0.33 | 0.40 | 0.48 | 0.51 | 0.04 | 0.04 |
| | Grid | 0.27 | 0.32 | 0.38 | 0.42 | 0.50 | 0.50 |

