# OpenReview forum: "Grids Often Outperform Implicit Neural Representation at Compressing Dense Signals"
_NeurIPS.cc/2025/Conference — NeurIPS 2025 poster_

### Official Review · Reviewer_xXWx · 2025-06-13

**Clarity:** 2
**Significance:** 2
**Originality:** 2
**Rating:** 3
**Confidence:** 4

**Summary:**

This paper investigates INRs to further our understanding of their representation capacity, scaling behavior, and implicit biases. The main message of the paper is that simple grid-based interpolation methods outperform INRs in some signal representation tasks.

**Questions:**

See weaknesses.

**Ethical Concerns:**

["NO or VERY MINOR ethics concerns only"]

**Final Justification:**

I acknowledge that the authors have addressed most of my concerns regarding the setup and results, and I recognize the value of many of the insights presented in the paper. However, my main concern remains: the paper is primarily a benchmark study (which is indeed valuable), as I do not see any clear novelty. There is neither a new algorithm or method that builds upon the experimental insights, nor any newly developed metrics. Consequently, I have only increased my score to 3 and still recommend against accepting the paper.

**Limitations:**

Yes.

**Paper Formatting Concerns:**

N/A.

**Quality:**

2

**Strengths And Weaknesses:**

## Strengths:

- The datasets (synthetic and real) and models chosen for evaluation are comprehensive which also includes classical and hybrid methods.

- The main message of the paper is empirically well-supported.



## Weaknesses:

- Main issue: This paper is more of a position, benchmarking, tutorial, and/or a survey paper as the only novelty here is quantification of the capacity of INRs (and other competing methods including classical ones) in terms of BW, signal type, and model size. The other two contributions are observations from running extensive experiments.

- Why the fine-tuning takes place on one dataset (the star dataset)? An explanation is needed here in order to fully trust the observations on other datasets/models.

- A justification is needed on why omega is set 90 instead of 30 for SIREN. This goes for any other selection of parameters that is different from the recommendation of original papers.

- Missing discussion/evaluation on a recent INR method with multiple output heads, STRAINSER [A]. This method showed improved results when compared to SIREN.

- For CT reconstruction, can the authors try AAPM?


## Minor comments:

- "and discrete models" in line 64.

- Citations are needed in line 69.

- ". All" in line 72.

- Define acronyms such as WIRE, GA-Planes, and NGP, etc, like what was done for Gaussian Splatting (2DGS).

- "Figs. 1 and 2" in line 138.


## References:

[A] https://neurips.cc/virtual/2024/poster/96276

---

> ### Author Rebuttal · Authors · 2025-07-30
>
> We thank the reviewer for their time and thoughtful engagement with our work. We are happy to respond to the reviewer’s questions and concerns below, and to follow up as needed during the discussion phase.
>
> >About the reviewer’s main concern
>
> We appreciate the reviewer’s thoughtful concern about our paper, so we would like to be clear about our work and its contributions. We (1) introduce a benchmark set of tasks including real and carefully designed synthetic signals, and different model sizes, to evaluate compression performance of diverse current and future INRs, (2) we evaluate a diverse suite of INRs, and a grid baseline, on this benchmark, and report comprehensive quantitative and qualitative results of our experiments, and (3) from this evaluation, we extract insights on the compressive capacity of different models and what types of signals are best suited to INR compression. Our study differs from prior work on INRs in that we explicitly focus on the compressive regime to understand the fundamental representation capacity of different INRs on different types of signals and inverse problems.
> This is not a position paper, a tutorial, or a survey; we are actively evaluating methods in a new way that provides meaningful insights to inform future INR application and development. Our experiments reveal that (1) INRs do not have infinite resolution, but instead their effective resolution scales with model size (for example, this is visually evident in the Star Target error maps in Figure 10), (2) on many signals, including natural images, CT scans, and bandlimited noise, INRs are unable to compress better than a simple interpolated grid, and (3) the only signals for which INRs do out-compress grids are those with underlying lower-dimensional structure, such as shapes with constant regions and sharp edges.
>
> >Why is the hyperparameter tuning limited to one dataset?
>
> Exhaustive hyperparameter tuning across all models, model sizes, and signals is impossible due to computational constraints. Considering this limitation, our goal was not to find a bespoke optimum for every dataset, but to test how a single, reasonable hyperparameter setting transfers across signals with different spectral and structural characteristics. We therefore tuned hyperparameters on the Star Target, which spans all frequency scales, and then evaluated how those settings generalize to other signals. This choice is practically useful for readers who will face tasks that inevitably differ from ours. Exhaustively re-tuning hyperparameters for every model - dataset pair would multiply the compute cost dramatically, both for our study and for future researchers who hope to apply our findings. Finally, our hyperparameter tuning strategy reveals that some models are more sensitive to hyperparameter selection than others, which we believe is also a useful finding to inform practice. We will clarify this reasoning in the revision.
>
> >Justification for SIREN hyperparameter
>
> We do acknowledge that the typical hyperparameter setting for SIREN is 30. However, as described on line 73, we selected the hyperparameter to maximize performance on the Star Target signal, chosen because it contains a range of frequencies in a single image. We used the same hyperparameter tuning strategy for all the models for fair comparison.
>
> >Missing discussion/evaluation on STRAINSER
>
> We appreciate the suggestion to include STRAINSER. However, that method is designed for a different setting—learning shared statistics across datasets with multiple output heads—whereas our benchmark trains each INR from scratch on each individual signal. Because of this mismatch, a direct quantitative comparison would not be apples-to-apples. We will clarify this distinction in the related work section and note how dataset-wide methods like STRAINSER can potentially allow INRs to improve compression by learning dataset-wide statistical properties.
>
> >AAPM for CT reconstruction
>
> AAPM provides periodic challenges with diverse medical imaging datasets. To ensure we address the reviewer’s intent, please specify if the comment refers to a specific example from one of these datasets, or of a method that has been applied to these datasets? Most of these datasets are not easy to find publicly online, but we did find one from this year’s challenge “Generalizable Dose Prediction for Heterogenous Multi-Cohort and Multi-Site Radiotherapy Planning challenge at AAPM 2025”. If that is what the reviewer had in mind, we are happy to include some of these CT images in our revision.
>
> During the rebuttal period, we went ahead and evaluated all the INRs as representations for CT reconstruction on the image “0617-259694+imrt+MOS_33896.npz” from this AAPM challenge. The table below shows results on this task across all model sizes. These results are qualitatively very similar to what we observe on other real or synthetic “dense” signals, in that the Grid baseline outperforms all the INRs.
>
> **Table 1 PSNR (dB) on the AAPM CT task**
> $$
> \\begin{array}{lrrrrrr}
> \\text{Model} & 1\\mathrm{e}4 & 3\\mathrm{e}4 & 1\\mathrm{e}5 &
>                  3\\mathrm{e}5 & 1\\mathrm{e}6 & 3\\mathrm{e}6 \\\\ \\hline
> \\text{FFN}         & 11.04 & 16.10 & 22.06 & 26.05 & 29.39 & 30.10 \\\\
> \\text{SIREN}       & 12.03 &  9.16 &  7.63 &  8.92 &  6.76 &  7.26 \\\\
> \\text{WIRE}        & 17.94 & 20.31 & 20.98 & 21.48 & 21.92 & 22.33 \\\\
> \\text{GA-Planes}   & 28.01 & 30.36 & 29.18 & 29.69 & 27.38 & 28.48 \\\\
> \\text{Instant-NGP} & 11.50 & 17.28 & 17.92 & 15.31 & 16.39 & 20.95 \\\\
> \\text{BACON}       & 15.14 & 15.45 & 11.85 & 12.71 &  7.13 &  5.09 \\\\
> \\text{FINER}       & 29.40 & 30.34 & 29.67 & 27.67 & 15.51 & 13.69 \\\\
> \\text{Grid}      & \\mathbf{34.50} & \\mathbf{38.61} & \\mathbf{39.80} &
>                      \\mathbf{38.67} & \\mathbf{37.53} & \\mathbf{31.68} \\\\
> \\end{array}
> $$
>
> >Minor comments
>
> We appreciate the reviewer’s careful reading; we will correct these typos in our revision.

---

> ### Author Response · Authors · 2025-08-03
>
> We thank the reviewer for engaging with our response, and are glad we have at least partially addressed your comments. Regarding your remaining concerns:
>
> > Limited novelty
>
> To our knowledge, our work is the first to systematically evaluate INRs in the compressive regime. Existing works typically evaluate only on a specific sparse-signal task (e.g. shape fitting or radiance fields) or only consider overparameterized INRs. Focusing on compression (i) tests how well different models can capture essential structure in diverse signals, (ii) reveals the implicit biases of each model, and (iii) reduces memory usage.
>
> > There could exist a set of hyperparameters and an INR architecture that outperform the grid
>
> We agree that this is possible, but believe it in no way diminishes the contribution or utility of our work. To clarify, we are not arguing that a grid is the best representation due to a fundamental limitation of INRs, or that INRs should be completely avoided. On the contrary, we are showing that current INRs achieve good compression only on certain signals, and are sensitive to hyperparameters, to encourage further development of INRs to reach their full potential for generalizable compression.
>
> > Justification for tuning hyperparameters on a single dataset (the Star Target)
>
> The decision to tune hyperparameters on a single dataset was out of necessity, as it is computationally prohibitive to tune hyperparameters separately on every dataset, every model architecture, and every model size. For our experiments, this would require 1,872 hyperparameter tuning sweeps (39 tasks, 8 models, and 6 sizes for each model). Given this computational reality, we chose to tune hyperparameters on the Star Target because it spans a wide range of frequencies.
>
> > AAPM CT results on a single image
>
> We have now repeated the same AAPM CT experiment on 6 images, and report average results (and standard deviations) in the table below. Qualitatively, these results match what we saw with a single AAPM CT image and with our original CT image.
>
> | Model | 1e4 | 3e4 | 1e5 | 3e5 | 1e6 | 3e6 |
> |-------|-----|-----|-----|-----|-----|-----|
> | FFN         | 11.26±0.96 | 16.35±0.79 | 22.64±1.01 | 26.56±0.84 | 29.75±1.09 | 29.28±1.13 |
> | SIREN       | 12.19±0.75 | 11.18±3.60 |  8.15±0.73 |  8.70±0.65 |  7.23±0.36 |  6.80±1.10 |
> | WIRE        | 18.43±0.71 | 20.48±0.68 | 20.99±0.48 | 21.46±0.34 | 21.83±0.29 | 22.28±0.14 |
> | GA-Planes   | 32.19±2.36 | 34.63±2.48 | 34.06±2.80 | 32.94±1.89 | 32.75±2.70 | 30.84±1.23 |
> | Instant-NGP | 11.46±0.75 | 18.20±2.13 | 18.20±0.73 | 15.76±1.66 | 17.62±1.64 | 21.29±1.15 |
> | FINER       | 31.79±1.44 | 31.44±1.09 | 29.87±1.14 | 28.13±1.14 | 16.16±3.29 | 15.42±2.01 |
> | BACON       | 15.82±0.64 | 15.47±1.27 | 12.11±0.84 | 12.74±2.62 |  9.13±2.22 |  5.16±0.27 |
> | Grid        | **39.93**±3.40 | **41.90**±2.81 | **42.30**±3.92 | **41.66**±4.91 | **38.55**±1.78 | **31.45**±1.59 |
>
> > Unfair to test INRs that were not designed for CT on CT
>
> We agree that most INRs were not designed for CT, and that several of these INRs show poor performance on CT. However, we point out that (i) grids were also not designed for CT, and yet generalize well to this task, and (ii) many papers have been published showing promising performance with INRs for CT, though these are not focused on the compressive regime [1-5]. Our results complement this literature by providing an important nuance: INRs struggle with compression for dense natural signals like CT scans, even though they can do well when over-parameterized.
>
> [1] M. Najaf and G. Ongie. "Accelerated Optimization of Implicit Neural Representations for CT Reconstruction," IEEE ISBI, 2025.
> [2] D. Ruckert, Y. Wang, R. Li, R. Idoughi, and W. Heidrich. "NeAT: Neural Adaptive Tomography," ACM ToG, 2022.
> [3] B. Xiong, C. Su, Z. Lin, Y. Zhou, and Z. Yu. "INeAT: Iterative Neural Adaptive Tomography," arXiv, 2023.
> [4] J. Lee and J. Baek. "Iterative reconstruction for limited-angle CT using implicit neural representation," Physics in Medicine and Biology, 2024.
> [5] J. Shi, J. Zhu, D. Pelt, J. Batenburg, and M. B. Blaschko. "Implicit Neural Representations for Robust Joint Sparse-View CT Reconstruction," TMLR, 2024.
>
> > Additional details about experimental settings
>
> We thank the reviewer for their interest in the details of our experiments. We have already provided the following specifications in the original submission, and would be happy to provide any further details that the reviewer would like to request.
> - Hyperparameters for all models, in Table 3 in the appendix
> - Architecture dimensions for all models and sizes, in Table 4 in the appendix (with explanation in section 5.5)
> - Complete code for full reproducibility, in the supplement zip file
>
> Please let us know if there are any additional details we can provide.

---

> > ### Comment · Reviewer_xXWx · 2025-08-05
> >
> > I would like to thank the authors for their response.
> >
> > - Novelty: I acknowledge that the work is valuable and provides (mostly) well-supported insights about INR methods and a grid representation. The only novelty seems to the  For this reason, I think that this work would be a better fit for the benchmark or the position track.
> >
> > - Tuning Hyper-parameters justification: I didn't request to run the tuning of the hyper-parameters on all datasets and combinations. I wanted an explanation of why one dataset was used. In terms of datasets that span wide range of frequencies, couldn't the author have tried another dataset?
> >
> > - CT Results: I acknowledge that the authors have included average results over six images. The results do support that the grid representation outperforms all other INR methods where many of the considered INR methods were not designed for CT. However, I am still unconvinced about this comparison even if the grid representation was also not designed for CT as there could be a set of parameters that would make an INR method (e.g. GA planes or SIREN) performs significantly well. Therefore, I see these results to be a bit inconclusive. Furthermore, I highly encourage the authors to provide some explanation onto why certain methods work (such as GA-Planes) and others report very low PSNRs (such as SIREN).
> >
> > - Experimental details: I meant the details for the experiments reported in the rebuttal.

---

> > > ### Author Response · Authors · 2025-08-05
> > >
> > > We thank the reviewer for their continued engagement with our work. Below we address the points raised by the reviewer; we are happy to answer any further questions as needed.
> > >
> > > Novelty: While there is a benchmarking component to our contribution, our work goes beyond a pure benchmarking study by extracting useful insights to guide deployment of existing INRs and development of future INRs. Our results suggest that underlying lower dimensional structure (such as a 2D surface embedded in 3D, or a 1D curve or edge embedded in 2D) is key for existing INRs to be able to compress. One of the interesting things we find is that many "dense" natural signals like CT and natural images behave more like bandlimited noise than like the “sparse” synthetic signals and shapes where there is underlying lower dimensional structure like sharp edges and constant regions. We believe that there is still some structure to these “dense” natural signals that should enable them to be compressed, but our work reveals that current INRs are not meeting that goal. We will emphasize this insight in the revised paper.
> > >
> > > Tuning hyper-parameters: We are not sure if the reviewer is asking about why we tuned hyperparameters on one dataset instead of on several datasets, or if the reviewer is asking why we chose the Star Target as the dataset for hyperparameter tuning, instead of a different dataset. If the question is why one dataset instead of several, this was for uniformity to be able to interpret the results. If we were to tune on multiple datasets but not all datasets, it’s not clear which hyperparameters to use for the remaining datasets. If the question is why we chose to tune hyperparameters on the Star Target in particular, this is because it is a single image that contains a wide range of spatial frequencies, with sharp details (high frequencies) in the center and larger-scale features (lower frequencies) around the periphery. There are other synthetic and real images that also span a range of frequencies, but we do not see a reason to prefer any of them over the Star Target. Does the reviewer have a particular image in mind that they believe would be a better target for hyperparameter tuning?
> > >
> > > CT Results: It seems the reviewer is not convinced by the additional requested experiments on AAPM CT reconstruction because the hyperparameters were not tuned on CT images. Since it is computationally intractable to tune hyperparameters individually on each dataset, we tuned hyperparameters on the Star Target and used these hyperparameters for all other signals, including the CT images. As the reviewer points out, some methods (e.g. Grid and GA-Planes) perform fairly well on the CT images, while others (e.g. SIREN) perform poorly. We believe this variation is partly due to the variation in hyperparameter sensitivity of different representations. Our experiments reveal that Grid and GA-Planes are robust to hyperparameter choices and perform well across datasets, while some other INRs including SIREN are sensitive to hyperparameter choices (note SIREN does well on the Star Target, in figure 2 and figure 10, but poorly on CT, in figure 18). This hyperparameter sensitivity of SIREN has also been noted in prior work. For example, in [1, 2] SIREN is pointed out to be sensitive to hyperparameter tuning, and in the GitHub for the official SIREN repository issue #31 points out that SIREN performs poorly when under-parameterized (compressive), which is exactly the regime we focus on exploring.
> > >
> > > [1] $SL^2 A-INR$: Single-Layer Learnable Activation for Implicit Neural Representation
> > >
> > > [2] H-SIREN: Improving implicit neural representations with hyperbolic periodic functions
> > >
> > > Experimental details: The rebuttal experiments on AAPM CT images use exactly the same setup (hyperparameters, model sizes, etc.) as the CT experiments in the original paper, just with different images. The 6 AAPM images we include are all from the “Generalizable Dose Prediction for Heterogenous Multi-Cohort and Multi-Site Radiotherapy Planning challenge at AAPM 2025”, using the following specific image files: HNC_001+A4Ac+MOS_25934.npz, HNC_001+9Ag+MOS_25934.npz, 0617-492081+imrt+MOS_10706.npz, 0617-492081+2Ac+MOS_10706.npz, 0617-259694+imrt+MOS_33896.npz, 0617-259694+2Ac+MOS_33896.npz.

---

### Official Review · Reviewer_y2XA · 2025-07-01

**Clarity:** 3
**Significance:** 4
**Originality:** 2
**Rating:** 5
**Confidence:** 4

**Summary:**

The paper investigates the performance of various representations for signals, such as implicit neural representations (INRs) and grids, for various signal based tasks like super-resolution. The aim is to show the trade-offs in different wanted properties (representation, generalization, computational performance) between methods. They conclude that Grids often give much more favourable trade-offs in many situations that INRs are currently favoured for.

**Questions:**

- As grids suffer from the curse of dimensionality, would you expect the results to hold in higher dimensions. Would you consider hierarchical or adaptive grids as a better alternative?
- What about parameterized interpolation, e.g. attention based etc
- Do you expect the same results to hold with grids when dimensions are not the same euclidean type, e.g. rotation dimensions in NeRF, or time in video?
- Have you tried hyperparameter tuning on each signal type at all? If so, were the hyperparameters from star target dataset comparable?

**Ethical Concerns:**

["NO or VERY MINOR ethics concerns only"]

**Final Justification:**

During the discussion I mainly further discussed my points about loss/regularisation and dimensionality. While I do not feel that these point have been completely addressed, I believe the authors understand my points and have a reasonable viewpoint. I am still very positive about this work, as it's implications are important to the field, and I hope it will convince other researchers to explore the finding of this paper further. As a result, I am increasing my score to an accept. In a revised or resubmitted version, I hope the authors properly address all raised points from all reviewers.

**Limitations:**

Not really, they discuss benefits and limitations of grid-based and INR approaches, but not of their work.

**Quality:**

2

**Strengths And Weaknesses:**

Strengths
- Great motivation, determining where each method has more favourable performance, and discussing the trade-offs
- Good investigation of where grids and INRs do better than each other
- Commitment to release everything for other researchers to try out and expand upon this

Weaknesses
- The simple loss of MSE is not enough for certain tasks, often regularization is used to improve performance or reduce issues. For example: INRs introduce very high frequencies due to spectral bias, but this causes too much high frequency artifacts to remain so regularization is often used to control this. On the other hand, this is somewhat baked into the bicubic and trilinear interpolation by design.
- Hyperparameter tuning on 1 (synthetic) dataset is very limiting
- 2D and 3D only have been investigated. As high dimensions are used for the input (e.g. NeRF or video) grids maybe become less appealing. There is no investigation or discussion of that.
- 3D dragon surf and occ are not very different, should instead be doing different shapes  (so a dataset like is done for images) as results will change a lot depending on shape properties
- I have seen most works reproting that out of the activation-based INRs, FINER (CVPR2024) has the best performance, would be nice to compare to that
- The hash collisions in Instant-NGP are a problem (work well for NeRF but not in general), would be nice to compare to their earlier work NGLOD or an adaptive version like ACORN
- I feel like there is a lot of hyperparameters here to play with that probably have not been explored. E.g. the hyperparameters for Instant-NGP provided by the code of that paper is very much NeRF based, and my experience is that vastly different configurations are optimal for shape fitting. I have also used SIREN very successfully on 3D shapes (have specifically used the dragon shape!) and have found it performs as well as WIRE, only FINER can slight outperform them. Furthermore, the performance for all three can be made much higher than reported here. An important part of getting it to work is having good regularization, for example the eikonal term, and good sampling strategies, e.g. denser around the surface but still covering the space.


 Overall:
 - I really like the direction the paper is heading towards, INRs are being used in areas that grids would suffice, and there needs to be better investigation on where each method is best suited for.
 - I have issues with the hyperparameter tuning, but it is impossible to search the huge hyperparameter space effectively. Thus, I think a benefit of this work is that it provides a set of experiments for determining tradeoffs that future papers can use to explore with. Researchers who like INRs can try improving the tradeoffs from this paper (I personally think performance can be significantly improved in many places), and researchers who like grids can counter back with better approaches as well.
 - I also have issues with the loss, would be good to have included regularization for INRs, but this can be explored in the future
 - However, this would make this a very useful paper if the experiments proposed have wide coverage, but 1 CT and 1 3D shape is quite limiting.

---

> ### Author Rebuttal · Authors · 2025-07-30
>
> We thank the reviewer for their time and thoughtful engagement with our work.
>
> >MSE loss, spectral bias, and regularization
>
> We agree that the MSE is not a perfect metric. In the appendix, we also include evaluation of SSIM and LPIPS for 2D signals and IoU for 3D shapes. Full visualizations and some error maps are also available in the appendix for qualitative comparison. Regarding spectral bias and INR regularization, we agree that some INRs (such as INGP) do introduce high frequency artifacts, and that most INRs struggle to represent perfectly constant regions like the background of the Star Target (this is evident in the error maps in Figure 10, for example). However, as we are specifically focusing on the compressive regime, the compression itself should act as a form of regularization–and indeed these “implicit biases” of INRs are often considered as a reason to use them. For clarity, we note that the term “spectral bias” typically refers to bias towards low frequencies; INRs often use modifications such as Fourier embedding to overcome spectral bias and allow them to capture higher frequencies. These modifications can have the unwanted side effect of inducing high-frequency artifacts.
>
> >Hyperparameter tuning
>
> We appreciate the reviewer's understanding that full hyperparameter tuning is impossible in this setting. Considering this limitation, our goal was not to find a bespoke optimum for every dataset, but to test how a single, reasonable hyperparameter setting transfers across signals with different spectral and structural characteristics. We therefore tuned hyperparameters on the Star Target, which spans all frequency scales, and then evaluated how those settings generalize to other signals. This choice is practically useful for readers who will face tasks that inevitably differ from ours. Exhaustively re-tuning hyperparameters for every model - dataset pair would multiply the compute cost dramatically, both for our study and for future researchers who hope to apply our findings. Finally, our hyperparameter tuning strategy reveals that some models are more sensitive to hyperparameter selection than others, which we believe is also a useful finding to inform practice. We will clarify this reasoning in the revision.
>
> >Experiments for NeRF and video
>
> We focus our experiments on 2D and 3D signals as these cover many practical use cases for INRs, though we agree that future work should explore higher-dimensional signals. That said, we note that NeRF uses an INR that is also 3D (for x, y, z, since the dependence on view direction is factored out in their architecture), and that standard videos are also 3D (x, y, t), though volumetric videos are 4D (x, y, z, t). We observe very similar qualitative results (comparisons between INRs) in 2D and 3D, so we expect that these trends will likely continue or be amplified in higher dimensions. For example, a signal that has a low intrinsic dimension (e.g. an object surface) embedded in a higher dimension should be even more compressible by an INR, while we expect “dense” signals will remain challenging to compress in higher dimensions.
>
> >3D surface and occupancy experiments
>
> Our rationale for including both the object and its surface as separate tasks was to distinguish whether INRs can only compress sparse signals (the surface) or can compress signals with underlying lower-dimensional structure (which applies to both the object and its surface). Our results show that INRs are generally able to compress signals with underlying lower dimensional structure, whether or not these signals are sparse. Unfortunately due to the restriction against including figures or links in our response we cannot include experiments on additional shapes here, but we are happy to include these in the revised paper.
>
> >Comparing against other models (FINER, NGLOD, ACORN)
>
> Due to limited computational resources and the large number of INRs in the literature, we selected representative models from broad categories of INR. Specifically, we tested SIREN and WIRE which are similar to FINER (all use a standard MLP with a modified activation function), and we tested BACON which is similar to NGLOD and ACORN (all use a multiscale or bandlimited strategy). However, we do agree that these models have shown promising results in the literature (albeit not in the compressive regime) and therefore warrant discussion; we will include them in the revised paper.
>
> During the rebuttal period, we were able to evaluate FINER on a subset of our benchmark tasks, and on CT reconstruction with an additional CT dataset from AAPM (as requested by Reviewer xXWx). Unfortunately we are not able to share figures or links in our rebuttal, and character limits prevent posting full tabular results. So, we share just the results on the new AAPM CT reconstruction task [Table 2] while also including the FINER results for the original CT data [Table 1], in which FINER does perform on par with or slightly better than the best INR, but the Grid baseline still performs best. So far we observe results for FINER that are generally consistent with what we observe on other INRs: for real and synthetic “dense” signals the Grid is still best, while for signals with sparsity or underlying lower-dimensional structure FINER (and a few other INRs) can beat the Grid.
>
> **Table 1  PSNR (dB) on the original CT task**
>
> $$
> \\begin{array}{lrrrrrr}
> \\text{Model} & 1\\mathrm{e}4 & 3\\mathrm{e}4 & 1\\mathrm{e}5 &
>                  3\\mathrm{e}5 & 1\\mathrm{e}6 & 3\\mathrm{e}6 \\\\ \\hline
> \\text{FFN}         & 19.63 & 21.31 & 11.77 & 16.44 & 14.32 &  4.86 \\\\
> \\text{SIREN}       & 18.44 & 22.05 & 26.32 & 28.32 & 29.70 & 23.82 \\\\
> \\text{WIRE}        & 12.29 &  9.65 &  7.61 &  7.40 &  6.38 &  6.53 \\\\
> \\text{GA-Planes}   & 20.69 & 21.93 & 21.59 & 21.87 & 22.07 & 22.12 \\\\
> \\text{Instant-NGP} & 27.27 & 28.53 & 27.99 & 26.80 & 25.76 & 24.74 \\\\
> \\text{GSplat}      & 18.04 & 24.21 & 22.91 & 21.69 & 22.72 & 26.33 \\\\
> \\text{BACON}       & 24.29 & 24.35 & 24.66 & 25.02 & 24.97 & 24.70 \\\\
> \\textbf{FINER}       & 27.14 & 28.28 & 28.82 & 28.56 & 23.63 & 21.27 \\\\
> \\text{Grid}      & \\mathbf{28.57} & \\mathbf{31.03} & \\mathbf{34.08} &
>                      \\mathbf{35.00} & \\mathbf{32.99} & \\mathbf{34.01} \\\\
> \\end{array}
> $$
>
> **Table 2 PSNR (dB) on the AAPM CT task**
> $$
> \\begin{array}{lrrrrrr}
> \\text{Model} & 1\\mathrm{e}4 & 3\\mathrm{e}4 & 1\\mathrm{e}5 &
>                  3\\mathrm{e}5 & 1\\mathrm{e}6 & 3\\mathrm{e}6 \\\\ \\hline
> \\text{FFN}         & 11.04 & 16.10 & 22.06 & 26.05 & 29.39 & 30.10 \\\\
> \\text{SIREN}       & 12.03 &  9.16 &  7.63 &  8.92 &  6.76 &  7.26 \\\\
> \\text{WIRE}        & 17.94 & 20.31 & 20.98 & 21.48 & 21.92 & 22.33 \\\\
> \\text{GA-Planes}   & 28.01 & 30.36 & 29.18 & 29.69 & 27.38 & 28.48 \\\\
> \\text{Instant-NGP} & 11.50 & 17.28 & 17.92 & 15.31 & 16.39 & 20.95 \\\\
> \\text{BACON}       & 15.14 & 15.45 & 11.85 & 12.71 &  7.13 &  5.09 \\\\
> \\textbf{FINER}       & 29.40 & 30.34 & 29.67 & 27.67 & 15.51 & 13.69 \\\\
> \\text{Grid}      & \\mathbf{34.50} & \\mathbf{38.61} & \\mathbf{39.80} &
>                      \\mathbf{38.67} & \\mathbf{37.53} & \\mathbf{31.68} \\\\
> \\end{array}
> $$
>
>
> >Would you consider hierarchical or adaptive grids as a better alternative?
>
> It is true that the memory of a grid representation faces the curse of dimensionality, assuming fixed resolution in each dimension. In our experiments we use different grid resolutions for 2D vs 3D signals, to keep the total model size and compression level fixed across dimensions. In practice when high resolution is required in 3D or 4D it makes sense to save some memory by using adaptive or multiresolution grids, especially when the signal to be fit is actually spatially varying in level of detail (for pure bandlimited signals this is unlikely to help). However, there is a risk that adaptive grids may struggle with optimization or be more sensitive to hyperparameter choices due to having to make discrete decisions about splitting and pruning.
>
> >About parameterized interpolation
>
> This is an interesting direction for future investigation. For example, "efunc: An Efficient Function Representation without Neural Networks" proposes a more adaptive interpolation function for a grid representation, and shows better memory efficiency on a shape fitting task compared to INRs. It may be that this sort of parameterized-interpolation grid can outperform INRs for compression even for signals with underlying lower dimensional structure, which is the setting where we find INRs outperform standard grids. We will add this discussion to the revised paper.
>
> >Grids in non-Euclidean spaces
>
> In general we do expect our findings to generalize to non-Euclidean signals, such as signals defined over the sphere. A grid in this setting would likely need to be parameterized carefully (e.g. in a transform domain) to ensure the effective resolution is matched to the type of signal.
>
> >Limitations of our work
>
> We will add a discussion of limitations of our study, namely that we tune hyperparameters on one dataset, and we consider a subset of possible models and a subset of possible types of signals.

---

> > ### Comment · Reviewer_y2XA · 2025-08-05
> >
> > Thanks to the authors for their response. I have some follow-up questions and comments (I apologise for responding so late).
> >
> > **"We agree that the MSE is not a perfect metric"**
> > - My issue is its use as the only loss, not as a metric.
> >
> > **"the compression itself should act as a form of regularization–and indeed these “implicit biases” of INRs are often considered as a reason to use them"**
> > - I don't get your point about the compression only should act as regularization, what do you mean by this? While the implicit smoothness bias of INRs are one reason to use them, there are many others that relate to this point. For example, in many cases first order or second order regularization loses are not only helpful but meaningful to the task (e.g., curvature losses). These can easily be computed with INRs, but are harder and less accurate to implement with grids. Not allowing regularization in the loss still seems like a major weakness to me, especially since smoothing is done with the grids though interpolation using specifically chosen regions (16 nearest rather than 4 nearest in 2D, etc.).
> >
> > **"That said, we note that NeRF uses an INR that is also 3D (for x, y, z, since the dependence on view direction is factored out in their architecture), and that standard videos are also 3D (x, y, t), - though volumetric videos are 4D (x, y, z, t)"**
> > - I don't agree, the standard NeRF model can be factored into two models (overlapped), a model for density that is 3D input 1D output, and a model for color that is 5D input and 3d output. Thus NeRF is quite high-dimensional compared to the current experiments.
> >
> > **"We observe very similar qualitative results (comparisons between INRs) in 2D and 3D, so we expect that these trends will likely continue or be amplified in higher dimensions"**
> > - The current 3D results aren't very dense, shapes are intrinsicly low dimensional compared to images, a point you yourself make w.r.t. other reviewers' points about iNGP. Standard and volumetric video would be very different, so I don't agree that you can assume the trends will continue. Thus, there is not much investigation in the paper about scaling to higher dimensions.
> >
> > **New FINER results**
> > - Thank you for providing these. I still think that the performance for all three (SIREN, WIRE, FINER) can be made much higher than reported, but as long as the authors release data and code this can be explored by others (thus making this work useful for the community). I especially think this is the case for 3D shapes which are very sparse signals: grids are very wasteful while INRs can allocate their capacity acoordingly as long as the correct guidance is given, which depends on the loss and sampling strategies.
> >
> > **Responses with other reviewers**
> > - In one of the responses you mentioned "__One of the interesting things we find is that many "dense" natural signals like CT and natural images behave more like bandlimited noise than like the “sparse” synthetic signals and shapes where there is underlying lower dimensional structure like sharp edges and constant regions. We believe that there is some structure to these “dense” natural signals that should enable them to be compressed, but our work reveals that current INRs are not meeting that goal. We will add this discussion to the revised paper.__" This would be an extremely useful insight if properly investigated, and I agree with that in general the literature seems to point towards this (INRs are not compressing as much as our intuition would expect). I don't agree that the paper currently reveals that as there are issues with the setup, and this is not really being investigated directly (though I don't know a good way to investigate this directly).

---

> > > ### Author Response · Authors · 2025-08-05
> > >
> > > We thank the reviewer for their thoughtful and continued engagement with our work. We are happy to clarify a few points raised, and to follow up as needed if there are further questions.
> > >
> > > Loss function and regularization: We use MSE as the training loss for consistency with the papers that introduced the different INRs we compare, all of which use MSE. Other losses are sometimes used for specific tasks, such as Eikonal constraints for training signed distance functions or curvature losses for learning surface normals, but these do not apply to the tasks we study. Does the reviewer have a particular paper in mind that uses a certain regularization loss on one of the types of tasks we study? If so we would be happy to consider it.
> > >
> > > We can also clarify the point in our rebuttal about compression itself being a form of regularization. Regularization is anything that constrains an underdetermined optimization problem to return a solution within a particular desired set. For example, L2 regularization tells the optimization to choose the smallest answer that is consistent with the data. Total variation regularization tells the optimization to choose the answer with the fewest edges. If a model is compressed, it can only represent a subset of possible signals, because it has limited flexibility due to its small parameter count. How these parameters are allocated (how the compression is done) controls exactly what possible signals can be represented with the compressed model. For example, for the Grid model compression is analogous to limited resolution, which regularizes the solution to be low-frequency. The compression-induced regularization for INRs is more complicated (and less well understood), but it is a form of regularization nonetheless.
> > >
> > > Regarding the input dimensionality of NeRF, we believe this is a notational difference rather than a practical one. Per Figure 7 in the original NeRF paper, the NeRF model factors out the dependence on position (3D) and view direction (2D). Because these are factored out we were thinking of them as separate lower-dimensional models rather than a single 5D model, but we acknowledge this is also a valid interpretation of their factored representation since the lower-dimensional models do share parameters. We agree that extending our study to 4D and 5D signals if of interest for future work.
> > >
> > > Regarding the density of our 3D signals, our real 3D signals are indeed sparse (the Stanford Dragon occupancy mask and surface mask), but our synthetic 3D signals include both sparse and dense signals. The 3D Spheres signal is sparse, but the 3D Bandlimited signal is dense. We acknowledge that our 3D experiments do not include any signals that are both real and dense, but in our 2D experiments the results on the dense Bandlimited synthetic signal were fairly predictive of performance on dense real signals. We agree that testing if this pattern continues for dense real 3D signals is of interest for future work.
> > >
> > > We appreciate that the reviewer read our response to other reviews, and finds the pattern of INRs struggling to compress “dense” signals to be insightful and useful. This pattern is something we hypothesize based on the results of our more general benchmarking analysis, but we acknowledge that our study was not explicitly designed around this hypothesis (since it arose after seeing our results). We agree that it is both interesting and nontrivial to directly test this hypothesis, and will point it as a priority for future work.

---

> > > > ### Comment · Reviewer_y2XA · 2025-08-08
> > > >
> > > > - For both shape representation and surface representation, I would recommend using SDFs (and thus having the eikonal loss for regularization) because they are much better regularized than occupancy functions. This is the reason why SDFs are much more widely used in the 3D shape reconstruction literature, and also in the shape reconstruction from multiple views (NeuS and VolSDF derived). Despite this being the general consensus there isn't a good paper for me to point to, the best I can provide is Sec 4.2 in NICER-SLAM [1] (SDF vs Occupancy) which explicitly evaluate this in their context.
> > > >
> > > > [1] Zhu et al, NICER-SLAM: Neural Implicit Scene Encoding for RGB SLAM, 3DV 2024
> > > >
> > > > - For images, could have something like reducing the Dirichlet energy to reduce high-frequency noise, or even ensuring there is weight decay in the optimization.
> > > >
> > > > - Also, just to be clear, I am not expecting you to run anything (especially this close to the end of the discussion period), just want to see if you think this is worthwhile for the paper in perhaps a revised edition or not.
> > > >
> > > > ---
> > > >
> > > > - I now understand your point about about compression itself being a form of regularization, but I feel like comparing grids with extra smoothing (interpolation of 16 nearest rather than 4 nearest) to INRs without extra smoothing (regularization losses) is unfair, regardless of whether both models are within the compressive regime
> > > >
> > > > ---
> > > >
> > > > - "Regarding the input dimensionality of NeRF, we believe this is a notational difference rather than a practical one" I don't agree, if you were to implement this as a grid, then you need a 5D input grid for color (at each point in space and view direction towards that point there needs to be a color value stored)
> > > >
> > > > ---
> > > >
> > > > - I forgot about your dense synthetic 3D signals, thanks for reminding me about that. It seems that BACON and WIRE are better than grids here for many configurations despite the signal not being sparse/low-rank. I think these results, which currently are in the supplementary, might be good to have in the main paper as they are the best indicator for what the trend is for higher-dimensions.

---

> > > > > ### Author Response · Authors · 2025-08-08
> > > > >
> > > > > Thank you for your continued engagement with our work.
> > > > >
> > > > > We appreciate the suggestion to consider SDF experiments, and we agree this is an interesting direction to extend our study. We expect that INRs will likely show strong performance at SDF fitting, based on our current shape and surface fitting experiments and strong results for INRs in the SDF literature. This prediction would also agree with our general finding that INRs show the strongest compression potential for signals with underlying lower-dimensional structure, such as objects and their surfaces. We are interested in exploring this finding more in followup work, which could include SDF fitting and additional INRs or INR regularizations.
> > > > >
> > > > > Regarding the smooth bicubic interpolation we use for our 2D Grid fitting, it’s true that this considers the 16 neighboring points. However, we do not believe that this choice of interpolation qualitatively affects our results, for the following reasons: (i) For some of our 2D synthetic experiments (Spheres, Sierpinski, and Star Target) we find INRs that outperform the interpolated Grid, and (ii) for our 3D Grid experiments we use the simpler trilinear interpolation, which only considers the 8 neighboring points in 3D, and observe similar trends as in 2D.
> > > > >
> > > > > Regarding NeRF, it is possible to parameterize it as a Grid either in 3D or 5D, and both would be interesting to explore for future work. The 3D version was demonstrated by Plenoxels [1], which factored out the two dimensions of view direction by learning a vector of spherical harmonic coefficients at each 3D voxel.
> > > > >
> > > > > [1] Fridovich-Keil*, Yu*, Tancik, Chen, Recht, and Kanazawa. “Plenoxels: Radiance Fields Without Neural Networks,” CVPR 2022.
> > > > >
> > > > > Regarding the 3D Bandlimited signal where Table 19 shows that some INRs can outperform the Grid, we agree that these results are interesting and will highlight them more in the discussion in the revised main paper. We note that these results are presented already in the main paper in the last row of Figure 3, but the differences in PSNR (where the INRs outperform the Grid) are not large enough to really stand out in the heatmap.

---

### Official Review · Reviewer_HUmV · 2025-07-01

**Clarity:** 4
**Significance:** 3
**Originality:** 3
**Rating:** 4
**Confidence:** 3

**Summary:**

The paper presents an empirical comparison between Implicit Neural Representations (INRs) and traditional grid-based representations across diverse signal fitting (2D, 3D) and inverse tasks (super-resolution, denoising, CT). The authors challenge the assumption that INRs are universally superior by conducting comprehensive experiments across different signal types, task categories (overfitting and generalization) and model sizes. The key finding is that simple interpolation grids often outperform INRs in both reconstruction quality and training efficiency, except for the cases when the signal has constant-value region with sharp edges - a known struggle for grid-based methods.

**Questions:**

Please see Weaknesses

**Ethical Concerns:**

["NO or VERY MINOR ethics concerns only"]

**Final Justification:**

My main concern was the lack of discussion of the method's failure cases, and the authors provided some insights and agreed to add this discussion in the final revision. My overall opinion remains positive about the paper being a valid contribution as a thorough empirical study of grid-based methods. However, I agree with the concerns about limited scientific novelty raised by other reviewers. I keep my score at borderline accept.

**Limitations:**

Yes

**Quality:**

3

**Strengths And Weaknesses:**

### Strengths

1) The paper provides valuable insight into grid-based approaches performing surprisingly well despite often being dismissed in the literature as inferior. The authors empirically show that grid-based methods remain competitive baselines for various tasks, demonstrating comparable reconstruction quality while excelling computational efficiency.
2) The experimental setup is comprehensive and ensures fair comparison across methods. The evaluation includes pure INRs, grid-based approaches, and hybrid models. Tasks span synthetic 2D and 3D signals as well as real-world datasets. The methods are benchmarked across different model sizes with accompanying computational analysis.
3) The paper is clearly written and structured, with both qualitative and quantitative results presented well to support the paper's findings.

### Weaknesses

While the paper effectively demonstrates the strengths of grid-based methods, it would benefit from a more thorough discussion of their limitations and clearer guidance on method selection. The authors mention that grids struggle with sharp edges, but I’d like to see a deeper investigation on that matter, e.g. identifying specific signal characteristics that trigger poor performance. An analysis of failure cases beyond sharp discontinuities, along with practical decision criteria for choosing between grids and INRs, would significantly enhance the paper's utility for real-world applications.

---

> ### Author Rebuttal · Authors · 2025-07-30
>
> We thank the reviewer for their time and thoughtful engagement with our work.
>
> >Discussion of guidance for model selection
>
> Our results suggest that underlying lower dimensional structure (such as a 2D surface embedded in 3D, or a 1D curve or edge embedded in 2D) is key for INRs to be able to compress. For pure bandlimited noise, it makes sense that the grid is best because of the Nyquist sampling/interpolation theorem. One of the interesting things we find is that many "dense" natural signals like CT and natural images behave more like bandlimited noise than like the “sparse” synthetic signals and shapes where there is underlying lower dimensional structure like sharp edges and constant regions. We believe that there is still some structure to these “dense” natural signals that should enable them to be compressed, but our work reveals that current INRs are not meeting that goal. We will add this discussion to the revised paper.

---

> > ### Comment · Reviewer_HUmV · 2025-08-05
> >
> > Thank you for your answer. I remain positive about this paper; however, I agree with the concerns about limited scientific novelty raised by other reviewers. I am maintaining my score.

---

> ### Author Response · Authors · 2025-08-05
>
> Thank you for engaging with our work during the discussion period. The reviewer states that they are maintaining their score (4), but in the current state of the review the official score seems to have disappeared...is there a way to correct this? [Update: we saw the email explaining that scores disappear once reviewers have completed their post-rebuttal updates]

---

### Official Review · Reviewer_4qxt · 2025-07-01

**Clarity:** 3
**Significance:** 2
**Originality:** 2
**Rating:** 3
**Confidence:** 4

**Summary:**

This paper investigates the strengths and weaknesses of implicit neural representations (INRs) versus traditional grid-based representations across a variety of tasks. Specifically, it compares the performance (accuracy and computational cost) of pure INRs, hybrid INRs, Gaussian Splatting (GS), and interpolation-based grid methods on vision tasks including image fitting, super-resolution, and CT reconstruction. The experimental results suggest that interpolation-based representations outperform INRs on several tasks.

**Questions:**

See Strengths and Weaknesses above, please.

**Ethical Concerns:**

["NO or VERY MINOR ethics concerns only"]

**Final Justification:**

This work aims to explore the advantages of grid representations over implicit neural representations (INRs) in various inverse problems. However, the experiments severely limit the learning capacity of the INR networks, and the results presented fail to convincingly demonstrate the superiority of grid representations over INRs. Therefore, I find the main claim that "grid often outperforms INR" unconvincing. After reviewing other reviewers’ comments, I believe the current version of this paper falls far below the NeurIPS standard in both novelty and rigor.

**Limitations:**

yes

**Quality:**

2

**Strengths And Weaknesses:**

### Strengths

+ This work conducts extensive experiments. Most mainstream INRs are included, and detailed empirical results, such as accuracy, memory footprint, and running time, are reported, providing valuable insights into the use of INRs.
+ The experiments cover a broad range of tasks, including 2D/3D signal fitting, both real and synthetic data, as well as CT reconstruction, super-resolution, and denoising, making the results comprehensive and convincing.
+ The authors carefully control the parameter count between INR and grid-based models to ensure fair comparison, which makes the results more credible.
+ The code is released in the SM, which improves the reproducibility of this work.
+ The paper is well-written and easy to follow.

---

### Weaknesses

+ The title of the paper, ``Grids Often Outperform Implicit Neural Representations'', is somewhat overstated. As shown from Table 5 to Table 18, grid representations only show a clear advantage on bandlimited signals, and not on other tasks. The title does not strictly reflect the experimental observations.
+ The work is fully empirical and lacks theoretical insights. It does not explain why grid-based representations might be better than INRs on several tasks
+ All experiments are constrained to models with parameter counts between $1\times 10^4$ and $3\times 10^6$. It remains unclear whether the empirical conclusions would hold under larger model capacities.
+ The performance of NGP appears questionable. For example, in Figure 2, NGP results consistently contain severe artifacts. Typically, NGP is considered much more powerful than SIREN and FFN, so these results seem inconsistent with community consensus.

---

> ### Author Rebuttal · Authors · 2025-07-30
>
> We thank the reviewer for their time and thoughtful engagement with our work. We respond to specific points below, and are happy to continue the discussion during the discussion phase if the reviewer has any further questions or concerns.
>
> >Reviewer’s concern about the title and the empirical results
>
> The results show that the grid performs the best for (1) the bandlimited synthetic signal in 2D and 3D, (2) the real images from DIV2K for overfitting, denoising, and super-resolution, and (3) CT reconstruction. INRs do best on the synthetic signals other than the bandlimited signal (Spheres, Star Target, Sierpinski) and on the shape fitting tasks, but do not beat the grid on any "dense" signals or real signals other than the shape fitting task. Although we include multiple types of these “sparse” synthetic and shape fitting type tasks, we find they are not reflective of many real signals and on the real signals the grids are generally best.
>
> >Why is the grid-based representation better?
>
> Our results suggest that underlying lower dimensional structure (such as a 2D surface embedded in 3D, or a 1D curve or edge embedded in 2D) is key for INRs to be able to compress. For pure bandlimited noise, it makes sense that the grid is best because of the Nyquist sampling/interpolation theorem. One of the interesting things we find is that many "dense" natural signals like CT and natural images behave more like bandlimited noise than like the “sparse” synthetic signals and shapes where there is underlying lower dimensional structure like sharp edges and constant regions. We believe that there is some structure to these “dense” natural signals that should enable them to be compressed, but our work reveals that current INRs are not meeting that goal. We will add this discussion to the revised paper.
>
> >Experiments for larger model capacities
>
> Our goal is to understand when different representations are capable of achieving compression; this is different from many papers that evaluate INRs in the overparameterized regime. We focus on compression because (1) it improves memory efficiency, and (2) if a representation is compressive, it must have captured some meaningful structure in the signal and thus will double as a useful prior for reconstruction and denoising. Since all of our signals have a raw size of $10^6$ parameters, any model larger than this (including the $3\times 10^6$ size models we include) are overparameterized rather than compressive.
>
> >Performance of INGP inconsistent with community consensus
>
> We believe that the difference is due to the confluence of dense signals and compressive model size. In the literature INGP is often tested on sparse signals like shapes or radiance fields, where hash collisions often occur in empty space and thus the model can achieve high-quality compression. However, when INGP is evaluated on dense signals like natural images, it is often overparameterized (not compressive). Our results reveal that when INGP is evaluated on dense signals in the compressive regime it struggles to recover from hash collisions. We believe that this finding adds important nuance to the community understanding of INGP.

---

> > ### Comment · Reviewer_4qxt · 2025-08-08
> >
> > Thank you for your responses. However, my main concerns remain. Specifically, from Table 5 to Table 18, the grid representation does not outperform INR in many cases. Moreover, all experiments are conducted on a limited range of INR model sizes (from 1$\times$10$^4$ to 3$\times$10$^6$ parameters). Therefore, I find the claim that "grid representation often outperforms INR" unreliable, as it lacks both sufficient experimental support and theoretical justification. After reviewing comments from other reviewers, I decide to maintain my rating of weak reject.

---

> > > ### Author Response · Authors · 2025-08-08
> > >
> > > We thank the reviewer for reading our response and clarifying their remaining concerns. The reviewer points out that our experimental results are nuanced (with Grid vs INR performing best on different tasks), and that this nuance is not reflected in the title of our paper. Accordingly, we are open to changing our title to “Grids Often Outperform Implicit Neural Representations at Compressing Dense Signals”. This new title would capture two important nuances of our benchmarking study: (i) that we focus on the compressive regime, with fewer model parameters than parameters in the raw signal [which is $10^6$], and (ii) that Grids tend to dominate INRs for representing “dense” signals like our Bandlimited synthetic signals [see Figure 3] and real images and CT scans [see Figure 6], while INRs can outperform grids for “sparse” signals or signals with underlying lower-dimensional structure, like our synthetic spheres and Sierpienski signals and the 3D shape fitting tasks [see Figure 3 and Tables 5-18, except Tables 8-10 which are Bandlimited]. Please let us know if this proposed change of title addresses your concerns.

---

### Public Comment · ~Sara_Fridovich-Keil1 · 2026-06-21

We have made a few updates to the paper (mainly in the appendix) after the camera-ready; please refer to the arXiv version (https://arxiv.org/abs/2506.11139) and GitHub (https://github.com/voilalab/INR-benchmark) for these updates.

---

### Note · Authors · 2025-08-12

We thank the reviewers for their time, engagement, and helpful suggestions. We summarize our contributions and discussion clarifications below.

1. Our study focuses on compression, where the representation has fewer parameters than the raw signal. Our main finding is that a simple grid baseline delivers both the highest fidelity and the fastest training and inference time for compressing diverse “dense” signals: bandlimited noise, natural images (overfitting, denoising, and super-resolution), and computed tomography. To better reflect this main finding, we plan to change the paper title to “Grids Often Outperform Implicit Neural Representations at Compressing Dense Signals.”
2. Our results suggest the insight that some INRs can outperform grids at compressing signals that are sparse or have lower-dimensional structure, such as a 2D surface embedded in 3D or a 1D curve/edge embedded in 2D. We believe that many natural signals such as images and CT scans do have underlying structure, but this structure is not exploited for compression by existing INRs. Thus, our work points to an important avenue for future work: to develop INRs that can leverage the complex structure in dense natural signals to achieve high-fidelity compressive representation.
3. Some reviewers noted the poor performance of some INRs, notably SIREN and INGP, in contrast to their high performance in the literature. Our work adds important nuance to the literature for these models by evaluating them in the compressive regime. While these models can reach high quality when overparameterized, our study reveals that they struggle in the compressive regime, likely due to more severe hash collisions and underfitting. We note that these models can also be sensitive to hyperparameters (see below), but that this sensitivity does not account for their poor compression, as these models also struggle to compress the Star Target on which they were tuned (see Figure 10).
4. Our study is set up to test how a single, reasonable hyperparameter setting transfers across signals with different characteristics. We therefore tuned all hyperparameters on the Star Target, which spans all feature scales, and evaluated how these hyperparameter settings generalize to all of the other signals we consider. This strategy offers a realistic picture of out-of-the-box model performance, which we hope will be useful for readers who wish to apply INRs to signals and tasks that inevitably differ from those in our benchmark.

---

### Decision · Program_Chairs · 2025-09-17

**Decision:**

Accept (poster)

**Comment:**

This paper benchmarks grid-based interpolations against diverse implicit neural representations for compressing many 2D and 3D signals, and it finds grids train faster and often reach better quality on dense data while INRs shine mainly on sparse data. Before the rebuttal, R 4qxt and R xXWx questioned the novelty and title scope and leaned toward rejection, whereas R HUmV and R y2XA praised the broad experiments and leaned toward acceptance. After the rebuttal, R y2XA increased to accept, R HUmV stayed borderline accept, and the two critical reviewers kept their concerns as borderline reject due to novelty concerns.

The AC recommends acceptance - one reviewer was willing to champion the paper, reviews are overall leaning positive, and the AC further believes that this study may aid in de-mystifying neural field representations and educate the community on their strengths and weaknesses.